# Gradient matters via filament diameter-adjustable 3D printing

Huawei Qu [1,2], Chongjian Gao [1], Kaizheng Liu [1], Hongya Fu[2], Zhiyuan Liu [3,4], Paul H. J. Kouwer [5], Zhenyu Han [2] ✉ & Changshun Ruan [1,4] ✉

Gradient matters with hierarchical structures endow the natural world with excellent integrity and diversity. Currently, direct ink writing 3D printing is attracting tremendous interest, and has been used to explore the fabrication of 1D and 2D hierarchical structures by adjusting the diameter, spacing, and angle between filaments. However, it is difficult to generate complex 3D gradient matters owing to the inherent limitations of existing methods in terms of available gradient dimension, gradient resolution, and shape fidelity. Here, we report a filament diameter-adjustable 3D printing strategy that enables conventional extrusion 3D printers to produce 1D, 2D, and 3D gradient matters with tunable heterogeneous structures by continuously varying the volume of deposited ink on the printing trajectory. In detail, we develop diameter-programmable filaments by customizing the printing velocity and height. To achieve high shape fidelity, we specially add supporting layers at needed locations. Finally, we showcase multi-disciplinary applications of our strategy in creating horizontal, radial, and axial gradient structures, letter-embedded structures, metastructures, tissue-mimicking scaffolds, flexible electronics, and time-driven devices. By showing the potential of this strategy, we anticipate that it could be easily extended to a variety of filament-based additive manufacturing technologies and facilitate the development of functionally graded structures.

The hierarchical architecture of gradient structure gives natural matters unique properties[1,2], such as bone[3], bamboo[4], and wood[5]. For example, natural bone has a multi-level gradient structure that strictly follows a fractal-like organization[3]. The porous cancellous bone filled with red marrow plays a hematopoietic role; the dense cortical bone with small pores mainly provides mechanical support. Although parametrization, optimization, and machine-learning algorithms provide the capability for implementing bionic designs of gradient structures, it is still a challenge to precisely fabricate complex architectures such as pore gradients. While additive manufacturing, also known as three-dimensional (3D) printing, offers the potential for the fabrication of gradient structures, the available technologies are limited to drop-by-drop stacking (e.g., drop-on-demand printing[6]), particle bonding (e.g., selective laser sintering[7] and selective laser melting[8]), and light-curing methods (e.g., digital light processing[9] and stereolithography[10]).

Direct ink writing (DIW) is an extrusion-based 3D printing approach with enormous potential[11–19] and has received widespread attention in diverse fields including porous materials[11,15,20], silica aerogels[12,13], cell-laden constructs[21–23], tissue engineering[24–26], flexible

[1]Research Center for Human Tissue and Organ Degeneration, Institute of Biomedicine and Biotechnology, Shenzhen Institute of Advanced Technology, Chinese Academy of Sciences, Shenzhen, China. [2]School of Mechatronics Engineering, Harbin Institute of Technology, Harbin, China. [3]Research Center for Neural Engineering, Shenzhen Key Laboratory of Smart Sensing and Intelligent Systems, Shenzhen Institute of Advanced Technology, Chinese Academy of Sciences, Shenzhen, China. [4]University of Chinese Academy of Sciences, Beijing, China. [5]Institute for Molecules and Materials, Radboud University, Nijmegen, The Netherlands. ✉e-mail: hanzy@hit.edu.cn; cs.ruan@siat.ac.cn

electronics[17,18], soft robots[27,28], batteries[29], concrete-printed architectures[14], and time-driven (4D printed) devices[30,31]. Its straightforward operation, low cost, and wide variety of available inks have kept DIW 3D printing in the spotlight for nearly three decades. The traditional DIW 3D printing (traditional-3DP) strategy adopts a parallel-planar-layer slicing method (Fig. 1a, b) with a constant printing velocity $V$ and printing height $H$ (define $H$ in Supplementary Fig. 1) during the intra-layer printing process (Fig. 1a(ii, iii) and Supplementary Fig. 2), resulting in printed filaments with fixed diameter $D$ (Fig. 1a(iv)) and samples with homogeneous pore structure (Fig. 1b and Supplementary Video 1). Although the traditional-3DP strategy is universally accepted and has become the gold standard for obtaining homogeneous porous structures, the fixed printing parameters ($V$ and $H$) and constant $D$ limit the creation of gradients and other complex architectures. Current strategies to address this problem by focusing on overall variation in the filament diameter, spacing, and intersection angle between different regions (Supplementary Table 1)[32–40], face challenges in controlling gradient dimension, gradient resolution, and shape fidelity.

In this work, we report a filament diameter-adjustable 3D printing (FDA-3DP) strategy for obtaining DIW 3D printed structures with controllable gradients in pore sizes using variable filling density. For this strategy, we built a design-to-fabrication workflow from parametric model design to extrusion printing. The characteristic length scale of the extruded material (i.e., filament cross-sectional area $S$ and associated filament diameter $D = (4 S/\pi)^{0.5}$) can be precisely manipulated at all locations of the target model by customizing $V$ and $H$ on the printing trajectory. Figure 1 illustrates the conceptual differences between the traditional-3DP (Fig. 1a, b) and the FDA-3DP strategy (Fig. 1c, d). In the FDA-3DP strategy, the variable $V$ breaks the limitation of constant $D$ caused by a single-nozzle tip and a constant extrusion flow rate, and the appropriate $H$ positions the printed filaments in a straight manner (rather than, for example, the coiling effect). To avoid the collapse of specific printed 3D structures, we developed a thickness compensation mechanism for selectively adding supporting filaments between FDA layers. In all, the new FDA-3DP strategy gives access to DIW printing of gradient structures, which used to rely on other more complex and costly devices.

## Results
### Concept of FDA-3D printing
As an illustration of the concept, Fig. 1d shows the creation of a horizontal gradient structure via the FDA-3DP strategy: controlling both $V$

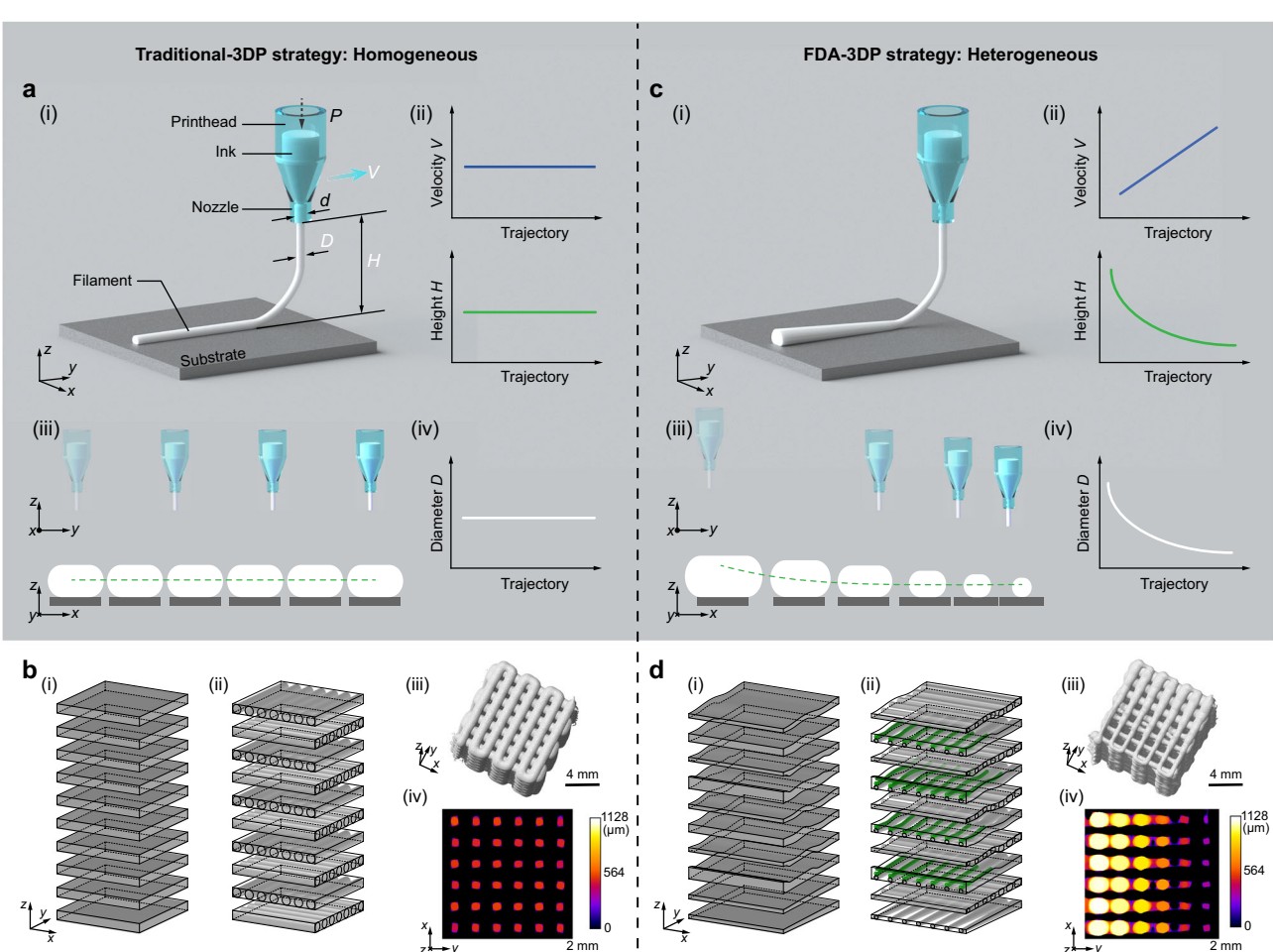

**Fig. 1 | Comparison of traditional- and FDA-3DP strategies. a** Traditional-3DP strategy with constant $V$ and $H$. $P$ is the pressure or force applied to the ink barrel; $d$ is the nozzle inner diameter; $D$ is the ideal filament diameter; $V$ is the printing velocity relative to the substrate; and $H$ is the printing height from the nozzle outlet to the substrate plane. **b** Homogeneous pore structures generated via the traditional-3DP strategy. **c** Our proposed FDA-3DP strategy with variable $V$ and $H$ and associated $D$. **d** Heterogeneous gradient pore structures created via the FDA-3DP strategy. In **a** and **c**, (i) Schematic diagram of different strategies; (ii) Plotting of $V$ and $H$ during the printing process; (iii) Illustration of nozzle movement showing changes of $V$ and $H$ in $y$-$z$ plane (upper image), and illustration of filament oblong cross section showing changes of its shape and filament cross-sectional area $S$ in the $x$-$z$ plane (lower image, see detailed information in Supplementary Fig. 3); and (iv) Plotting of $D$, where $D = (4 S/\pi)^{0.5}$. In **b** and **d**, (i) Slicing target model to obtain layer domains; (ii) Filling filament models into the slicing domains; (iii) μ-CT-based 3D reconstruction of printed structures with a cube boundary (10 mm × 10 mm × 10 mm); and (iv) a representative 2D pore size image (full series of 2D images along $z$-axis see Supplementary Video 1), where the black and colored areas are the sample material and the pore structure, respectively.

and $H$ generates a well-defined architecture. Since this printing concept yields non-parallel slices in the $x$-$y$ plane (Fig. 1d(i)), in contrast to traditional printing (Fig. 1c(i)), we compensate for potential structural collapse by introducing supporting layers (Fig. 1d(ii) and Supplementary Fig. 4, marked in green). To simplify the thickness compensation mechanism and to minimize the effect of adding supporting layers on the local pore size, we chose $D_{min}$ (minimum $D$ ranging in all FDA layers) as the constant diameter for all supporting layers. The micro-computed tomography (μ-CT) reconstruction verifies the congruence of the printed sample's dimensions (10.30 ± 0.40 mm, 10.40 ± 0.26 mm, and 9.83 ± 0.15 mm in the $x$-$y$-$z$ directions, respectively) with those of the computer-aided design (CAD) cube model (10 mm × 10 mm ×10 mm), as shown in Fig. 1d(ii, iii), Supplementary Fig. 5, and Supplementary Table 2. Additionally, it clearly illustrates the anticipated horizontal gradient in pore sizes from 0–1128 μm (Fig. 1d(iv) and Supplementary Video 1).

## Working principle

In DIW 3D printing, an increase in material volume leads to a decrease in pore size and porosity, and vice versa. To manipulate the local material volume (i.e., filament diameter $D$), the non-Newtonian fluid inks (Supplementary Fig. 6) are extruded from the nozzle tip at 100% extrusion multiplier and flow rate $Q$, and then deposited on the substrate at a printing velocity $V$. Based on fluid continuity and volume conservation law, we obtain the following equation:

$$D = (4Q/V\pi)^{0.5} \tag{1}$$

where $D$ is the filament diameter (mm), $Q$ is the extrusion flow rate (mm$^3$ s$^{-1}$), and $V$ is the printing velocity (mm s$^{-1}$). To determine $Q$ at different processing parameters (nozzle tip shape, nozzle tip's inner diameter, extrusion air pressure, and working temperature) and inks with different rheology, we developed a computational fluid dynamics (CFD) model to simulate the flow of viscoelastic ink within the nozzle and tip in DIW 3D printing. The CFD simulation calculates the average feeding velocity, defined as $v$ (mm s$^{-1}$), of viscoelastic ink at the nozzle tip's outlet. Subsequently, following the fluid flow law, the extrusion flow rate $Q$ can be determined from the CFD simulation, as indicated in the following equation:

$$Q = v \times \pi d^2 / 4 \tag{2}$$

where $Q$ is the extrusion flow rate (mm$^3$ s$^{-1}$), $v$ is the average ink feeding velocity at the nozzle tip's outlet obtained from CFD simulation (mm s$^{-1}$), $d$ is the inner diameter of the nozzle tip's outlet (mm), and π is the ratio of a circle's circumference to its diameter. As shown in Fig. 2a, b, the CFD results for the flow rate $Q$ coincide with the weighing and imaging results (Supplementary Fig. 7 and Supplementary Note 1).

Although Eq. (1) indicates that $D$ is independent of the printing height $H$, single-filament deposition state experiments with different rheological inks (L-PCL and H-PCL, see "Methods") and printing parameters using traditional-3DP strategy (Fig. 2c and Supplementary Fig. 8) revealed a significant influence of $H$ on the filament quality (e.g., coiling effects and discontinuity). This is also confirmed by previous reports from other groups[38,41]. Inspired by dimensionless quantity[38,42], we introduced dimensionless velocity $H^*$ and height $V^*$ when exploring the effect of $V$ and $H$ on the filament deposition state. Based on $H^*$ and $V^*$ in the region of the acceptable general deposition state (solid triangle) in Fig. 2c and Supplementary Fig. 9c, f, we obtained the following equation:

$$H^* = k \times (V^*)^{-0.5} \tag{3}$$

where $H^* = H/d$ is the nondimensional height, which is the printing height $H$ divided by the inner diameter $d$ of the nozzle tip's outlet.

Additionally, $V^* = V/v$ is the non-dimensional velocity, which is the printing velocity $V$ divided by the average ink feeding velocity $v$ at the nozzle tip's outlet. The coefficient k, in the range of about 0.6–1.0 (unitless coefficient), is denoted in the gray banded area in Fig. 2c. Equation (3) was validated using two viscoelastic inks with different rheological properties (H-PCL and L-PCL, Supplementary Fig. 6), considering various processing parameters (Supplementary Fig. 9). The relationship between $V$ and $D$ was derived by solving simultaneous Eqs. (1–3) (derivation see Supplementary Note 2) as follows:

$$H = k \times D \tag{4}$$

where k ranges about from 0.6 to 1.0. In this study, we selected k = 0.8 (i.e., $H = 0.8 \times D$). Notably, Equation (4) works for single-layer filaments, and for multi-layer filaments it should be $h = k \times D$ (define $H$ and $h$ in Supplementary Fig. 1).

The impact of $V$ and $H$ on the print quality was further demonstrated using 20 mm-long filaments printed with constant and variable $H$, and at two $V$ (Fig. 2d). Filaments printed with constant $H$ exhibit coiling effects (Supplementary Video 2), while filaments printed with variable $H$ remain straight (Supplementary Video 3). In short, $V$, determining filament dimensions (i.e. pore sizes), as well as $H$, controlling filament quality, are crucial parameters in the FDA-3DP strategy.

The desired printing design is stored in G-code files that are used to command extrusion-based 3D printers. As the fabrication complexity increases, the resulting G-codes file size increases, for instance, the file size increases inverse proportionally to the resolution (Fig. 2e and Supplementary Note 3), which leads to an increase in code recognition and processing time. It is important to balance the filament fabrication resolution and the size of its G-code file. To minimize the G-codes file while continuously and uniformly controlling $D$ (avoid the step effect), we FDA-3D printed 40 mm-long single filaments at different fabrication resolutions to yield filament samples with a continuously and uniformly variable $D$ with fabrication resolution down to 1.25 and 0.625 mm (Fig. 2f and Supplementary Figs. 11, 12). Since the G-codes file size at 1.25 mm is only half of the counterpart at 0.625 mm, we chose 1.25 mm as the fabrication resolution for the rest of the work.

For the parametric design of gradient pore structure models and one-click access to their fabrication G-codes file, we developed a design-to-fabrication workflow (Supplementary Fig. 13). This workflow bridges the gap from CAD digital models to extrusion 3D printers, and the G-codes files are the key link in this bridge. Before obtaining G-codes for FDA-3D printing, we performed the design of gradient pore structure models. Considering the fluid flow of extruded viscoelastic inks, the filament cross section is not ideally circular but oblong[43,44]. Accordingly, we established the relationship between filament cross-section (width $l_w$ and height $l_h$) and $V$ (Supplementary Note 4), and then created 3D gradient models with FDA and supporting layers based on the FDA-3DP strategy.

As a proof of concept, Fig. 3 illustrates the design and fabrication of the horizontal gradient pore structure in a cube (10 mm × 10 mm × 10 mm), also shown in Fig. 1d. Using the parametric design software Rhinoceros 3D and its tool Grasshopper, a grid curve trajectory and a design gradient of pore size were merged to obtain an FDA single-layer model and its corresponding printing parameters (velocity $V$ and height $H$) and filament diameter $D$ (Fig. 3a–d). Repeating the above operation generates an FDA multi-layer model and its corresponding printing parameters (Fig. 3e and Supplementary Fig. 15). However, for this horizontal gradient requirement, the generated multi-layer model suffers from a collapse defect (green part, Fig. 3f). To address this issue, a thickness compensation mechanism was developed, involving the addition of supporting layers at necessary locations. We simplified this compensation process by adopting the minimum value within the range of filament diameters produced in this work as the constant

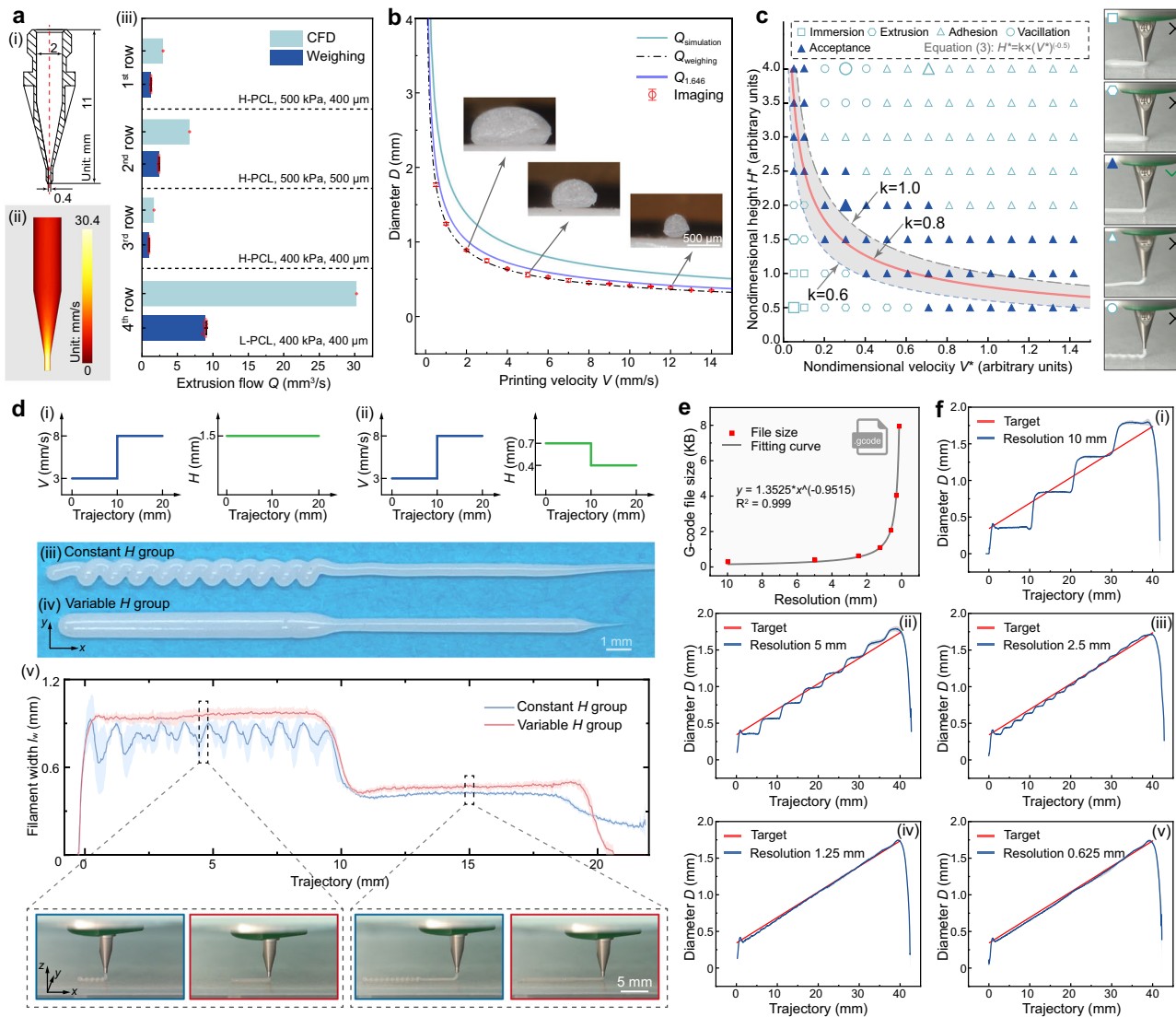

**Fig. 2 | Key parameters for FDA-3DP strategy. a** CFD simulation and weighing results of flow rate $Q$ (mean ± s.d., $n = 8$). (i) shows the profile of the nozzle tip #1 (Regenovo Co., Ltd.). In (ii), the CFD result within the needle tip is from the 1st row of subfigure (iii) (all four CFD results see Supplementary Fig. 7). In (iii), detailed process parameters are shown in Supplementary Table 3. **b** Relationship between $D$ and $V$ at the working parameters in the 1st row of Fig. 2a(iii). Three filament cross sections are from imaging results (mean ± s.d., $n = 4$; scale bar, 500 μm). Both CFD simulation and weighing results adhere to $D = (4Q/V\pi)^{0.5}$ (i.e., Eq. (1)), where $Q_{simulation}$ is 2.963 mm³ s⁻¹ (cyan) and $Q_{weighing}$ is 1.240 mm³ s⁻¹ (black dashdotted), respectively. We used $Q = 1.646$ mm³ s⁻¹ (purple) for the FDA-3DP strategy. **c** Map of the effect of $V^*$ and $H^*$ on filament deposition states at $Q_{weighing} = 1.240$ mm³ s⁻¹ (Supplementary Figs. 8, 9). Enlarged symbols on the left correspond to the deposition states on the right. The solid triangles represent an acceptable

deposition state and are fitted to obtain $H^* = k \times (V^*)^{-0.5}$. The fitted values of k range about from 0.6 to 1.0. In this work, we adopt k = 0.8 (red curve). **d** Showing the ability of our proposed strategy to maintain the filament deposition state. (i) and (ii) demonstrate the $V$ and $H$ settings of the constant and variable $H$ groups on a 40 mm-long filament, respectively. (iii) and (iv) illustrate the top views of printed filaments. (v) The upper figure quantitatively evaluates the filament deposition state by the filament cross-section width $l_w$ using software ImageJ (mean ± s.d., $n = 4$) (see processing details in Supplementary Fig. 10), and the lower figure shows the keyframes of the filament deposition during printing (frames captured from Supplementary Videos 2, 3). **e** Plotting of fabrication resolution and corresponding G-codes file size. **f** Design (target) and experimental results of variable $D$ at fabrication resolution of 10, 5, 2.5, 1.25, and 0.625 mm (mean ± s.d., $n = 4$).

diameter ($D_{min} = 0.2$ mm, corresponding to $V_{max} = 13.1$ mm s⁻¹) (Supplementary Fig. 16). The length of each supporting layer was obtained via a Boolean intersection operation between the collapse 3D model and the double-layer grid printing trajectories (Supplementary Fig. 17). Notably, the FDA layers were laminated with the supporting layers to form a collapse-free horizontal gradient pore model (Fig. 3g–i and Supplementary Fig. 18). The viscoelastic hot-melt H-PCL ink may flow due to gravity before solidifying in room-temperature air (about 25 °C), leading to a lower height of the printed sample than the intended design value (approximately 50–100% of the k value from the theoretical fitting; $k_{accept} = 0.3$–1.0). To mitigate this concern, we

strategically added FDA and supporting layers to compensate for the overall deviation in structural height, aligning with the design gradient in pore sizes. $V$ and $H$ of the FDA and supporting layers that matched the collapse-free model (see phase diagram of printing parameters in Fig. 3i) were written into a customized fabrication G-codes file (Fig. 3j and Supplementary Code 1). Then, we printed the horizontal gradient sample by running the G-codes file via our commercial extrusion 3D printer (Figs. 3k, l, 1d(iii), and Supplementary Video 4). The serial numbers of the keyframes of the FDA-3D printing process in Fig. 3l correspond to those in the phase diagram of the printing parameters in Fig. 3i.

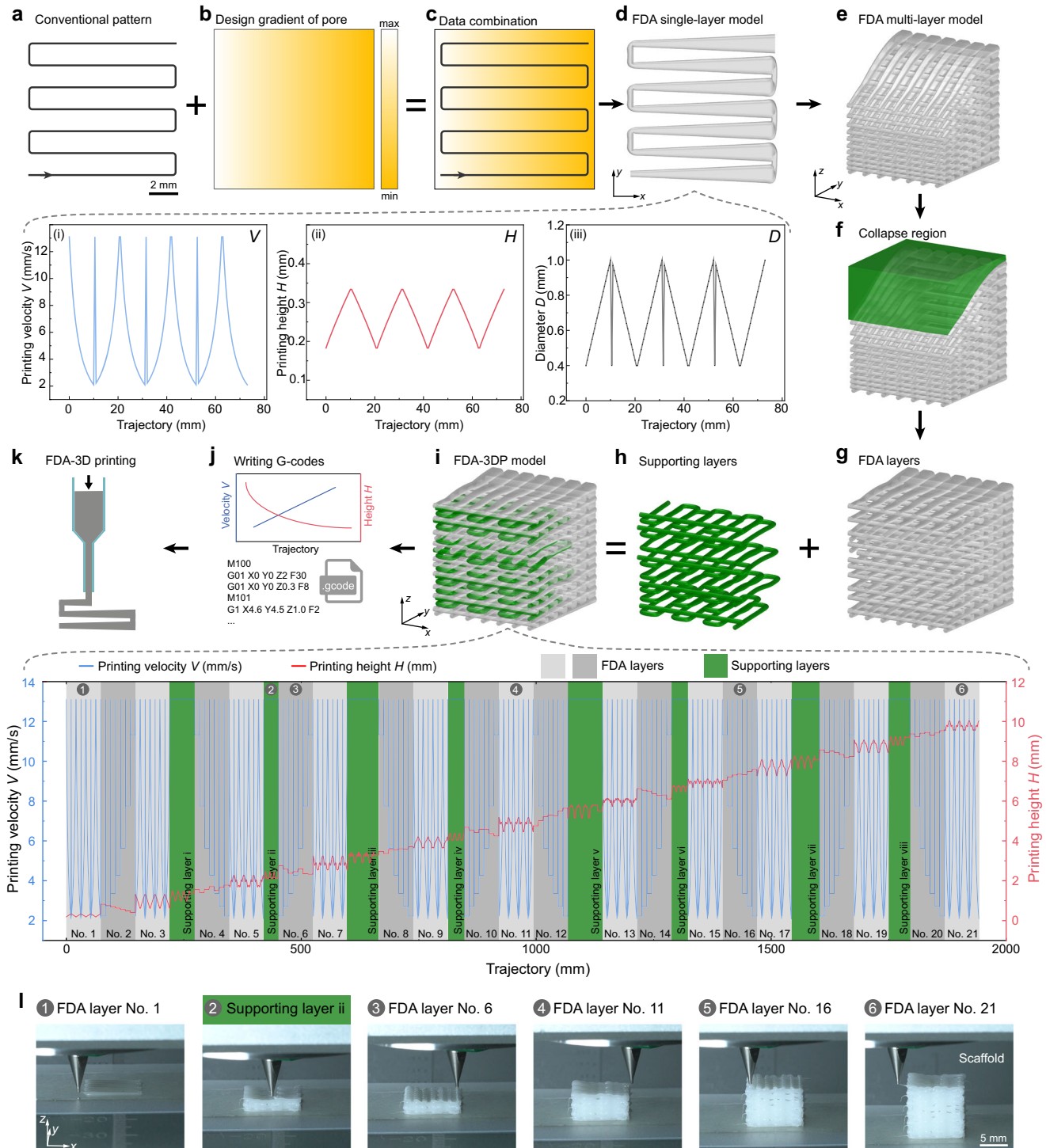

**Fig. 3 | Design and fabrication of the horizontal gradient structure via our proposed FDA-3DP strategy. a** Conventional grid pattern for 3D printing trajectory. **b** Design gradient of the pore structure. The pore size is the value of filament spacing minus filament width $l_w$ (Supplementary Fig. 3). **c** Data combination of conventional pattern and design gradient based on Eq. (1) and Supplementary Note 4. **d** FDA single-layer model with the horizontal gradient in pore sizes and the customized $V$ and $H$ (i, ii) and $D$ (iii). $Q = 1.646$ mm$^3$s$^{-1}$. To compensate for the height deviation due to ink flow caused by gravity, an acceptable coefficient $k_{accept}$ can be set to approximately 50–100% of the $k$ value from the theoretical fitting ($k_{accept} = 0.3–1.0$). **e** FDA multi-layer model obtained through layer-by-layer stacking of FDA single-layer models. **f** A collapse region existing between the target

cube and the designed FDA multi-layer model (Supplementary Fig. 4). **g–i** Acceptable FDA-3DP model without collapse (**i**) created by complementary stacking of FDA layers (**g**) and supporting layers (**h**). Customized printing parameters corresponding to the designed model are shown in the phase diagram at the bottom of (**i**). In the phase diagram, gray area, FDA layer; green area, supporting layer; blue line, $V$ and red line, $H$. **j** G-codes file obtained from the customized printing parameters in (**i**) following motion rules of the Regenovo 3D printer. **k** FDA-3D printing of the designed gradient model by executing the customized G-codes file. **l** Keyframes of FDA layers and supporting layers in the FDA-3D printing process. The serial numbers in (**l**) correspond to those in the phase diagram of printing parameters in (**i**).

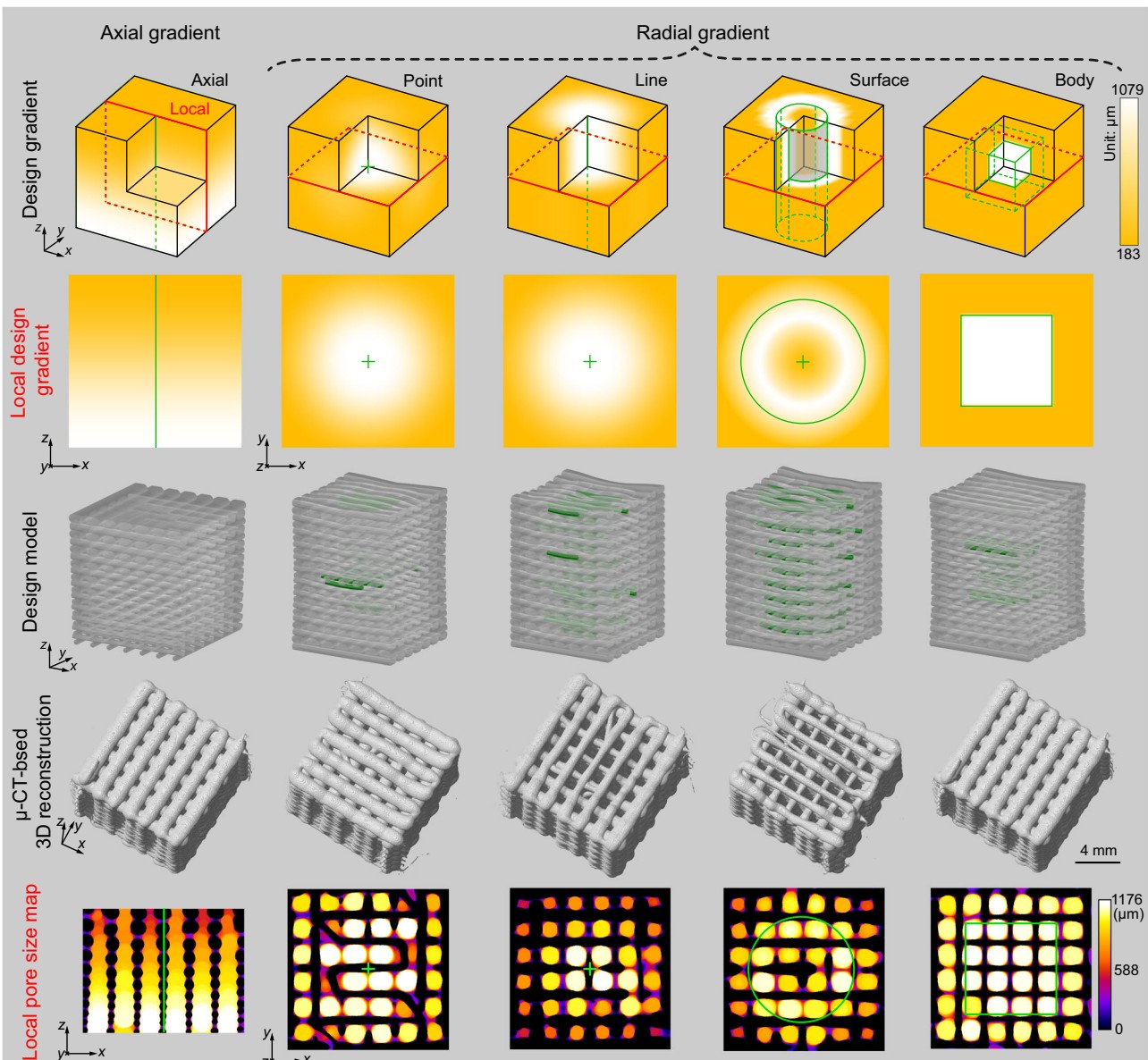

**Fig. 4 | Design, fabrication, and evaluation of axial and radial gradient pore structures.** The detailed design and printing parameters of the axial and radial gradient pore structures are shown in Supplementary Table 4. To demonstrate the internal gradient pore, the vertical and horizontal planes (enclosed by red lines, see Design gradient) are selected as local regions of interest for the axial and radial gradients, respectively. The designed 3D models were obtained based on our proposed FDA-3DP strategy using the software Rhinoceros and its embedded parametric tool Grasshopper. Pore size maps are evaluated quantitatively by the software ImageJ (full series of 2D images along the z-axis, see Supplementary Video 5), where the black and colored areas are the sample material and the pore structure, respectively.

## Applicability in complex architectures

To demonstrate the unlimited potential and advantages of the FDA-3DP strategy for creating 1D, 2D, and 3D gradient structures, we designed and fabricated axial and radial gradient pore structures (printing parameters in Supplementary Table 4, models in Supplementary Fig. 19 and Fig. 4). The pore size maps of the printed structures are in excellent agreement with the designed gradients (Fig. 4 and Supplementary Video 5). Then, we developed gradient structures showing 'HIT' (Harbin Institute of Technology, Supplementary Video 6) in three modes: through the entire sample (penetration), in the middle of the sample (centration), and rotated in the middle of the sample (centration & rotation) (Fig. 5a). The μ-CT images indicate that only the 'HIT' letters in the penetration group can be observed before slicing (Fig. 5a, middle left), which is confirmed by their pore size maps (Fig. 5a, middle right). In contrast, the 'HIT' letters in the centration and

centration & rotation experiments can be found only at the x-y horizontal plane or at the 45° oblique plane after slicing (Fig. 5a, right). In addition, we developed four V-shaped metastructures, with no, small, medium, and large gradient pores (Fig. 5b–d), confirmed by μ-CT imaging and pore size maps. Quantitative evaluation of pore sizes in the V and non-V regions reveals significant differences in all groups with gradients (Fig. 5c). Compression test results show that the deformation starts in the V region and then propagates to the non-V region, which highlights the potential of programming local mechanical properties inside 3D materials via gradient design approaches (Fig. 5d, Supplementary Figs. 20 and 21, and Supplementary Video 7).

## Multi-disciplinary applications and outlook

Recent years have shown a steeply increasing trend to combine additive manufacturing with the care research and health industry. As such,

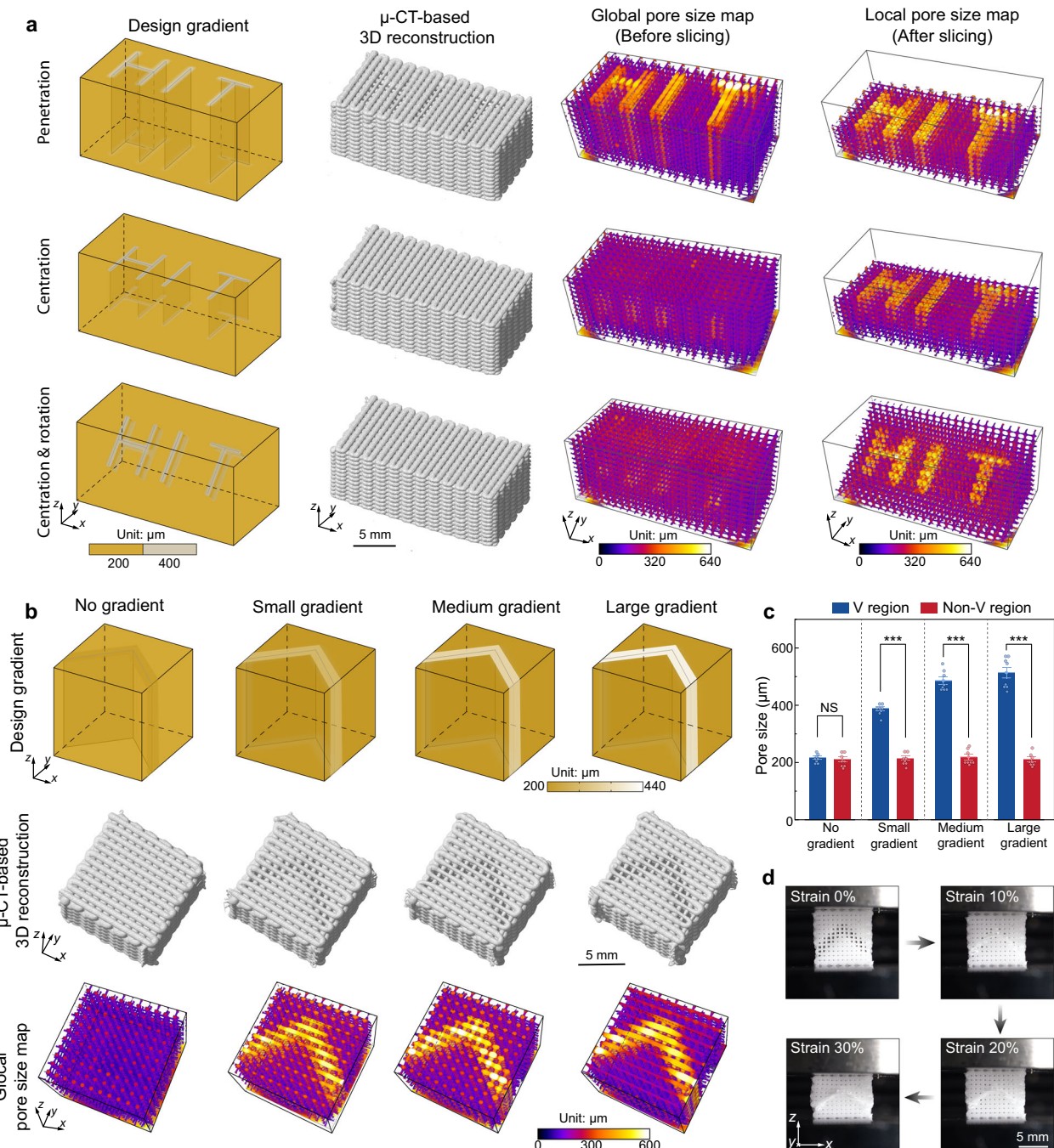

**Fig. 5 | Gradient structures with highly tunable pore size. a** Gradient structures showing 'HIT' (Harbin Institute of Technology), including penetration, centration, and centration & rotation. **b** V-shaped metastructures with no, small, medium, and large gradients. In (**a**, **b**), the design gradients, μ-CT-based 3D reconstructions of printed objects, and pore size maps before (global) and after (local) slicing are shown. **c** Pore size distributions in V and non-V regions. All gradient groups were significantly different between the V and non-V regions (means ± s.d., $n = 8$). NS indicates no significant difference. **d** Representative pictures of compression test of large gradient group. Deformation occurs first in the V region with larger size pores (in strain 0%–20%) and then extends to the non-V region with smaller size pores (in strain > 20%).

we investigated potential applications of the FDA-3DP strategy in biomedical engineering. We designed, prepared, and evaluated a bone-like scaffold that mimics the natural cancellous-to-cortical structure (Fig. 6a), and a wedge-shaped meniscus-like scaffold (Fig. 6b). Gradient pore sizes function in hierarchical tissue regeneration[24,32,45]. Pores with diameters of 150–800 μm provide channels for nutrient delivery and metabolite excretion; and smaller, 40–100 μm sized pores promote the entry and growth of non-mineralized tissues[45]. With our proposed strategy, it becomes possible to generate integrated molds with complex gradient pore structures at high fabrication resolutions (Supplementary Fig. 22). Such structures are well beyond the limitations of traditional-3DP "methods" [25,46–52] (see a comparison in Supplementary Table 5 for bone scaffolds and a comparison in Supplementary Table 6 for meniscal scaffolds). To further implement the FDA-3DP strategy into coaxial printing (Supplementary Fig. 23), we developed a gradient vascular graft with uniform straightness or controllable fluctuation (Fig. 6c and Supplementary Video 8). Additionally, a pressure sensor with variable

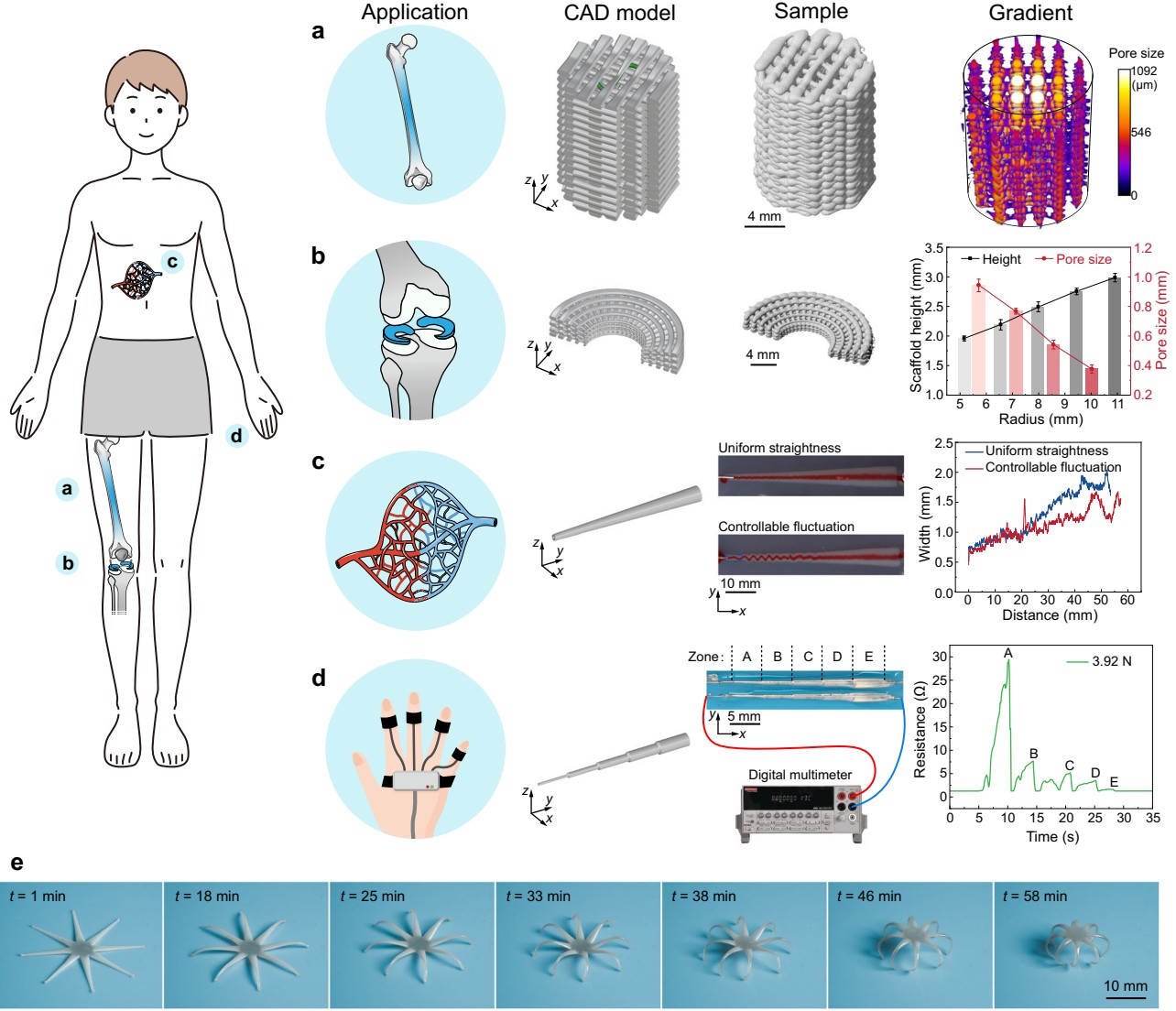

**Fig. 6 | Potential applications of gradient structures manufactured via the FDA-3DP strategy. a** Cancellous-cortical bone-mimicking scaffold with radial gradient in pore sizes. **b** Wedge-shaped meniscus-mimicking scaffold fabricated in one step without post-processing (mean ± s.d., *n* = 3). **c** Vascular scaffold with continuously controllable width and deposition state based on coaxial FDA-3D printing.

**d** Flexible electronics. A pressure sensor was obtained from liquid metal (LMs) and polydimethylsiloxane (PDMS) based on coaxial FDA-3D printing. **e** 4D printing. After calcium ion ($Ca^{2+}$) cross-linking, the octopus-like alginate sample could support itself as the water evaporates at room temperature (25 °C) (Supplementary Video 9).

*D* fabricated from a liquid metal (LM) core and a polydimethylsiloxane (PDMS) shell (Fig. 6d) indicated that our strategy can be used with a wide range of different inks. As a final example, we printed an octopus-mimicking alginate structure using the FDA-3DP strategy and achieved its 4D deformation by solvent evaporation at room temperature (25 °C) after $Ca^{2+}$ cross-linking (Fig. 6e and Supplementary Video 9).

## Discussion

We have introduced an FDA-3DP strategy that allows programming and printing filaments with variable but highly defined diameters *D* by carefully optimizing the printing velocity *V* and printing height *H*. To address localized height gaps resulting from continuously variable filament diameters, supporting layers with the constant $D_{min}$ are selectively added between different FDA layers through Boolean operations. Our developed design-to-fabrication workflow provides a platform for the parametric design of gradient pore structures and one-click access to its fabrication G-codes file[20,51]. Experimental results show that our proposed FDA-3DP strategy enables the creation of 1D, 2D, and 3D gradient pore structures using traditional DIW extrusion 3D

printers, which makes this strategy broadly available to the extrusion printing community. Despite the advancements in extrusion-based 3D printing of gradient structures using the FDA-3DP strategy, some improvements could be made. First, to simplify the compensation mechanism, the printing velocity *V* of all supporting layers in this study was uniformly set to the constant $V_{max}$ (corresponding to $D_{min}$). However, such an approach may lead to non-monotonic variations in pore gradients within the resulting structure[39]. Hence, investigating continuous FDA-3D printing without a constant $D_{min}$ throughout the printing process could prove advantageous and present an exciting avenue for future research. Second, to simplify the CFD simulation model in this study, we derived the relationship between the printing height *H* and the filament diameter *D*, expressed in Eq. (4) (i.e., $H = k \times D$). Notably, filaments resulting from the extrusion of viscoelastic ink often exhibit an oblong cross-section[43,53,54]. The development of advanced CFD simulation models for the direct calculation of the filament cross-section height $l_h$ (Supplementary Fig. 3) is a promising avenue for future exploration. On this basis, the printing height *H* in FDA-3D printing can be set to $l_h$ (i.e., $H = l_h$).

Overall, our FDA-3DP strategy breaks the limitations of the traditional-3DP strategy for which it is very difficult to design and fabricate gradient pore structures with highly controlled gradient dimensions and gradient resolution and with high shape fidelity. We foresee that the FDA-3DP strategy has promising potential in applications in diverse fields across science and engineering, especially for functionally graded/multi-material heterogeneous structures.

# Methods

## Materials
Low Mn polycaprolactone (PCL) (Aldrich, average Mn -10,000, product No. 440752), high Mn PCL (Aldrich, average Mw 45,000, product No. 704105), β-tri-calcium phosphate (β-TCP) (Sigma-Aldrich, product No. 49963), polyethylene-polypropylene glycol (Polyether F127) (Macklin, average Mn -13,000, product No. P822479), alginate (Sigma, low viscosity, product No. A1112), calcium chloride anhydrous ($Ca^{2+}$) (Aldrich, product No. C110766), gelatin from porcine skin (Vetec, product No. V900863), methacrylic anhydride (Aldrich, product No. 276685), photoinitiator 2-hydroxy-2-methyl-1-phenyl-1-propanone (Sigma-Aldrich, Irgacure 1173, product No. 405655), poly-dimethylsiloxane (PDMS) (Dowsil, SE1700 clear), and liquid metals (LMs) EGaIn (Dingguan metal technology company, China) were procured and employed for experimentation directly without any modification.

## Ink preparation
To prepare L-PCL, low Mn PCL and β-TCP powder with a 4:1 weight-to-weight ratio were heated in a beaker at 70 °C for 30 min and then stirred with a glass rod. The above steps were repeated five times. Next, the mixed material was extruded through a lab-accessible 3D printer with a nozzle tip's inner diameter of 400 μm after holding at 70 °C for 30 min. The preparation process of H-PCL was similar to that of L-PCL ink, except that L-PCL was replaced by H-PCL and the heating temperature was increased to 110 °C. The rheological properties of the two inks, L-PCL and H-PCL, were characterized in Supplementary Fig. 6.

The inks for coaxial printing include core and shell material (Supplementary Fig. 23). For vascular scaffolds (Fig. 6c), the core ink, a sacrificial material, was obtained by mixing F127 and deionized water in a 4:1 weight-to-volume ratio. The shell ink was a mixture of gelatin methacryloyl (GelMA) hydrogel and 1 wt% photoinitiator (Irgacure 1173), where GelMA was obtained from gelatin and methacrylic anhydride according to the reported method[55]. For flexible electronics (Fig. 6d), the core material is a liquid metal, and the shell material is PDMS obtained by mixing the substrate and catalyst in a 10:1 weight-to-weight ratio. The F127, GelMA, and PDMS inks were transferred into the barrel and then centrifuged at 2000 rpm for 5 min to remove air bubbles.

For the ink preparation of 4D printing (Fig. 6e), a 5:2 weight-to-weight ratio of deionized water and pure alginate was mixed homogeneously at room temperature (25 °C). The obtained mixture was transferred to the barrel and then centrifuged at 2000 rpm for 5 min.

## Rheological characterizations
The rheological properties of the inks L-PCL and H-PCL were assessed using a modular compact rheometer equipped with a 25 mm-diameter steel plate (Anton Paar, MCR302). Shear viscosity was characterized through rotation tests, employing a logarithmic sweep of controlled shear rate ($10^{-3}$–$10^3 s^{-1}$). Each ink (L-PCL and H-PCL) underwent three repetitions of the shear rate vs. viscosity test. Yield stress behavior was examined through rotation tests with a logarithmic sweep of controlled shear stress ($10^{-1}$–$10^3$ Pa). In these rotation tests (shear viscosity and yield stress), the working temperatures for the materials L-PCL and H-PCL were 70 °C and 110 °C, respectively. A linear temperature sweep in oscillatory mode was employed to assess the storage modulus $G'$ and loss modulus $G''$ of inks at a shear strain of 1% and a frequency of

1 Hz. The inks L-PCL and H-PCL were swept over the temperature ranges of from 70 °C to 25 °C and from 110 °C to 25 °C, respectively. During the temperature sweep, the temperature dropped at a rate of 1 °C per minute. The storage modulus $G'$ and loss modulus $G''$ vs. strain γ were measured using amplitude oscillatory shear sweeps from 0.01% to 3000% at a fixed angular frequency of 10 rad/s. In amplitude sweeps, the working temperatures of H-PCL and L-PCL are 110 °C and 70 °C, respectively.

## Equation derivation of basic fluid continuity
Given volume conservation, the following three equations can be obtained: (i) the volume $\Delta V_{ink}$ of ink extruded from the nozzle at unit time $\Delta t$ is $\Delta V_{ink} = Q \times \Delta t$; (ii) the volume of deposited filament at unit length $\Delta l$ is $\Delta V_{ink} = S \times \Delta l$, where $S = \pi D^2/4$ is the cross-sectional area of deposited filament, and $D$ is the ideal filament diameter; and (iii) the length of deposited filament at unit time $\Delta t$ is $\Delta l = V \times \Delta t$, where $V$ is the printing velocity. The above equations are combined to obtain Eq. (1).

## Design-to-fabrication workflow
A CAD software (Rhinoceros 3D embedded with a parametric design tool Grasshopper, Robert McNeel & Associates, USA) (https://www.rhino3d.com/), a workstation desktop computer (Precision T5820, Dell Inc., USA), a commercially available extrusion 3D printer (Regenovo Bio-Architect WS, Hangzhou Regenovo Biotechnology Co., Ltd., China), a control software (3D Bio-Architect, Hangzhou Regenovo Biotechnology Co., Ltd., China) matching the Regenovo 3D printer, and a CFD simulation software (COMSOL Multiphysics 6.0, Dassault Systèmes, USA) were used to building the design-to-fabrication workflow for the FDA-3DP strategy. As a detailed demonstration of this workflow, Fig. 3 depicts the design and fabrication of the horizontal gradient pore structure.

In the design section, first, the original grid curves (0°/90°) were drawn in the parametric software Grasshopper according to the geometry of the target model, and the results were displayed simultaneously in the working interface of the software Rhinoceros. Second, the number of aliquot points on the grid curves was determined based on the fabrication resolution of 1.25 mm. Third, the filament diameter at each point of the grid curves was determined based on the design gradient of the pore size, whereby constructing a filament-filled 3D model with FDA layers. The fourth is optional, if there was a local collapse in the model, supporting layers with a constant minimum filament diameter $D_{min}$ (relative to the range of FDA layers) were added at the needed location. The printing trajectory of the supporting layer is also the grid curve, which differs in that its length depends on Boolean intersection operations. Notably, the deposition of FDA and supporting layers was crossed, and each supporting layer was added to the FDA layers with judgment. If the thickest location of the collapsed area obtained without adding a supporting layer exceeds the minimum diameter, a supporting layer needs to be placed. Finally, the 3D model of gradient pore structures was developed by the software Grasshopper. The aliquot points, $V$, and $H$ were written into fabrication G-codes (*.gcode) files using the Grasshopper software's built-in battery (Fig. 3 and Supplementary Fig. 13).

In the fabrication section, first, the 3D printer (Regenovo Bio-Architect WS) was initialized by its control software (3D Bio-Architect). Second, after ink preparation, the gradient pore samples were manufactured by running customized fabrication G-codes files through the Regenovo 3D printer's control software (3D Bio-Architect).

## FDA-3D printing
To demonstrate the advantages of the FDA-3DP strategy in creating gradient pore structures, we designed and manufactured various gradient structures. These samples were manufactured from H-PCL ink using the extrusion 3D printer (Regenovo Bio-Architect WS) with a heating temperature of 110 °C, a nozzle inner diameter of 400 μm, and

an extrusion pressure of 500 kPa (see detailed printing parameters in Supplementary Table 4). The important 3D printing processes were recorded using a digital camera (Sony α7 III, Japan), including the coiling effect from the traditional-3DP strategy (Supplementary Video 2), the general deposition state from the FDA-3DP strategy (Supplementary Video 3), and the FDA-3D printing process of the horizontal gradient pore structure (Supplementary Video 4).

## Coaxial-based FDA-3D printing

For coaxial printing, we fabricated the vascular sample and the pressure sensor using a coaxial nozzle with an inner needle of 17 G and an outer needle of 22 G. For the vascular sample (Fig. 6c), the core and shell materials were F127 and GelMA, respectively. The printed sample was UV cross-linked for 30 s, followed by placing them in deionized water to remove the F127. The obtained sample was immersed in deionized water for 3 h and perfused using a syringe to remove F127. Finally, the perfusion experiment of the vascular scaffold was performed using a mixture of red dye and deionized water as a medium and recorded using a digital camera (Sony α7 III, Japan) (see Supplementary Video 8).

For the pressure sensor (Fig. 6d), the core and shell materials were F127 and PDMS, respectively. For PDMS curing, the printed sample was placed in an incubator and heated at 70 °C for 6 h. The obtained sample was immersed in deionized water for 3 h and perfused using a syringe to remove F127. Then, the liquid metal EGaIn was filled into the cavity of the coaxial tube, and silver wires were placed at both ends of the tube. Finally, the sensor was sealed with PDMS and cured by heating again.

## FDA-4D printing

To demonstrate the potential application of our proposed strategy in 4D printing (Fig. 6e), we designed an octopus-mimicking structure with eight 15 mm-long tentacles and prepared it using pure alginate ink. The obtained alginate sample was cross-linked using a 1.0 M calcium ion solution ($Ca^{2+}$) for 30 s. As the solvent evaporation at room temperature (25 °C), eight tentacles of the sample gradually bent downward and eventually supported the entire sample after about 1 h. Their deformation was recorded using a digital camera (Sony α7 III, Japan) (Supplementary Video 9).

## Filament cross-section quantification

Single filament samples with fixed diameters were cut using a knife, and their cross sections were photographed using a light microscope (HiROX MXB-5040RZ, Japan). We thus acquired the area $S$ and aspect ratio $AR$ of the cross sections using the software ImageJ (Fiji Is Just ImageJ, https://imagej.net/software/fiji/). The cross-section results of the μ-CT scans demonstrated a high level of agreement with those obtained by cutting with a knife, showing that the knife treatment did not induce deformation in the H-PCL samples (Supplementary Fig. 14). The filament diameter $D$ was derived from the equation $D = (4 S/\pi)^{0.5}$, where $S$ is the filament cross-sectional area obtained from knife-cutting. We named the $D$ gotten by this method as the imaging result, as shown in Fig. 2b.

## Fabrication resolution

We divided a 40 mm-long line segment into 4, 8, 16, 32, and 64 equal parts, respectively (i.e., fabrication resolution = 10, 5, 2.5, 1.25, and 0.625 mm, respectively). All 40 mm-long trajectories were given a $V$ gradient of 0.5–13.1 mm/s (Supplementary Fig. 11), resulting in $D \approx 0.347–1.736$ mm based on Eq. (1) (i.e., $D = (4Q/V\pi)^{0.5}$, where $Q_{weighing} = 1.240$ mm$^3$ s$^{-1}$). The trajectory distance as a function of the filament diameter is $D = 0.139 \times L + 0.347$ (red line, Fig. 2f), where $L$ is the trajectory distance from the current print position to the starting point. The 40 mm-long filaments printed via FDA-3DP strategy were scanned at 20 μm resolution using micro-computed tomography

(μ-CT) equipment (SCANCO, Switzerland). Then, the obtained μ-CT images were treated with the tools CTAn and DataViewer from the SkyScan 1176 software package for gray segmentation and image rotation, thereby obtaining a series of filament cross-section images. Further, all cross-sectional images were batch-processed using the software ImageJ to obtain the cross-sectional area $S$, and the filament diameter $D$ was derived from the equation $D = (4 S/\pi)^{0.5}$, as depicted in Fig. 2f.

## Pore size quantification

To quantitatively evaluate the gradient pores of printed samples, we scanned the samples at 12–20 μm resolution using μ-CT equipment (SCANCO, Switzerland). First, the obtained μ-CT images were treated with the tools CTAn and DataViewer from the SkyScan 1176 software package for gray segmentation, region of interest extraction, and x-y-z axis rotation. Note that grayscale segmentation enables the cancellation of background noise and extraction of the pore structure within the sample. Then, the above obtained x-y plane images were analyzed using the BoneJ plugin's command 'thickness' in the software ImageJ, obtaining a series of x-y plane images. These x-y plane images were used to create the 3D model of the pore size map of the printed sample in ImageJ using the command '3D Viewer', as shown in Fig. 1, Fig. 4, and Fig. 5. In addition, to further demonstrate the pore size distribution, these x-y plane images of the pore size maps were imported into software Adobe Premiere Pro CC (Adobe, USA) for generating videos showing the pore size variation in the z-axis direction (Supplementary Videos 1, 5, and 6).

## μ-CT-based 3D reconstruction

The μ-CT images of all 3D printed samples were obtained by μ-CT equipment (SCANCO, Switzerland), and handled with tools CTAn for gray segmentation and region of interest selection. Then, the handled images were rotated to a horizontal-vertical orientation. Finally, 3D models were generated via the tool CTAn using the command 'Create 3D-model'.

## Mechanical compression test of V-shaped metastructures

Metastructure samples with no, small, medium, and large gradients were subjected to uniaxial compression of 45% strain with a moving speed of 1 mm s$^{-1}$ at room temperature (25 °C) using a universal mechanical machine with a pressure sensor of 500 kg (Dongguan Zhiqu company, China). The compression process was recorded using a digital camera (Sony α7 III, Japan) (Supplementary Fig. 20 and Supplementary Video 7).

## Pressure sensor test

Resistance measurements of the pressure sensor were performed using a digital multimeter (Keithley 2000, USA) in five zones (A to E, see Fig. 6d) through an 800 g weight (Suce company, China). The pressure on the sensor can be obtained as $F = mg$, where $m$ is the mass of the weight, and $g$ is the acceleration of gravity (9.78 m s$^{-2}$). For measurement accuracy, we printed two pressure sensors in parallel, then placed a thin plate above them at the interest zone, and finally placed the weights on the plate. Notably, only the resistance of the sensor at one position was measured for each experiment. Each measurement was conducted at an interval of ~5 seconds to make the LMs flow back to the initial position.

## Measurement of flow rate

The ink flow rate $Q$ was measured by weighing ink extruded from the nozzle at a certain time ($Q = m/(\rho \times t)$), where $m$ is the weight of the extrusion ink in $t$ time, $\rho$ is the ink density, and $t$ is the extrusion time. The extrusion times for polymer inks, L-PCL and H-PCL, are denoted as $t_{L-PCL} = 3$ min and $t_{H-PCL} = 8$ min, respectively. The densities of the L-PCL and H-PCL were given by $\rho_{L-PCL} = 1.304$ g cm$^{-3}$ and $\rho_{H-PCL} = 1.217$ g cm$^{-3}$,

respectively. The flow rate $Q$ obtained by this method was called the weighing result, as shown in Fig. 2a, b. The processing parameters for the L-PCL ink included an extrusion air pressure of 400 kPa, a flow rate of $Q = 3.748$ mm$^3$ s$^{-1}$, an average feeding velocity of $v = 29.825$ mm s$^{-1}$, and the nozzle tip's inner diameter of 400 μm (nozzle tip #2 from Qingyuan Fuying Electronics Co., Ltd., China). These parameters were used in Supplementary Figs. 8c, d and 9d–f.

## CFD simulation of ink flow

The commercially available software COMSOL Multiphysics 6.0 (Dassault Systèmes, USA) was employed to develop a CFD simulation model for calculating the average ink feeding velocity $v$ at the nozzle tip's outlet under various printing parameters. The rheological data, as depicted in Supplementary Fig. 6a, illustrate that the polymer materials (H-PCL and L-PCL) display constant shear viscosity at small shear rates and low (near zero) shear viscosity at very high shear rates, consistent with the characteristics of the Carreau Yasuda model for non-Newtonian fluids. The intrinsic equation of the Carreau Yasuda model is presented below:

$$\eta = \eta_\infty + (\eta_0 - \eta_\infty) \times [1 + (\dot{\gamma}\lambda)^a]^{(n-1)/a}$$

where $\eta$, $\dot{\gamma}$, $\lambda$, $n$, and $a$ are the shear viscosity, shear rate, relaxation time, flow behavior index, and Yasuda index, respectively. $\eta_0$ and $\eta_\infty$ are the zero shear viscosity and infinite shear viscosity, respectively. The rheological data (shear viscosity vs. shear rate) were used to fit the Carreau Yasuda model, resulting in R-Square values ($R^2$, also known as the coefficient of determination) of 0.992 and 0.989 for H-PCL and L-PCL, respectively[56]. Note that the Carreau Yasuda model is only a simple approximation of the flow behavior without considering elastic contributions and normal stresses. All parameters of the Carreau Yasuda model were fitted using software OriginPro 2022 (OriginLab Corporation, USA) by the least squares (for H-PCL ink, $\eta_0 = 392.805 \pm 1.516$ Pa s, $\eta_\infty = 0 \pm 36.563$ Pa s, $\lambda = 0.010 \pm 0.003$ s, $a = 1.946 \pm 0.322$, and $n = 0 \pm 0.444$; for L-PCL ink, $\eta_0 = 36.436 \pm 0.072$ Pa s, $\eta_\infty = 0$  $15.263$ Pa s, $\lambda = 0.004 \pm 0.001$ s, $a = 4.129 \pm 1.117$, and $n = 0.319 \pm 0.551$).

Using the Geometric Drawing tool, we constructed the computational domain of the nozzle tip based on their actual dimensions. The top and bottom of the computational domain were given as the inlet and outlet, respectively. The inlet air pressure was set according to the experimental requirements (400 or 500 kPa). The remaining walls were set to be smooth and slip-free. The reference temperature was set to room temperature (25 °C). Assuming that the ink flows inside the nozzle tip without experiencing heat changes. The viscoelastic inks (L-PCL and H-PCL) were considered incompressible and continuous. The fluid flow was set to be laminar. The computational domain was meshed using regular cell sizes. In COMSOL, the flow velocity distribution inside the nozzle and tip was obtained using the 2D plot command (Supplementary Fig. 7a–d), and the flow velocity distribution at the nozzle tip's outlet was measured using the linear resultant plot command (Supplementary Fig. 7e). The average ink feeding velocity $v$ at the outlet of the nozzle tip was calculated in COMSOL using the boundary probe (Fig. 2a, b). The flow rate $Q$ can be determined by substituting the obtained average ink feeding velocity $v$ into Eq. (2) (i.e., $Q = v \times \pi d^2/4$). The flow rate $Q$ obtained in this way was called the CFD result, shown in Fig. 2a, b. In this study, consistent processing parameters were applied to all H-PCL printed samples, including Figs. 1, 2b–d, 3–5, 6a, b, Supplementary Figs. 3, 5, 8a, b, 9a–c, 10, 11, 14, and 20. These parameters included an extrusion air pressure of 500 kPa, a flow rate of $Q_{simulation} = 2.963$ mm$^3$ s$^{-1}$, an average feeding velocity of $v_{simulation} = 23.579$ mm s$^{-1}$, and a nozzle tip's inner diameter of 400 μm (nozzle tip #1, from Hangzhou Regenovo Biotechnology Co., Ltd., China).

## Statistical analysis

The commercially available software OriginPro 2022 (OriginLab Corporation, USA) was used to analyze the data in this work. All data were reported as means ± standard deviation (means ± s.d.). Data were analyzed using one-way ANOVA and Tukey's test. When *$P < 0.05$, **$P < 0.01$, and ***$P < 0.001$, the differences in statistical analysis were accepted as significant. NS indicates no significant difference.

## Data availability

All relevant data supporting the findings of this study are available within the article and its Supplementary Information files as well as Source Data. Additional datasets are available from the corresponding author upon reasonable request. Source data are provided in this paper.

## Code availability

The fabrication file (G-codes) corresponding to the horizontal gradient porous scaffold shown in Fig. 3 and Supplementary Video 4 is available within Supplementary Code 1 and through Zenodo (https://zenodo.org/records/10695518)[57].

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

## Acknowledgements

The authors gratefully acknowledge the support for this work from the National Natural Science Foundation of China (32122046, grant recipient: C.R.; 82072082, grant recipient: C.R.; and 32201097, grant recipient: K.L.), the China Postdoctoral Science Foundation (grant No. 2023M733668, grant recipient: H.Q.), the Shenzhen Fundamental Research Foundation (JCYJ20210324115814040, grant recipient: C.R.), the Shenzhen Science and Technology Program (JSGGKQTD20210831174330015, grant recipient: C.R.), and the Key Laboratory of Biomedical Imaging Science and System, Chinese Academy of Sciences (grant recipient: C.R.). Schematic diagrams of bone, joint, and blood vessels in Fig. 6a–c were generated using Servier

Medical Art, provided by Servier, licensed under a Creative Commons Attribution 3.0 unported license.

## Author contributions

H.Q. and C.R. conceived the FDA-3DP strategy for gradient matters. H.Q. designed this study. H.Q. prepared inks, developed the CFD simulation model, built the design-to-fabrication workflow, wrote customized fabrication G-codes, took videos, captured μ-CT images, printed gradient samples, reconstructed the 3D models of the samples, and analyzed the data. H.Q., C.G. and K.L. tested the rheological properties of viscoelastic inks. H.Q. wrote the manuscript with inputs from all authors (especially H.F., Z.L., Z.H. and P.H.J.K.). Z.H. and C.R. supervised the entire project.

## Competing interests

The authors declare no competing interests.
