## [Peer Review File · Nature Communications]

Gradient matters via filament diameter-adjustable 3D printingREVIEWER COMMENTS

Reviewer #1 (Remarks to the Author):

This paper reports a strategy to print gradient porous structures using DIW technology, extending the conventional extrusion molding control modes. Although the theoretical innovation and technological advancement of this article are not groundbreaking, plenty of application examples are showcased, which may inspire the manufacturing of more complex target products through continually variable process parameters.

The fundamental principle of the study is straightforward: continuously varying the key parameters H and V to print structures with gradient porous. A noteworthy merit includes establishing a comprehensive workflow and compensation mechanism to conduct rapid print path planning and product manufacture based on different requirements. The experimental details are exhaustive. However, there are still a few issues that need to be addressed, to make a substantial contribution to the extrusion-based additive manufacturing:

1. The volume conservation law is used in this paper as the fundamental guideline for adjusting the filament diameter. Besides, the ink rheology is also crucial for controlling the line morphology of the extruded ink (lots of papers have explored this phenomenon, one of the typical works was reported in ref. 37 by X Zhao). Authors fall short of addressing what the fundamental rheological characteristics of the used inks are.

2. Line 105 of the paper mentions that the optimal value for the nozzle height is $H = 0.8D$, which may only be valid for a flow rate $Q = 3.75 \text{ mm}^3/\text{s}$ (material L-PCL, nozzle diameter $400 \mu\text{m}$, air pressure 400 kPa). In the following work of the paper, is the flow rate Q kept at $3.75 \text{ mm}^3/\text{s}$? If the flow rate Q at print time is not $3.75 \text{ mm}^3/\text{s}$, is $H = 0.8D$ still valid? Is the data in Figure 2b obtained with a flow rate Q of $3.75 \text{ mm}^3/\text{s}$? If not, please provide the exact value. Is the data in Figure 2c also obtained at a flow rate Q of $3.75 \text{ mm}^3/\text{s}$? If so, please check the unit of nozzle velocity V in Figure 2c.

3. Line 128 of the paper states that it is necessary to establish the relationship between the filament cross-section and the nozzle velocity V , as the filament's cross-section is not ideally circular. However, in Supplementary Note 4, the relationship between V and AR is fitted for a flow rate $Q = 1.646 \text{ mm}^3/\text{s}$. Can this fitting formula be used in conjunction with $H = 0.8D$ ($Q = 3.75 \text{ mm}^3/\text{s}$)?

4. In my opinion, in an ideal situation, to print fibers of the general deposition type, it is sufficient to satisfy $H = D$ (which has not been experimentally verified). Nevertheless, due to gravity and the rheological properties of the material, the above condition becomes $H = \alpha D$. There is not a linear relationship between αD and D . When the nozzle height H is taken as $H = \alpha D$ (α is a constant), if the nozzle velocity V is too fast, the filament may transition from the general deposition type to the extrusion type. On the other hand, if the nozzle velocity V is too slow, the filament might change to adhesion type or even vacillation type. Is there a suitable range for $H = \alpha D$? When $H = \alpha D$, can print precision be improved? How will the cost be affected?

5. When the height of the printed structure is too high, the filaments at the bottom may flatten due to the weight, causing filaments at the top to become the adhesion type, and thus reducing print quality. In this paper, were there height constraints in the printing model? Should the effect of gravity be compensated for?

6. The application of support layers may result in non-monotonic changes in the gradient of pores in certain sections of the printed structure, potentially limiting the further enhancement of the work's innovation.

Reviewer #2 (Remarks to the Author):

REVIEW FOR NATURE COMS

For Manuscript entitled: Gradient matters via filament diameter-adjustable 3D printing, submitted by Huawei Qu et al.

The manuscript provides an interesting approach to complement DIW technology and increase the complexity of the structures that can be realised by using a programmable flow rate to create gradient structures. The images and videos are very nice, the manuscript is well presented. The

topic is of interest in a growing research field, and the structures might indeed have some potential.

It is clear the authors have put a lot of effort in the presentation of figures, images, and videos. However, there are some weaknesses in fundamental aspects that bring some concerning doubts about the underlying scientific work. In particular, the "CFD" work and the rheology of the inks should be clarified, explained, and reported in more detail.

The underlying science seems a bit obscure (feedback to the authors explained in more detail below) and therefore the manuscript in its current form, it is not strong enough to meet scientific standards with confidence. I do not recommend it for publication in Nature Communications.

Major revisions

The explanation of the CFD model is lacking, it comes a bit out of the blue. The model should be explained in more detail, and the authors should better explain what the relationships between Q and all the different working conditions that they mention, not just referring to equation 1. What is the role of the "viscoelasticity", how is it measured, how the methodology is adjusted for materials with different viscoelastic properties?

To determine Q at different working conditions (nozzle shape, nozzle inner diameter, extrusion pressure, and viscoelasticity of the ink), we established a computational fluid dynamics (CFD) model that gives excellent agreement with both weighing and imaging results of experimental samples (Fig. 2a,b and Supplementary Figure 6). Although equation (1) indicates that D is independent of H , our printing experiments (Fig. 2c and Supplementary Figure 7) and previous reports^{37,40} revealed a significant influence of H on the filament quality (e.g. coiling effects and discontinuity).

The CFD model, from the little information provided, seems that it is only considering the continuity equation inside the syringe/nozzle. However, the boundary conditions are different once the material comes out of the nozzle, and there is more to it here! That's why the equation cannot account for the influence of H . Once the material is extruded the shape of the filament will depend on the ink's rheological properties (how it recovers once it exits the nozzle). The H will also have an impact due to the gravity pull of the filament which will also be depending on the rheological parameters ("yield stress" and stiffness). It is a bit ambiguous how the "model" is used and compared with the "imaging" and "weighing".

The properties of the ink and its rheology are neither discussed and nor measured. The manuscript should: 1) refer to the importance of the inks properties in terms of rheology (there is a lot of work being done in this area, and it is shocking there is no reference to it in this work), 2) how their methodology is adjusted depending on inks properties (again providing more context here), 3) the manuscript should provide at least some evidence of the ink properties that is being printed. The materials and methods section contains some information on the properties of the ink: L-PCL (dynamic viscosity, 39 Pa s; density, 1303.5 kg/m³) and H-PCL (dynamic viscosity, 318 Pa s; density, 1216.8 kg/m³) were used as printing inks, and the inks were considered incompressible, continuous, and isotropic in the CFD simulation. The flow rate Q obtained by this method was called the CFD result, shown in Fig. 2a-b.

What does isotropic means here? Were the inks considered as Newtonian fluids with constant viscosity? Because this is obviously not the case, the materials used here do show a yield stress and shear thinning behaviour. Figure 5 in the supplementary information shows two viscosity curves for the inks, how were they measured? And how reliable are they? They are too good to be true! Viscosity is not enough to characterise the complex behaviour of DIW formulations. And it is very strange that formulations that (from the images provided) have a yield stress, and elastic components, show such a plateau in viscosity at those shear rate values.

The authors should improve this area of the work because it is directly related to the flow properties of the material and therefore to the use of Q manipulation to create gradient structures. Results in figure 2a are not enough to support and discuss the CFD work done (what's the purpose of the CFD?). It must be clarified what the "model" within the pipe (syringe/nozzle) is and it has to be clarified that the "model" is not investigating the flow and state of the material once it comes out of the nozzle.

Figure 6 in supplementary information provides some velocity profiles for different conditions. More details needs to be provided on how the velocity profiles within the nozzle are determined.

Minor comments

Typos in the text.

Check the format of the references is following the journal's system. They are all numbered before punctuation which looks odd in some places, with commas at the start of a line.

Reviewer #3 (Remarks to the Author):

The organization of the paper is confusing, with the "Materials and methods section" at the end of the document. It also has other concerns:

- What is the main objective of the paper?
- What is the novelty of the paper with respect to previous works?
- Line 89. The dimensional results of the computerized tomography should be added. What is the value of dimensional error?
- Please provide the values of the rest of the printing parameters: extrusion multiplier, infill ratio, etc.
- Line 266. The explanation about the transfer of the data from CAD model to the 3D printing machines should be explained in more detail.
- Line 307. When cutting the samples with a knife, do they suffer deformation?
- Line 380. A section with the conclusions is expected here.
- Discussion should be expanded with the comparison to other authors' results.

Point-by-point Responses to the Reviewers' Comments

Dear reviewers,

We thank all reviewers for reviewing our manuscript (entitled: *Gradient matters via filament diameter-adjustable 3D printing*). We value your constructive feedback and comments on this work. In response to your suggestions, we have carefully revised the manuscript, and the updated portions are highlighted in yellow. Below are point-by-point responses to the reviewers' comments and questions. Black indicates the original comments from the reviewers, and our responses are marked in blue.

Responses to Reviewer #1:

This paper reports a strategy to print gradient porous structures using DIW technology, extending the conventional extrusion molding control modes. Although the theoretical innovation and technological advancement of this article are not groundbreaking, plenty of application examples are showcased, which may inspire the manufacturing of more complex target products through continually variable process parameters. The fundamental principle of the study is straightforward: continuously varying the key parameters H and V to print structures with gradient porous. A noteworthy merit includes establishing a comprehensive workflow and compensation mechanism to conduct rapid print path planning and product manufacture based on different requirements. The experimental details are exhaustive. However, there are still a few issues that need to be addressed, to make a substantial contribution to the extrusion-based additive manufacturing:

Response: Thank you for acknowledging our proposed FDA-3D printing strategy and its diverse applications across disciplines. As researchers passionate about 3D printing, addressing challenges in direct ink writing (DIW) 3D printing motivates us to advance our work. DIW technology is one of the most widely used additive manufacturing technologies, and it has received attention from a variety of fields. However, limited by constant process parameters, this technology presents significant challenges in printing complex structures with gradient porosity. Inspired by the study from Prof. Xuanhe Zhao et al. (ref. 38, *Advanced Materials*, 2018, 30(6): 1704028), we confirmed that different process parameters can bring many exciting results for DIW 3D printing technology. Therefore, we developed the FDA-3D printing strategy for complex gradient pore matters through continuously varying key parameters (velocity V and height H). During FDA-3D printing, V and H control the unit volume and state of the deposited filaments, respectively. For this strategy to run smoothly, on the one hand, we established a design-to-manufacturing workflow, using the fabrication code (G-codes) as a bridge from CAD-based parametric design to DIW printing. On the other hand, to compensate for the gap in filament volume caused by variable V , we developed a compensation mechanism that selectively adds supporting filaments with a constant diameter D_{\min} (corresponding to V_{\max})

between different FDA layers. Some application examples in this paper demonstrated the feasibility of the FDA-3D printing strategy in various fields, such as tissue engineering (bone, meniscus, and blood vessel), flexible electronics, and 4D printing. We hope that this work could inspire academics working on DIW 3D printing to open new possibilities. Following your helpful comments, we have thoroughly revised our manuscript and significantly improved the quality of our work. Therefore, we sincerely ask you to re-evaluate the possibility of our work to be published in Nature Communications.

1. The volume conservation law is used in this paper as the fundamental guideline for adjusting the filament diameter. Besides, the ink rheology is also crucial for controlling the line morphology of the extruded ink (lots of papers have explored this phenomenon, one of the typical works was reported in ref. 37 by X Zhao). Authors fall short of addressing what the fundamental rheological characteristics of the used inks are.

Response: Greatly thanks for your comment, we fully agree that the ink rheology is also crucial for controlling the line morphology of the extruded ink. We sincerely apologize for the lack of fundamental rheological characteristics in the original manuscript. Previously, given our use of common and commercial DIW printing ink materials (Polycaprolactone, PCL), we exclusively examined the dynamic viscosities of the inks H-PCL and L-PCL in the range of 10^{-1} - 10^2 s^{-1} shear rate in the old Supplementary Fig. 5 in the original manuscript. Following your comment, we have added the characterization of ink rheology for L-PCL and H-PCL, including the dynamic viscosity η versus the shear rate $\dot{\gamma}$, the shear stress τ versus the shear strain ϵ , and the storage modulus G' and loss modulus G'' versus working temperature, as shown in the newly added Supplementary Fig. 6 (corresponding to the old Supplementary Fig. 5).

old Supplementary Fig. 5 | Rheology of H-PCL and L-PCL.

Indeed, the deformation, instability and fracture of the extruded ink determine the line morphology of the printed filament samples (Supplementary Fig. 9). In this work, to precisely control the volume of the extruded ink on the print trajectory, our focus was to maintain the uniform straightness of the deposited filament (acceptable general state, marked as solid triangle, see Fig. 2c and Supplementary Figs. 9-10), avoiding the coiling effect or discontinuity. Inspired by the dimensionless approach (ref. 38 by X Zhao and ref. 42 by E. J Markvicka), the relationship between printing height H and filament diameter D was newly derived by solving simultaneous Equations (1-3) as follows: $H = k \times D$ (i.e., Equation (4)), where k ranges about from 0.6 to 1.0 (unitless coefficient). In this work, we choose $k = 0.8$. The detailed derivation is provided in the revised “Supplementary Note 2” of the Supplementary Information (which focuses on responding to your Comment #2). The obtained Equation (4) (i.e., $H = 0.8 \times D$) fits all viscous inks that can be DIW 3D printed at the acceptable general filament deposition state (Supplementary Fig. 10). However, it does not mean that the rheological properties of viscous inks are not useful in DIW 3D printing (lots of publications have demonstrated this phenomenon).

In our proposed FDA-3DP strategy, Equation (1) (i.e., $D = (4Q/V\pi)^{0.5}$) is the key to customizing the gradient in pore sizes. How to calculate the extrusion flow rate Q is the next important concern. Based on the volume conservation during DIW 3D printing, Q can be obtained as follows: $Q = v \times \pi d^2 / 4$ (i.e., Equation (2)), where π is the ratio of a circle's circumference to its diameter, d is the (constant) inner diameter of nozzle tip, and v is the average ink feeding velocity. To calculate v at different processing parameters (nozzle tip shape, nozzle tip inner diameter, extrusion air pressure, and working temperature) and inks with different rheology, we established a computational fluid dynamics (CFD) model that simulates the flow of viscous ink inside the nozzle and tip. The newly revised rheological properties of the viscous ink (i.e., the dynamic viscosity η versus the shear rate $\dot{\gamma}$) were used to fit the Carreau Yasuda model, thus constructing the intrinsic equation of the non-Newtonian fluid for this CFD simulation model.

In conclusion, following your comments, the relevant figure (Supplementary Fig. 6) and experimental descriptions (*Rheological characterizations*, Page 10, Lines 291-303) have been updated in the revised manuscript and are also shown below.

New text (*Rheological characterizations*) added in the “Methods” section:

Page 10, Lines 291-303: “The rheological properties of the inks L-PCL and H-PCL were assessed using a modular compact rheometer equipped with a 25 mm-diameter steel plate (Anton Paar, MCR302). Shear viscosity was characterized through rotation tests, employing a logarithmic sweep of controlled shear rate (10^{-3} - 10^3 s $^{-1}$). Each ink (L-PCL and H-PCL) underwent three repetitions of the shear rate vs. viscosity test. Yield stress behavior was examined through rotation tests with a logarithmic sweep of controlled shear stress (10^{-1} - 10^3

Pa). In these rotation tests (shear viscosity and yield stress), the working temperatures for the materials L-PCL and H-PCL were 70 °C and 110 °C, respectively. A linear temperature sweep in oscillatory mode was employed to assess the stored modulus G' and loss modulus G'' of inks at a shear strain of 1% and a frequency of 1 Hz. The inks L-PCL and H-PCL were swept over the temperature ranges of from 70 °C to 25 °C and from 110 °C to 25 °C, respectively. During the temperature sweep, the temperature dropped at a rate of 1 °C per minute.”

New Supplementary Fig.6 revised in the Supplementary Information:

Supplementary Fig. 6 | Rheological characteristics of the polymer inks (L-PCL and H-PCL) used in this study. a, Dynamic viscosity of L-PCL and H-PCL at 70 °C and 110 °C, respectively. Both L-PCL and H-PCL exhibit non-Newtonian behavior, demonstrating the shear-thinning phenomenon at a high shear rate (about

10^2 s^{-1}). Three repetitions were conducted. The shear rate is from 10^{-3} s^{-1} to 10^3 s^{-1} . The zero-shear viscosities η_0 of L-PCL and H-PCL are approximately 382.299 Pa·s and 36.323 Pa·s, respectively. **b**, Relationship between shear stress and shear strain using rotational tests with controlled shear stress (CSS). Both L-PCL and H-PCL exhibit the yield stress. **c**, Storage G' and loss G'' modulus of the H-PCL ink for temperature scanning (from 110 °C to 25 °C) through oscillatory measurement. **d**, Storage G' and loss G'' modulus of the L-PCL ink for temperature scanning (from 70 °C to 25 °C) through oscillatory measurement. During material deformation, elasticity and viscosity are measured by the storage modulus G' and loss modulus G'' , respectively.

2.Line 105 of the paper mentions that the optimal value for the nozzle height is $H = 0.8D$, which may only be valid for a flow rate $Q = 3.75 \text{ mm}^3/\text{s}$ (material L-PCL, nozzle diameter 400 μm , air pressure 400 kPa). In the following work of the paper, is the flow rate Q kept at $3.75 \text{ mm}^3/\text{s}$? If the flow rate Q at print time is not $3.75 \text{ mm}^3/\text{s}$, is $H = 0.8D$ still valid? Is the data in Figure 2b obtained with a flow rate Q of $3.75 \text{ mm}^3/\text{s}$? If not, please provide the exact value. Is the data in Figure 2c also obtained at a flow rate Q of $3.75 \text{ mm}^3/\text{s}$? If so, please check the unit of nozzle velocity V in Figure 2c.

Response: We are grateful for your helpful suggestions to improve the quality of our manuscript. Before beginning to respond to this comment, we would like to explain that in our revised manuscript we have standardized on retaining values to three decimal places, so we have replaced $Q = 3.75 \text{ mm}^3/\text{s}$ with $Q = 3.748 \text{ mm}^3/\text{s}$. Indeed, as you mentioned, $H = 0.8 \times D$ in the original manuscript is only a special case for $Q = 3.748 \text{ mm}^3/\text{s}$. Following your comments, in order to determine whether different flow rates Q undermine the validity of $H = 0.8 \times D$, new experiments with two flow rates ($Q_{\text{L-PCL}} = 3.748 \text{ mm}^3/\text{s}$ and $Q_{\text{H-PCL}} = 1.880 \text{ mm}^3/\text{s}$) were conducted for filament deposition states at different printing velocities V and printing heights H (see newly added and Supplementary Figs. 9-10). Besides, a careful equation derivation was re-carried out, as shown in the “Results” (Pages 4-5, Lines 103-135) and “Supplementary Note 2” sections of the revised manuscript. Note that in this derivation, special attention has been given to the average feeding velocity v at the nozzle tip's outlet and the inner diameter d of the nozzle tip. We have newly introduced the nondimensional height $H^* = H/d$ and velocity $V^* = V/v$ inspired by the reported literature (ref. 38 by X Zhao and ref. 42 by E. J Markvicka). This derivation results in Equation (4) (i.e., $H = k \times D$), where k is the coefficient and in the suggested range of 0.6-1.0 (unitless coefficient). In this study, we chose $k = 0.8$ (i.e., $H = 0.8 \times D$), as shown in the newly added Fig. 2c and Supplementary Figs. 9-10. Therefore, we conclude that, in FDA-3D printing at the acceptable general deposition state, $H = 0.8 \times D$ holds for all flow rates Q obtained for various processing parameters and inks with different rheologies. In response to this comment, the detailed additions and modifications are shown below and in the revised manuscript, including new Results (Pages 4-5, Lines 103-135), new Methods (Page 14, Lines

451-455; and Page 15, Lines 488-495), Supplementary Note 2, Equations (2-4), Fig. 2c (see response Figure R1 below), and Supplementary Figs. 9-10 (Supplementary information).

In response to the flow rate Q , we apologize for not clearly demonstrating your concern in the original manuscript. In this paper, to demonstrate the generalizability of our proposed FDA-3DP strategy, we employed both L-PCL and H-PCL in Fig. 2a and Supplementary Figs. 9-10. The fluid flow rate of the H-PCL ink was $Q_{\text{H-PCL}} = 1.880 \text{ mm}^3/\text{s}$, which was used in Fig. 1, Fig. 2b-d, Fig. 3, Fig. 4, Fig. 5, Fig. 6a-b, Supplementary Fig. 3, Supplementary Fig. 5, Supplementary Fig. 9a-b, Supplementary Fig. 10a-c, Supplementary Fig. 11, Supplementary Fig. 12, Supplementary Fig. 15, and Supplementary Fig. 21. The fluid flow rate of the L-PCL ink was $Q_{\text{L-PCL}} = 3.748 \text{ mm}^3/\text{s}$, which was used in Supplementary Fig. 9c-d and Supplementary Fig. 10d-f. In the revised manuscript, we have added explanatory notes on the flow rate Q in the “*Measurement of flow rate*” (Page 14, Lines 451-455) and the “*CFD simulation of ink flow*” (Page 15, Lines 488-495) of the “Methods” section and the Supplementary Figs. 9-10 (Supplementary Information).

In response to the unit of nozzle velocity V in Fig. 2c, we checked and confirmed that there is an error in the unit for the printing velocity V in the original manuscript. Thank you for your time. The incorrect mm/min should be replaced with mm/s. We have corrected it in the revised manuscript, as shown in the new Supplementary Fig. 10e (corresponding to old Fig. 2c) in the Supplementary Information.

In summary, based on your comments, we have added and revised the following in the resubmitted manuscript.

New text revised in the “Results” section of the Main Text:

Pages 4-5, Lines 103-135: “To determine Q at different processing parameters (nozzle tip shape, nozzle tip's inner diameter, extrusion air pressure, and working temperature) and inks with different rheology, we developed a computational fluid dynamics (CFD) model to simulate the flow of viscous ink within the nozzle and tip in DIW 3D printing (Supplementary Fig. 7). The CFD simulation calculates the average feeding velocity, defined as v (mm s^{-1}), of viscous ink at the nozzle tip's outlet. Subsequently, following the fluid flow law, the extrusion flow rate Q can be determined from the CFD simulation, as indicated in the following equation:

$$Q = v \times \pi d^2 / 4 \quad (2)$$

where Q is the extrusion flow rate ($\text{mm}^3 \text{ s}^{-1}$), v is the average ink feeding velocity at the nozzle tip's outlet obtained from CFD simulation (mm s^{-1}), d is the inner diameter of the nozzle tip's outlet (mm), and π is the ratio of a circle's circumference to its diameter. As shown in Fig. 2a,b, the CFD results for the flow rate Q coincide with the weighing and imaging results (Supplementary Fig. 8 and Supplementary Note 1).

Although Equation (1) indicates that D is independent of the printing height H , single-filament deposition state experiments with different rheological inks (L-PCL and H-PCL, see

Methods) and printing parameters using traditional-3DP strategy (Fig. 2c and Supplementary Fig. 9) revealed a significant influence of H on the filament quality (e.g., coiling effects and discontinuity). This is also confirmed by previous reports from other groups^{38,41} Inspired by dimensionless quantity^{38,42}, we introduced dimensionless velocity H^* and height V^* when exploring the effect of V and H on the filament deposition state. Based on H^* and V^* in the region of the acceptable general deposition state (solid triangle) in Fig. 2c and Supplementary Fig. 10c,f, we obtained the following equation:

$$H^* = k \times (V^*)^{-0.5} \quad (3)$$

where $H^* = H/d$ is the nondimensional height, which is the printing height H divided by the inner diameter d of the nozzle tip's outlet. Additionally, $V^* = V/v$ is the nondimensional velocity, which is the printing velocity V divided by the average ink feeding velocity v at the nozzle tip's outlet. The coefficient k , in the range of about 0.6-1.0 (unitless coefficient), is denoted in the gray banded area in Fig. 2c. Equation (3) was validated using two viscous inks with different rheological properties (H-PCL and L-PCL, Supplementary Fig. 6-7), considering various processing parameters (Supplementary Fig. 10). The relationship between V and D was derived by solving simultaneous Equations (1-3) (derivation see Supplementary Note 2) as follows:

$$H = k \times D \quad (4)$$

where k ranges about from 0.6 to 1.0. In this study, we selected $k = 0.8$ (i.e., $H = 0.8 \times D$)."

New text on flow rate Q in the “Measurement of flow rate” of the “Methods” section:

Page 14, Lines 451-455: “The processing parameters for the L-PCL ink included an extrusion air pressure of 400 kPa, a flow rate of $Q = 3.748 \text{ mm}^3 \text{ s}^{-1}$, an average feeding velocity of $v = 29.825 \text{ mm s}^{-1}$, and the nozzle tip's inner diameter of 400 μm (nozzle tip #2 from Qingyuan Fuying Electronics Co., Ltd., China). These parameters were used in Supplementary Fig. 9c-d and Supplementary Fig. 10d-f.”

New text on flow rate Q in the “CFD simulation of ink flow” of the “Methods” section:

Page 15, Lines 488-495: “In this study, consistent processing parameters were applied to all H-PCL printed samples except for Fig. 2a, including Fig. 1, Fig. 2b-d, Fig. 3, Fig. 4, Fig. 5, Fig. 6a-b, Supplementary Fig. 3, Supplementary Fig. 5, Supplementary Fig. 9a-b, Supplementary Fig. 10a-c, Supplementary Fig. 11, Supplementary Fig. 12, Supplementary Fig. 15, and Supplementary Fig. 21. These parameters included an extrusion air pressure of 500 kPa, a flow rate of $Q = 1.880 \text{ mm}^3 \text{ s}^{-1}$, an average feeding velocity of $v = 14.963 \text{ mm s}^{-1}$, and a nozzle tip's inner diameter of 400 μm (nozzle tip #1, from Hangzhou Regenovo Biotechnology Co., Ltd., China).”

New text revised in the “Supplementary Note 2” of the Supplementary Information:

“In the map of filament deposition states, the acceptable general state was marked as solid

triangles (Fig. 2c and Supplementary Figs. 8-9). To establish the relationship between the printing height H and the filament diameter D , we performed the following step-by-step derivation.

(1) The transformation of Equation (2) (i.e., $Q = v \times \pi d^2 / 4$) is performed to obtain the following equation:

$$v = 4Q/\pi d^2 \quad (S1)$$

where Q is the extrusion flow rate (mm^3/s), v is the average feeding velocity through the nozzle tip (mm/s), d is the inner diameter of the nozzle tip's outlet (mm), and π (Pi) is the ratio of a circle's circumference to its diameter.

(2) $H^* = H/d$ and $V^* = V/v$ are substituted into Equation (3) (i.e., $H^* = k \times (V^*)^{-0.5}$) in order to remove the dimensionless parameters H^* and V^* . The new equation is shown below.

$$H/d = k \times (V/v)^{-0.5} \quad (S2)$$

(3) Equation (S1) (i.e., $v = 4Q/\pi d^2$) is substituted into Equation (S2) (i.e., $H/d = k \times (V/v)^{-0.5}$) to remove the average ink feeding velocity (i.e., v) through the nozzle tip. The new equation is shown below.

$$H/d = k \times [V/(4Q/\pi d^2)]^{-0.5} \quad (S3)$$

Equation (S3) above is adjusted and simplified to obtain the following equation.

$$H = k \times (4Q/V\pi)^{0.5} \quad (S4)$$

(4) Equation (1) (i.e., $D = (4Q/V\pi)^{0.5}$) is switched left and right (i.e., $(4Q/V\pi)^{0.5} = D$), and then substituted into Equation (S4) above (i.e., $H = k \times (4Q/V\pi)^{0.5}$) to remove the constant, Q , V and π . The new equation about H and D is shown below.

$$H = k \times D \quad (S5)$$

Equation (S5) above is also shown in the Results section of the main text as Equation (4).”

New Figure revised (Fig. 2c) in the Main section:

response Figure R1 | The left and right pictures are old and new Fig. 2c, respectively.

New Supplementary Figs. 9-10 revised in the Supplementary Information:

Supplementary Fig. 9 | Printing trajectories and top views of printed H-PCL and L-PCL filaments with five deposition states. **a** and **c**, Printing trajectories of Condition 1 (**a**) and Condition 2 (**c**). **b** and **d**, 3D printed single-filament samples with different V and H for Condition 1 (**b**) and Condition 2 (**d**), showing five deposition states (immersion, extrusion, acceptance, adhesion, vacillation). In Condition 1 (**a** and **b**), the print ink is H-PCL, the flow rate is $Q = 1.880 \text{ mm}^3/\text{s}$, the average feeding velocity is $v = 14.963 \text{ mm/s}$, and the inner diameter of the nozzle tip #1 (from Hangzhou Regenovo Biotechnology Co., Ltd., China) is $400 \text{ }\mu\text{m}$. In Condition 2 (**c** and **d**), the print ink is L-PCL, the flow rate is $Q = 3.748 \text{ mm}^3/\text{s}$, the average feeding velocity is $v = 29.825 \text{ mm/s}$, and the inner diameter of the nozzle tip #2 (from Qingyuan Fuying Electronics Co., Ltd., China) is $400 \text{ }\mu\text{m}$.

Supplementary Fig. 10 | Effect of printing parameters V and H on filament deposition state at different flow rates Q . **a** and **d**, Categories of filament deposition states. **b** and **e**, Key printing parameters V and H on deposition state. **c** and **f**, Nondimensional V^* and H^* on deposition state, where $V^* = V/v$ and $H^* = H/d$. **a-c**, the working condition is H-PCL ink, extrusion flow rate of $Q = 1.880 \text{ mm}^3/\text{s}$, ink feeding velocity of $v = 14.963 \text{ mm/s}$, and nozzle tip #1 (from Hangzhou Regenovo Biotechnology Co., Ltd., China) inner diameter of $400 \mu\text{m}$. **d-f**, the working condition is L-PCL ink, extrusion air pressure of 400 kPa , extrusion flow rate of $Q = 3.748 \text{ mm}^3/\text{s}$, ink feeding velocity of $v = 29.825 \text{ mm/s}$, and nozzle tip #2 (from Qingyuan Fuying Electronics Co., Ltd., China) inner diameter of $400 \mu\text{m}$. The filament deposition states include immersion, extrusion, acceptance, adhesion, and vacillation. The fitting of H^* and V^* corresponding to the acceptable deposition state results in Equation (3) (i.e., $H^* = k \times (V^*)^{-0.5}$), where k ranges about from 0.6 to 1.0 (gray banded area in sub-figure **c** and **f**). We chose $k = 0.8$ in this study.

3. Line 128 of the paper states that it is necessary to establish the relationship between the filament cross-section and the nozzle velocity V , as the filament's cross-section is not ideally circular. However, in Supplementary Note 4, the relationship between V and AR is fitted for a flow rate $Q = 1.646 \text{ mm}^3/\text{s}$. Can this fitting formula be used in conjunction with $H = 0.8D$ ($Q = 3.75 \text{ mm}^3/\text{s}$)?

Response: Thank you for your comment. In Supplementary Note 4, we established the relationship between printing velocity V and aspect ratio AR for the H-PCL ink and working condition (nozzle tip #1's inner diameter of $400 \mu\text{m}$ from Hangzhou Regenovo Biotechnology Co., Ltd., extrusion air pressure of 500 kPa , flow rate of $Q = 1.880 \text{ mm}^3/\text{s}$ from CFD simulation). Note that these two flow rates ($Q_{\text{old}} = 1.646 \text{ mm}^3/\text{s}$ and $Q_{\text{new}} = 1.880 \text{ mm}^3/\text{s}$) from the CFD simulation are different because we reconstructed the Carreau Yasuda model of the viscosity ink H-PCL (non-Newtonian fluid) and thus performed a new CFD simulation based on the reviewers #1 and #2 comments. Therefore, we replace $Q_{\text{old}} = 1.646 \text{ mm}^3/\text{s}$ with $Q_{\text{new}} = 1.880 \text{ mm}^3/\text{s}$. In this paper, for the H-PCL ink, all processing parameters are the same as described above for all positions except for Fig. 2a, including Fig. 1, Fig. 2b-d, Fig. 3, Fig. 4, Fig. 5, Fig. 6a-b, Supplementary Fig. 3, Supplementary Fig. 5, Supplementary Fig. 9a-b, Supplementary Fig. 10a-c, Supplementary Fig. 11, Supplementary Fig. 12, Supplementary Fig. 15, and Supplementary Fig. 21. Supplementary Note 4 corresponds to Supplementary Fig. 3, and we fitted the relationship between V and AR at the flow rate $Q_{\text{new}} = 1.880 \text{ mm}^3/\text{s}$. We apologize for not explaining this clearly in the original manuscript. Following your comments, we have added descriptions in the “*CFD simulation of ink flow*” (Page 15, Lines 488-495) of the “Methods” section and the “Supplementary Note 4” (marked in yellow).

As shown in responses to your Comments #1 and #2, inspired by dimensionless quantity (ref. 37 by X Zhao and ref. 41 by E. J Markvicka), we carefully re-derived the equation for the relationship between printing height H and filament diameter D (i.e., Equation (4), $H = 0.8 \times D$). When printed filaments are kept in the acceptable general deposition state (solid triangle in Fig. 2c and Supplementary Figs. 9-10), Equation (4) (i.e., $H = 0.8 \times D$) holds for all flow rates Q , and we verified it using inks L-PCL and H-PCL at $Q_{\text{L-PCL}} = 3.748 \text{ mm}^3/\text{s}$ and $Q_{\text{H-PCL}} = 1.880 \text{ mm}^3/\text{s}$, respectively (Supplementary Figs. 9-10).

New text on flow rate Q in the “*CFD simulation of ink flow*” of the “Methods” section:

Page 15, Lines 488-495: “In this study, consistent processing parameters were applied to all H-PCL printed samples except for Fig. 2a, including Fig. 1, Fig. 2b-d, Fig. 3, Fig. 4, Fig. 5, Fig. 6a-b, Supplementary Fig. 3, Supplementary Fig. 5, Supplementary Fig. 9a-b, Supplementary Fig. 10a-c, Supplementary Fig. 11, Supplementary Fig. 12, Supplementary Fig. 15, and Supplementary Fig. 21. These parameters included an extrusion air pressure of 500 kPa , a flow rate of $Q = 1.880 \text{ mm}^3 \text{ s}^{-1}$, an average feeding velocity of $v = 14.963 \text{ mm s}^{-1}$, and a nozzle tip's inner diameter of $400 \mu\text{m}$ (nozzle tip #1, from Hangzhou Regenovo Biotechnology Co., Ltd.,

China).”

4. In my opinion, in an ideal situation, to print fibers of the general deposition type, it is sufficient to satisfy $H = D$ (which has not been experimentally verified). Nevertheless, due to gravity and the rheological properties of the material, the above condition becomes $H = lh$. There is not a linear relationship between lh and D . When the nozzle height H is taken as $H = \alpha D$ (α is a constant), if the nozzle velocity V is too fast, the filament may transition from the general deposition type to the extrusion type. On the other hand, if the nozzle velocity V is too slow, the filament might change to adhesion type or even vacillation type. Is there a suitable range for $H = \alpha D$? When $H = lh$, can print precision be improved? How will the cost be affected?

Response: We thank the reviewer for these comments. Indeed, ideally, the printing height H should be set to a value equal to the filament diameter D . However, the actual DIW-based 3D printing produces filaments with an oblong cross-section due to gravity and viscous ink flow (ref. 43 by J. J. Nijdam and ref. 44 by M. P. Serdeczny).

On the one hand, there is a functional relationship between printing velocity V and filament diameter D (V vs. D). In our proposed FDA-3DP strategy, based on the volume conservation law, the pore size determines the distance between fixed-pitch parallel filaments, which defines the area S of the filament cross section and then results in D using $S = \pi D^2/4$. Based on fluid continuity, after determining the processing parameters (obtaining a constant flow rate Q from the CFD simulation), we developed the relationship between V and S (i.e., $Q = S \times V$). Substituting $S = \pi D^2/4$ into $Q = S \times V$, we got V as a function of S (i.e., Equation (1), $D = (4Q/V\pi)^{0.5}$), where Q is a constant and can be calculated using CFD simulation model. Equation (1) (i.e., $D = (4Q/V\pi)^{0.5}$) shows that D changes as V changes.

On the other hand, there is a functional relationship between printing height H and filament diameter D (H vs. D). With printed filaments maintained in an acceptable general deposition state (solid triangle in Fig. 2c), we carefully derived the relationship between printing height H and filament diameter D (i.e., Equation (4), $H = k \times D$). The detailed derivation is provided in the “Results” section (Pages 4-5, Lines 103-135) and the revised Supplementary Note 2 (Supplementary Information, which focuses on responding to your Comment #2). In $H = k \times D$, k ranges about from 0.6 to 1.0 (unitless coefficient). We chose $k = 0.8$ (i.e., $H = 0.8 \times D$) in this study.

Taken in conjunction with Equation (1) (i.e., $D = (4Q/V\pi)^{0.5}$, where Q is a constant value) and Equation (4) (i.e., $H = k \times D$, where $k = 0.8$), if the printing velocity V is too fast or too slow, the filament diameter D will change accordingly. Although k is also a constant, the printing height H will change as D changes based on Equation (4) (i.e., $H = k \times D$). In conclusion, the above theoretical basis ensures that the printed filament samples remain in the acceptable general deposition state at all printing velocities V in our proposed FDA-3D printing strategy,

as demonstrated in Fig. 3 and Supplementary Video 4.

In response to your comment about the suitable range for $H = \alpha D$, we fit the dimensionless V^* and H^* in the acceptable general deposition state, and then newly obtain Equation (3) (i.e., $H^* = k \times V^*$), where k ranges about from 0.6 to 1.0 (unitless coefficient), as shown in Fig. 2c and Supplementary Fig. 10. Based on Equations (1-3), we perform an equation derivation to obtain Equation (4) (i.e., $H = k \times D$), as indicated in the ‘‘Supplementary Note 2’’ of the Supplementary Information. The new coefficient k added in the revised manuscript is the α from the equation $H = \alpha D$ that you mentioned, which ranges about from 0.6 to 1.0 (unitless coefficient).

In response to your comment about $H = l_h$, we believe that $H = l_h$ can accurately control the amount of deposited ink and ensure the print precision, which is the purpose of our newly proposed Equation (4) (i.e., $H = k \times D$) to guide the FDA-3D printing strategy. Following your suggestion, we have derived the equation (R1) from the DIW 3D printing process, as shown below:

$$\begin{cases} V = Q/S \\ S = l_w \times l_h - l_w^2 \times (1 - \pi/4) \\ AR = l_w/l_h \\ AR = 2.457 \times V^{-0.297} \end{cases} \quad (R1)$$

where V is the printing velocity, Q is the flow rate, S is the filament cross-sectional area, l_w is the cross-sectional width, l_h is the cross-sectional height (Supplementary Fig. 3o), AR is the aspect ratio of filament cross section, and $AR = 2.457 \times V^{-0.297}$ is an empirical formula from Supplementary Fig. 3a-n. Notably, it is difficult to solve the relationship between printing velocity V and filament height l_h under the existing conditions. Establishing this relationship ($V \sim l_h$) directly through CFD simulation is an interesting project for the future. However, this might increase the expense of purchasing simulation software and additional project time. In this study, to simplify the issue, we switch the idea by newly introducing the dimensionless V^* and H^* (Equation (3), $H^* = k \times V^*$) and the filament diameter D , ultimately obtaining Equation (4) (i.e., $H = k \times D$), as shown earlier. The FDA-3D printing process of the horizontal gradient pore structure (Fig. 3 and Supplementary Video 4) shows that Equation (4) (i.e., $H = k \times D$, where $k = 0.8$ in this study) is feasible, and the printed filaments present an expected acceptable general deposition state.

In summary, in response to your comments, we have revised the Results section (Pages 4-5, Lines 103-135), Discussion section (Page 8, Line 245-251), Supplementary Note 2, Equations (2-4) and (S1-S5), and Fig. 2c (also see response Figure R1).

New text added in the ‘‘Discussion’’ section of the Main Text:

Page 8, Line 245-251: ‘‘Second, to simplify the CFD simulation model in this study, we derived

the relationship between the printing height H and the filament diameter D , expressed in Equation (4) (i.e., $H = k \times D$). It is worth noting that the filaments produced by the extrusion of viscous inks usually exhibit an oblong cross-section^{43,53,54}. The development of advanced CFD simulation models for the direct calculation of the filament cross-section height l_h (Supplementary Fig. 3) is a promising avenue for future exploration. On this basis, it is clear that $H = l_h$ will be set in FDA-3D printing.”

5. When the height of the printed structure is too high, the filaments at the bottom may flatten due to the weight, causing filaments at the top to become the adhesion type, and thus reducing print quality. In this paper, were there height constraints in the printing model? Should the effect of gravity be compensated for?

Response: We thank the reviewer for insightful comments. Indeed, as the height of the printed structure increases, the filaments at the bottom may be flattened due to the weight of the top filaments, which in turn causes the actual height of the printed structure to deviate from the designed height. This phenomenon is more evident in the processing of commonly used hydrogel materials (such as Gelatin-methacryloyl (GelMA)).

In this study, we mainly use hot-melt polymers (H-PCL at 110 °C) as inks to demonstrate our proposed FDA-3DP strategy, which transforms into solid structures upon being extruded in contact with room-temperature air (about 25 °C). β -TCP-doped polymer H-PCL was used in order to scan using μ -CT equipment and further qualitatively and quantitatively analyze the distribution of pore sizes inside the printed structures. Due to the high modulus of the polymer mixtures, we believe that the gravitational compression of the printed structure on the filaments at the bottom is negligible. Although there is theoretically no limit to the height of the fabricated structures, the actual height is less than the design height. We analyze that in this work we used the cross-sectional contours of a single-layer (only first layer) filament to design and construct the overall 3D model. Notably, the first layer contacts a perfectly flat substrate, while the remaining layers starting with the second layer are deposited onto the filaments of the previous layer. The flow of viscous hot-melt polymer mixtures before solidification due to gravity leads to a deviation between the actual and designed height of the porous gradient structures.

To address this issue, we have added an overall height compensation (Page 6, Lines 177-181). Note that this height compensation is not the local thickness compensation (in Page 3, Line 87; and Page 6, Lines 170-174) due to the buildup of variable D . Specifically, if there is a height deviation after the actual FDA-3D printing, the number of FDA layers and supporting layers is increased to compensate for the structural height given the design gradient requirement. We believe that this approach is also applicable to soft materials such as hydrogels that have anomalous structural heights due to the weight of the filaments at the top. In order to show these

clearly to the reader, we have added descriptions in the “Results” section of the Main Text (Page 6, Lines 177-181).

New text revised in the “Results” section of the Main Text:

Page 6, Lines 177-181: “The viscous hot-melt H-PCL ink may flow due to gravity before solidifying in room-temperature air (about 25 °C), leading to a lower height of the printed sample than the intended design value. To mitigate this concern, we strategically added FDA and supporting layers to compensate for the overall deviation in structural height, aligning with the design gradient in pore sizes.”

6. The application of support layers may result in non-monotonic changes in the gradient of pores in certain sections of the printed structure, potentially limiting the further enhancement of the work's innovation.

Response: Thanks for your comment. Support layers were optionally added between different FDA layers by Boolean intersection operations in order to infill the local gap in height caused by the continuously variable filament diameter D (i.e., printing velocity V). To simplify this compensation mechanism, in this study, a constant diameter was given to all support layer filaments. Further, to minimize the possible influence of the application of the support layers on the pore gradient, we chose the minimum value in the range of filament diameters used in this work (D_{\min} , corresponding to V_{\max}) as the constant diameter.

Indeed, as you commented, the support layers with the constant D_{\min} in this work might affect the gradient in pore sizes based on the volume conservation law, as shown in Supplementary Fig. 17. Giving the support layers continuously variable process parameters like the FDA layers is an interesting project, and was planned as such early on. However, the support layers with varying parameters significantly increase the difficulty and workload of this study. Therefore, we simplified the compensation mechanism using the constant D_{\min} . Currently, the pore size maps of printed gradient matters (e.g., Supplementary Videos 1, 5, and 6) showed that this mechanism was effective for all the complex gradient porous structures demonstrated in this study.

Thank you for this constructive concern. We think that, in the future, all continuous FDA-3D printing without the support layers with D_{\min} throughout the printing process is an exciting but challenging project, which will completely incorporate the support layers into the FDA layers. The above project is the next step in our plan. Following your comment, we have newly added a discussion (Page 8, Lines 238-245) in the “Discussion” section to inform the reader of potential limitations with the use of support layers and to outlook interesting future work based on our study.

New text added in the “Discussion” section of the Main Text:

Page 8, Lines 238-245: “Despite the advancements in extrusion-based 3D printing of gradient structures using the FDA-3DP strategy, some improvements could be made. First, to simplify the compensation mechanism, the printing velocity V of all supporting layers in this study was uniformly set to the constant V_{\max} (corresponding to D_{\min}). However, such an approach may lead to non-monotonic variations in pore gradients within the resulting structure³⁹. Hence, investigating continuous FDA-3D printing without a constant D_{\min} throughout the printing process could prove advantageous and present an exciting avenue for future research.”

New Supplementary Fig. 17 added in the Supplementary Information:

Supplementary Fig. 17 | Supporting layers with a constant diameter (D_{\min}) obtained using the simplified compensation mechanism in this study. To simplify the process and reduce workload, we chose the minimum value in the range of filament diameters produced by this work (D_{\min} , corresponding to V_{\max}) as the constant diameter. In this illustration, the filament model (green) of the 1st supporting layer is stacked onto three FDA layers (light gray) with a horizontal gradient in pore sizes. **a**, 3D view. **b**, Front view. **c**, Top view and its partial enlargement.

Responses to Reviewer #2:

For Manuscript entitled: Gradient matters via filament diameter-adjustable 3D printing, submitted by Huawei Qu et al. The manuscript provides an interesting approach to complement DIW technology and increase the complexity of the structures that can be realised by using a programmable flow rate to create gradient structures. The images and videos are very nice, the manuscript is well presented. The topic is of interest in a growing research field, and the structures might indeed have some potential. It is clear the authors have put a lot of effort in the presentation of figures, images, and videos. However, there are some weaknesses in fundamental aspects that bring some concerning doubts about the underlying scientific work. In particular, the “CFD” work and the rheology of the inks should be clarified, explained, and reported in more detail. The underlying science seems a bit obscure (feedback to the authors explained in more detail below) and therefore the manuscript in its current form, it is not strong enough to meet scientific standards with confidence. I do not recommend it for publication in Nature Communications.

Response: We thank you for the constructive comments on our manuscript. Direct ink writing (DIW) 3D printing has received widespread attention in diverse fields thanks to its strategic simplicity, low cost, and plenty of available inks. However, DIW technology is difficult to produce complex gradient structures. In this manuscript, we developed a filament diameter-adjustable 3D printing (FDA-3DP) strategy that enables conventional 3D printers to print complex porous matters by continuously varying the printing velocities V and heights H . The proverb says that a picture is worth a thousand words. We aimed to vividly illustrate the goals, principles, and results of our work to the reader through interesting figures and videos. We sincerely thank the reviewer for appreciating our efforts.

As you say, indeed, our original manuscript needs further improvement. Following your comments and suggestions, we have carefully revised our manuscript and significantly improved the quality of our work. On the one hand, to address your concerns about the rheological characterizations of inks, we have analyzed in detail the rheological characterizations of the viscous inks (L-PCL and H-PCL) used in this work, and added a new picture (see Supplementary Fig. 6) for them. Besides, we have added a textual description of the additional rheological experiments, as shown in the “*Rheological characterizations*” of the “Methods” section (Page 10, Lines 291-303) in the revised manuscript. On the other hand, to address your concerns about the computational fluid dynamics (CFD) simulation model, the newly added rheological properties of inks L-PCL and H-PCL (i.e., dynamic viscosity η versus shear rate $\dot{\gamma}$) were used to fit the Carreau Yasuda model, and thus construct the intrinsic equation of the non-Newtonian fluid. On this basis, we re-developed the CFD simulation model to simulate the fluid flow of viscous inks within the nozzle and tip. This CFD model outputs the average feeding velocity v at the nozzle tip's outlet, and then the obtained v was used to calculate

the flow rate Q of ink using the newly added Equation (2) (i.e., $Q = v \times \pi d^2 / 4$, where d is the inner diameter of the nozzle tip's outlet). We have added descriptions about the revised CFD simulation in the new “Results” section (Pages 4-5, Lines 103-115) and the “*CFD simulation of ink flow*” (Pages 14-15, Lines 457-495) of the “Methods” section.

Thanks for your time. We sincerely ask you to re-evaluate the possibility of our work to be published in Nature Communications.

Major revisions

Q1: The explanation of the CFD model is lacking, it comes a bit out of the blue. The model should be explained in more detail, and the authors should better explain what the relationships between Q and all the different working conditions that they mention, not just referring to equation 1. What is the role of the “viscoelasticity”, how is it measured, how the methodology is adjusted for materials with different viscoelastic properties?

To determine Q at different working conditions (nozzle shape, nozzle inner diameter, extrusion pressure, and viscoelasticity of the ink), we established a computational fluid dynamics (CFD) model that gives excellent agreement with both weighing and imaging results of experimental samples (Fig. 2a,b and Supplementary Figure 6). Although equation (1) indicates that D is independent of H , our printing experiments (Fig. 2c and Supplementary Figure 7) and previous reports^{37,40} revealed a significant influence of H on the filament quality (e.g. coiling effects and discontinuity).

Response: We thank the reviewer for the comments and for kindly copying relevant information about these comments from the original manuscript, as shown in the gray text above.

In response to the reviewer's comments about the computational fluid dynamics (CFD) model, we have described it in more detail and added new formulas and textual descriptions in the “Results” section of the Main Text (Page 4, Lines 105-115) and the “*CFD simulation of ink flow*” of the “Methods” section (Pages 14-15, Lines 457-495) in the revised manuscript. We also provide the following explanation in this response letter.

Specifically, Equation (1) (i.e., $D = (4Q/V\pi)^{0.5}$) shows that there is a functional relationship between printing velocity V and filament diameter D . For DIW-based 3D printing, the processing parameters (such as nozzle tip shape, nozzle tip's inner diameter, and extrusion air pressure) and the ink rheology determine the value of the flow rate Q in Equation (1). Although it is possible to calculate Q through formulaic derivation, this is laborious and difficult. Based on fluid continuity and volume conservation law, the flow rate Q (mm^3/s) is the product of the average feeding velocity (defined as v , mm/s) of the viscous ink being extruded from the nozzle tip and the nozzle tip's cross-sectional area (defined as S , mm^2), as shown in newly added Equation (2) (i.e., $Q = v \times \pi d^2 / 4$). d is the inner diameter of the nozzle tip's outlet, which is

determined after selecting the nozzle tip. To calculate the average feeding velocity v at the nozzle tip's outlet, the CFD simulation model was developed using the commercial software COMSOL Multiphysics 6.0. The inputs to this simulation include the processing parameters and ink rheology mentioned above. This CFD simulation model is set up as follows (Pages 14-15, Lines 456-494):

- (1) The computational domain of the nozzle and tip was constructed in the software COMSOL Multiphysics 6.0 using the Geometric Drawing tool based on its actual dimensions (shape and inner diameter).
- (2) The computational domain was meshed in COMSOL using regular cell sizes.
- (3) The reference temperature was set to room temperature (25 °C). It was assumed that the ink flows inside the nozzle tip without heat change.
- (4) The viscous inks (L-PCL and H-PCL) were considered incompressible, continuous, and isotropic. L-PCL and H-PCL were simulated using the Carreau Yasuda model (non-Newtonian fluid).
- (5) The top and bottom of the computational domain were given as the inlet and outlet, respectively. The inlet air pressure was set based on the experimental requirements (400 or 500kPa). The remaining walls were set to be smooth and slip-free.
- (6) The flow velocity distribution inside the nozzle and tip was obtained using the 2D plot command (Supplementary Fig. 8a-d), and the flow velocity distribution at the nozzle tip's outlet was measured using the linear resultant plot command (Supplementary Fig. 8e).
- (7) The average ink feeding velocity v at the outlet of the nozzle tip was calculated in COMSOL using the boundary probe (further calculating Q based on Equation (2)).

The flow rate Q can be determined by substituting the average ink feeding velocity v obtained from the CFD simulation into Equation (2) (i.e., $Q = v \times \pi d^2 / 4$). Finally, the obtained flow rate Q is used in Equation (1) (i.e., $D = (4Q / V\pi)^{0.5}$) which is fundamental to the FDA-3DP strategy.

In response to the “viscoelasticity” of the print ink, following your suggestion, we have analyzed in detail the rheological characterizations of the polymer inks (L-PCL and H-PCL), including dynamic viscosity η versus shear rate γ (Supplementary Fig. 6a), shear stress τ versus shear strain ε (Supplementary Fig. 6b), and storage modulus G' and loss modulus G'' versus working temperature (Supplementary Fig. 6c). Besides, we have added a textual description of the rheological experiments, as shown in the “*Rheological characterizations*” of the “Methods” section (Page 10, Lines 291-303) in the revised manuscript. We also provide the following explanation in this response letter.

Specifically, thick materials for DIW 3D printing (such as hot-melt polymers and hydrogels) not only exhibit the viscous flow properties possessed by liquids, but also show the elastic properties possessed by solids. Some non-Newtonian fluids are both viscous and elastic,

and one calls such materials viscoelastic. During material deformation, elasticity and viscosity are measured by the storage modulus G' and loss modulus G'' , respectively. The storage modulus G' and loss modulus G'' of inks L-PCL and H-PCL were performed using a modular compact rheometer with a 25 mm-diameter steel plate (Anton Paar, MCR302) at a shear strain of 1% and a frequency of 1 Hz, as displayed in Supplementary Fig. 6c-d. We thank the reviewer for this constructive comment. In order to express more rigorously, we have replaced viscoelasticity with viscosity in the revised manuscript.

New text revised in the “Results” section of the Main Text (Detailed description for CFD simulations):

Page 4, Lines 105-115: “..., we developed a computational fluid dynamics (CFD) model to simulate the flow of viscous ink within the nozzle and tip in DIW 3D printing (Supplementary Fig. 7). The CFD simulation calculates the average feeding velocity, defined as v (mm s^{-1}), of viscous ink at the nozzle tip's outlet. Subsequently, following the fluid flow law, the extrusion flow rate Q can be determined from the CFD simulation, as indicated in the equation below:

$$Q = v \times \pi d^2 / 4 \quad (2)$$

where Q is the extrusion flow rate ($\text{mm}^3 \text{s}^{-1}$), v is the average ink feeding velocity at the nozzle tip's outlet obtained from CFD simulation (mm s^{-1}), d is the inner diameter of the nozzle tip's outlet (mm), and π (Pi) is the ratio of a circle's circumference to its diameter. As shown in Fig. 2a,b, the CFD results for the flow rate Q coincide with the weighing and imaging results (Supplementary Fig. 8 and Supplementary Note 1).”

New text (*Rheological characterizations*) added in the “Methods” section:

Page 10, Lines 291-303: “The rheological properties of the inks L-PCL and H-PCL were assessed using a modular compact rheometer equipped with a 25 mm-diameter steel plate (Anton Paar, MCR302). Shear viscosity was characterized through rotation tests, employing a logarithmic sweep of controlled shear rate (10^{-3} - 10^3 s^{-1}). Each ink (L-PCL and H-PCL) underwent three repetitions of the shear rate vs. viscosity test. Yield stress behavior was examined through rotation tests with a logarithmic sweep of controlled shear stress (10^{-1} - 10^3 Pa). In these rotation tests (shear viscosity and yield stress), the working temperatures for the materials L-PCL and H-PCL were 70 °C and 110 °C, respectively. A linear temperature sweep in oscillatory mode was employed to assess the stored modulus G' and loss modulus G'' of inks at a shear strain of 1% and a frequency of 1 Hz. The inks L-PCL and H-PCL were swept over the temperature ranges of from 70 °C to 25 °C and from 110 °C to 25 °C, respectively. During the temperature sweep, the temperature dropped at a rate of 1 °C per minute.”

New text (*CFD simulation of ink flow*) added in the “Methods” section of the Main Text:

Pages 14-15, Lines 457-495: “The commercially available software COMSOL Multiphysics 6.0 (Dassault Systèmes, USA) was employed to develop a CFD simulation model for

calculating the average ink feeding velocity v at the nozzle tip's outlet under various printing parameters. The rheological data, as depicted in Supplementary Fig. 6a, illustrate that the polymer materials (H-PCL and L-PCL) display constant shear viscosity at small shear rates and low (near zero) shear viscosity at very high shear rates, consistent with the characteristics of the Carreau Yasuda model for non-Newtonian fluids. The intrinsic equation of the Carreau Yasuda model is presented below:

$$\eta = \eta_{\infty} + (\eta_0 - \eta_{\infty}) \times [1 + (\gamma\lambda)^a]^{(n-1)/a}$$

where η , γ , λ , n , and a are the shear viscosity, shear rate, relaxation time, flow behavior index, and Yasuda index, respectively. η_0 and η_{∞} are the zero shear viscosity and infinite shear viscosity, respectively. The rheological data (shear viscosity vs. shear rate) were used to fit the Carreau Yasuda model, resulting in R-Square values (R^2 , also known as the coefficient of determination) of 0.943 and 0.807 for H-PCL and L-PCL, respectively⁵³. The Carreau Yasuda model parameters for inks H-PCL ($\lambda=0.051$, $n=0.802$, $a=217.786$, $\eta_0=382.299$ Pa s, and $\eta_{\infty}=0$ Pa s) and L-PCL ($\lambda=0.014$, $n=0.840$, $a=217.786$, $\eta_0=36.323$ Pa s, and $\eta_{\infty}=0$ Pa s) are illustrated in Supplementary Fig. 7.

Using the Geometric Drawing tool, we constructed the computational domain of the nozzle tip based on their actual dimensions. The top and bottom of the computational domain were given as the inlet and outlet, respectively. The inlet air pressure was set according to the experimental requirements (400 or 500 kPa). The remaining walls were set to be smooth and slip-free. The reference temperature was set to room temperature (25 °C). Assuming that the ink flows inside the nozzle tip without experiencing heat changes. The viscous inks (L-PCL and H-PCL) were considered incompressible, continuous, and isotropic. The fluid flow was set to be laminar. The computational domain was meshed using regular cell sizes. In COMSOL, the flow velocity distribution inside the nozzle and tip was obtained using the 2D plot command (Supplementary Fig. 8a-d), and the flow velocity distribution at the nozzle tip's outlet was measured using the linear resultant plot command (Supplementary Fig. 8e). The average ink feeding velocity v at the outlet of the nozzle tip was calculated in COMSOL using the boundary probe (Fig. 2a,b). The flow rate Q can be determined by substituting the obtained average ink feeding velocity v into Equation (2) (i.e., $Q = v \times \pi d^2 / 4$). The flow rate Q obtained in this way was called the CFD result, shown in Fig. 2a-b. In this study, consistent processing parameters were applied to all H-PCL samples except for Fig. 2a, including Fig. 1, Fig. 2b-f, Fig. 3, Fig. 4, Fig. 5, Fig. 6a-b, Supplementary Fig. 3, Supplementary Fig. 5, Supplementary Fig. 9a-b, Supplementary Fig. 10a-c, Supplementary Fig. 11, Supplementary Fig. 12, Supplementary Fig. 17, and Supplementary Fig. 23. These parameters included an extrusion air pressure of 500 kPa, a flow rate of $Q = 1.880 \text{ mm}^3 \text{ s}^{-1}$, an average feeding velocity of $v = 14.963 \text{ mm s}^{-1}$, and a nozzle tip's inner diameter of 400 μm (nozzle tip #1, from Hangzhou Regenovo Biotechnology Co., Ltd., China)."

New Supplementary Fig. 6 added in the Supplementary Information (see next page):

Supplementary Fig. 6 | Rheological characteristics of the polymer inks (L-PCL and H-PCL) used in this study. **a**, Dynamic viscosity of L-PCL and H-PCL at 70 °C and 110 °C, respectively. Both L-PCL and H-PCL exhibit non-Newtonian behavior, demonstrating the shear-thinning phenomenon at a high shear rate (about $10^2 s^{-1}$). Three repetitions were conducted. The shear rate is from $10^{-3} s^{-1}$ to $10^3 s^{-1}$. The zero-shear viscosities η_0 of L-PCL and H-PCL are approximately 382.299 Pa·s and 36.323 Pa·s, respectively. **b**, Relationship between shear stress and shear strain using rotational tests with controlled shear stress (CSS). Both L-PCL and H-PCL exhibit the yield stress. **c**, Storage G' and loss G'' modulus of the H-PCL ink for temperature scanning (from 110 °C to 25 °C) through oscillatory measurement. **d**, Storage G' and loss G'' modulus of the L-PCL ink for temperature scanning (from 70 °C to 25 °C) through oscillatory measurement. During material

deformation, elasticity and viscosity are measured by the storage modulus G' and loss modulus G'' , respectively.

Q2: The CFD model, from the little information provided, seems that it is only considering the continuity equation inside the syringe/nozzle. However, the boundary conditions are different once the material comes out of the nozzle, and there is more to it here! That's why the equation cannot account for the influence of H . Once the material is extruded the shape of the filament will depend on the ink's rheological properties (how it recovers once it exist the nozzle). The H will also have an impact due to the gravity pull of the filament which will also be depending on the rheological parameters ("yield stress" and stiffness). It is a bit ambiguous how the "model" is used and compared with the "imaging" and "weighing".

Response: Thank you for your comments. We apologize for not explaining the CFD simulation model clearly in the original manuscript. Indeed, as you say, this CFD model in our paper is only used to simulate the fluid flow of viscous ink inside the nozzle tip, and to calculate the average ink feeding velocity v at the nozzle tip's outlet after determining the processing parameters and ink rheology. Following the reviewers' comments, we have newly derived the relationship between printing height H and filament diameter D , as shown in Equation (2-4). The ink flow rate Q is calculated using Equation (2) (i.e., $Q = v \times \pi d^2 / 4$), where d is the diameter of the nozzle tip's outlet and v is obtained from CFD simulation. Based on fluid continuity and volume conservation law, when filaments are printed in the general deposition state (solid triangle, see Fig. 2c and Supplementary Figs. 9-10), we develop Equation (1) (i.e., $D = (4Q/V\pi)^{0.5}$), where D is the filament diameter, Q is the ink flow rate from Equation (2) above (i.e., $Q = v \times \pi d^2 / 4$), and V is the printing velocity. Equation (1) provides the theoretical basis for our FDA-3DP strategy of customizing D by continuously varying V and H . In order to quantify the general deposition state of 3D printing, we investigated various deposition states at different V (from 0.5 mm/s to 14.0 mm/s) and H (from 0.2 mm to 1.6 mm) using viscous inks L-PCL and H-PCL at $Q = 3.748 \text{ mm}^3/\text{s}$ and $Q = 1.880 \text{ mm}^3/\text{s}$, respectively, as depicted in Supplementary Figs. 9-10. Inspired by the dimensionless treatment in ref. 38 by X Zhao and ref. 42 by E. J Markvicka, we introduced the nondimensional height $H^* = H/d$ and velocity $V^* = V/v$, and we re-derived equations to establish the relationship between D and H , as illustrated in Fig. 2c, Supplementary Fig. 10, and Equation (4) (i.e., $H = k \times D$). The detailed derivation was provided in the revised "Supplementary Note 2" (Supplementary Information). Overall, in this study, Equation (1) (i.e., $D = (4Q/V\pi)^{0.5}$) and Equation (4) (i.e., $H = k \times D$) guide our proposed FDA-3DP strategy to customize V and H for all moving points on the print trajectory, respectively. In response to your comments and to clearly show the reader the CFD simulation model developed in this work, we have added content to the "Results" section (Page 4, Lines 105-115) and the "CFD simulation of ink flow" of the "Methods" section (Pages 14-15, Lines 456-494) in the

revised manuscript and as shown below.

We appreciate your constructive comments. CFD simulation of the whole manufacturing process for FDA-3D printing is an exciting, feasible, and challenging project. In particular, as you mentioned, the process from the ink being extruded from the nozzle tip's outlet to contacting the substrate involves numerous boundary conditions (e.g. gravity pull, shear thinning, yield stress, liquid-solid phase transition). In order to reduce the workload in this study, the CFD model we built only simulated the ink flow inside the nozzle tip to get the average feeding velocity v . Based on the Equations (1-4), the CFD simulation model of the average feeding velocity v is the engine of the "design-to-fabrication" workflow for the FDA-3D printing of gradient matters.

In response to your concern about the “model”, “imaging” and “weighing” comparisons, we have revised the textual description for Fig. 2a-b (Page 22, Lines 666-673) and added new “Supplementary Note 1” (this response letter, page 25) in Supplementary Information in order to present it clearly to the reader in the revised manuscript.

New text revised in the “Results” section of the Main Text (Detailed description for CFD simulations):

Page 4, Lines 105-115: “..., we developed a computational fluid dynamics (CFD) model to simulate the flow of viscous ink within the nozzle and tip in DIW 3D printing (Supplementary Fig. 7). The CFD simulation calculates the average feeding velocity, defined as v (mm s^{-1}), of viscous ink at the nozzle tip's outlet. Subsequently, following the fluid flow law, the extrusion flow rate Q can be determined from the CFD simulation, as indicated in the equation below:

$$Q = v \times \pi d^2 / 4 \quad (2)$$

where Q is the extrusion flow rate ($\text{mm}^3 \text{s}^{-1}$), v is the average ink feeding velocity at the nozzle tip's outlet obtained from CFD simulation (mm s^{-1}), d is the inner diameter of the nozzle tip's outlet (mm), and π (Pi) is the ratio of a circle's circumference to its diameter. As shown in Fig. 2a,b, the CFD results for the flow rate Q coincide with the weighing and imaging results (Supplementary Fig. 8 and Supplementary Note 1).”

New legend revised in Fig. 2a,b of the Main Text:

Page 22, Lines 666-673: “a CFD simulation and weighing results of the extrusion flow rate Q (mean \pm s.d., $n = 8$). (i) shows the profile of the nozzle tip #1 (Regenovo Co., Ltd.). In (ii), the CFD result within the print head's needle tip is from the first row of subfigure (iii) (all four CFD results see Supplementary Fig. 8). In (iii), detailed process parameters are shown in Supplementary Table 3. b Plotting the relationship between D and V using CFD, weighing, and imaging methods (mean \pm s.d., $n = 4$). The same working parameters in all three methods (Supplementary Table 3). Both CFD simulation and weighing results adhere to $D = (4Q/V\pi)^{0.5}$ (i.e., Equation (1)), where $Q_{\text{simulation}}$ is $1.880 \text{ mm}^3 \text{ s}^{-1}$ (cyan) and Q_{weighing} is $1.240 \text{ mm}^3 \text{ s}^{-1}$ (black),

respectively.”

New text (CFD simulation of ink flow) added in the “Methods” section of the Main Text:

Same as responding to your Comment #1 (in Pages 20-21 of this Response letter, see New text (CFD simulation of ink flow) added in the “Methods” section of the Main Text:).

New Supplementary Note 1 added in the Supplementary Information:

“In this study, we developed a CFD model using COMSOL Multiphysics 6.0 to simulate the fluid flow of viscous ink inside the nozzle and tip. After determining processing parameters and ink rheology, we substituted the average feeding velocity (v) obtained from the CFD simulation model into Equation (2) (i.e., $Q = v \times \pi d^2 / 4$), resulting in the flow rate ($Q_{\text{simulation}}$), which we defined as the CFD simulation result, as depicted in Fig. 2a,b. In addition, we plotted the printing velocity V as a function of the filament diameter D using Equation (1) (i.e., $D = (4Q_{\text{simulation}}/V\pi)^{0.5}$), with $Q_{\text{simulation}} = 1.880 \text{ mm}^3/\text{s}$, as shown in Fig. 2b. Details of the CFD simulation are provided in the “CFD simulation of ink flow” of the Methods section.

For weighing results in Fig. 2b, the flow rate Q_{weighing} was measured by weighing ink extruded from the nozzle over a certain time ($Q = m/(\rho \times t)$), where m is the weight of the extrusion ink in t time, ρ is the ink density, and t is the extrusion time. For polymer inks L-PCL and H-PCL, the extrusion times are $t_{\text{L-PCL}} = 3$ minutes and $t_{\text{H-PCL}} = 8$ minutes, respectively. The densities of the inks L-PCL and H-PCL were $\rho_{\text{L-PCL}} = 1.3035 \text{ g cm}^{-3}$ and $\rho_{\text{H-PCL}} = 1.2168 \text{ g cm}^{-3}$, respectively. The flow rate Q_{weighing} obtained by this method is referred to as the weighing result, as shown in Fig. 2a,b. In addition, we plotted the printing velocity V as a function of the filament diameter D using Equation (1) (i.e., $D = (4Q_{\text{weighing}}/V\pi)^{0.5}$), where $Q_{\text{weighing}} = 1.240 \text{ mm}^3/\text{s}$, as shown in Fig. 2b. Further details can be found in the “Measurement of flow rate” of the Methods section.

For imaging results in Fig. 2b, constant-diameter D filament samples were 3D printed at different printing velocities V (0.5, 1, 2, 3, 4, 5, 6, 7, 8, 9, 10, 11, 12, 13, and 14 mm s^{-1}), followed by cutting with a knife. Subsequently, filament cross sections were photographed using a light microscope (HiROX MXB-5040RZ, Japan). The cross-sectional area S was determined through ImageJ software (Fiji Is Just ImageJ, <https://imagej.net/software/fiji/>). Filament diameters D corresponding to the various printing velocities V were computed using the formula $D = (4S/\pi)^{0.5}$. These results (data points) are denoted as imaging results in Fig. 2b. Further details are provided in the “Filament cross-section quantification” of the Methods section.”

New text revised in the “Supplementary Note 2” of the Supplementary Information:

“In the map of filament deposition states, the acceptable general state was marked as solid triangles (Fig. 2c and Supplementary Figs. 8-9). To establish the relationship between the

printing height H and the filament diameter D , we performed the following step-by-step derivation.

(1) The transformation of Equation (2) (i.e., $Q = v \times \pi d^2 / 4$) is performed to obtain the following equation:

$$v = 4Q / \pi d^2 \quad (\text{S1})$$

where Q is the extrusion flow rate (mm^3/s), v is the average feeding velocity through the nozzle tip (mm/s), d is the inner diameter of the nozzle tip's outlet (mm), and π (Pi) is the ratio of a circle's circumference to its diameter.

(2) $H^* = H/d$ and $V^* = V/v$ are substituted into Equation (3) (i.e., $H^* = k \times (V^*)^{-0.5}$) in order to remove the dimensionless parameters H^* and V^* . The new equation is shown below.

$$H/d = k \times (V/v)^{-0.5} \quad (\text{S2})$$

(3) Equation (S1) (i.e., $v = 4Q / \pi d^2$) is substituted into Equation (S2) (i.e., $H/d = k \times (V/v)^{-0.5}$) to remove the average ink feeding velocity (i.e., v) through the nozzle tip. The new equation is shown below.

$$H/d = k \times [V / (4Q / \pi d^2)]^{-0.5} \quad (\text{S3})$$

Equation (S3) above is adjusted and simplified to obtain the following equation.

$$H = k \times (4Q / V \pi)^{0.5} \quad (\text{S4})$$

(4) Equation (1) (i.e., $D = (4Q / V \pi)^{0.5}$) is switched left and right (i.e., $(4Q / V \pi)^{0.5} = D$), and then substituted into Equation (S4) above (i.e., $H = k \times (4Q / V \pi)^{0.5}$) to remove the constant, Q , V and π . The new equation about H and D is shown below.

$$H = k \times D \quad (\text{S5})$$

Equation (S5) above is also shown in the Results section of the main text as Equation (4).”

Q3: The properties of the ink and its rheology are neither discussed and nor measured. The manuscript should: 1) refer to the importance of the inks properties in terms of rheology (there is a lot of work being done in this area, and it is shocking there is no reference to it in this work), 2) how their methodology is adjusted depending on inks properties (again providing more context here), 3) the manuscript should provide at least some evidence of the ink properties that is being printed.

Response: Thank you for your comment, we apologize for the lack of fundamental rheological characteristics in the original manuscript. In the past, considering that we used common and commercial DIW printing ink materials, only the viscosities of the inks H-PCL and L-PCL with a logarithmic sweep of controlled shear rate (10^{-1} - 10^2 s^{-1}) were shown in the old Supplementary Fig. 5 in the original manuscript.

old Supplementary Fig. 5 | Rheology of H-PCL and L-PCL.

Following your suggestion, we have characterized the rheological properties of the inks L-PCL and H-PCL used in this work in detail, as shown in revised Supplementary Fig. 6 (corresponding to the old Supplementary Fig. 5). Specifically, we showed the relationships of the dynamic viscosity η vs. the shear rate γ (Supplementary Fig. 6a), the shear stress τ vs. the shear strain ϵ (Supplementary Fig. 6b), and the storage modulus G' and loss modulus G'' vs. working temperature (Supplementary Fig. 6c). Details of the rheological experiments are presented in *Rheological characterizations* (Page 10, Lines 291-303) of the “Methods” section in the revised manuscript.

In response to your comment about the importance of ink rheological properties, in our work, to precisely control the volume of the extruded ink on the print trajectory, our focus was to maintain the uniform straightness of the deposited filament (acceptable general deposition state, marked as solid triangles, see Fig. 2c and Supplementary Figs. 9-10), avoiding its coiling effect and discontinuity. Inspired by the dimensionless approach (ref. 38 by X Zhao and ref. 42 by E. J Markvicka), the relationship between printing height H and printed filament diameter D was newly derived by solving simultaneous Equations (1-3), resulting in Equation (4) (i.e., $H = k \times D$), where k ranges about from 0.6 to 1.0 (unitless coefficient). The detailed derivation is shown in the revised “Supplementary Note 2” of the Supplementary Information. The obtained Equation (4) (i.e., $H = k \times D$) fits all viscous inks that can be DIW 3D printed. Notably, as you commented, it does not mean that the rheological properties of viscous inks are not useful in DIW printing (lots of publications have demonstrated this phenomenon).

Equation (1) (i.e., $D = (4Q/V\pi)^{0.5}$), obtained from fluid continuity and volume conservation law, is the key to the FDA-3DP strategy for fabricating gradient structures. In this strategy, the gradient in pore sizes is manufactured by continuously varying the printing velocities V and heights H . To determine the flow rate Q of viscous inks at the nozzle tip's outlet, based on volume conservation law, we newly establish Equation (2) (i.e., $Q = v \times \pi d^2 / 4$), where

d is the inner diameter of the nozzle tip's outlet, and v is the average feeding velocity at the nozzle tip's outlet. After determining the process parameters (nozzle tip shape, nozzle tip inner diameter, extrusion air pressure, working temperature), the rheological properties of viscous inks were used to calculate the average feeding velocity v through the CFD simulation that simulates the ink flow inside the nozzle and tip.

In general, in response to your comments, the revised Results (Page 4, Lines 105-115), Supplementary Fig. 6 (Supplementary Information), and experimental descriptions (see *Rheological characterizations* in Methods section, Page 10, Lines 291-303) have been updated in the revised manuscript and also shown below.

New text revised in the “Results” section of the Main Text (Detailed description for CFD simulations):

Page 4, Lines 105-115: “..., we developed a computational fluid dynamics (CFD) model to simulate the flow of viscous ink within the nozzle and tip in DIW 3D printing (Supplementary Fig. 7). The CFD simulation calculates the average feeding velocity, defined as v (mm s^{-1}), of viscous ink at the nozzle tip's outlet. Subsequently, following the fluid flow law, the extrusion flow rate Q can be determined from the CFD simulation, as indicated in the equation below:

$$Q = v \times \pi d^2 / 4 \quad (2)$$

where Q is the extrusion flow rate ($\text{mm}^3 \text{s}^{-1}$), v is the average ink feeding velocity at the nozzle tip's outlet obtained from CFD simulation (mm s^{-1}), d is the inner diameter of the nozzle tip's outlet (mm), and π (Pi) is the ratio of a circle's circumference to its diameter. As shown in Fig. 2a,b, the CFD results for the flow rate Q coincide with the weighing and imaging results (Supplementary Fig. 8 and Supplementary Note 1).”

New text (*Rheological characterizations*) added in the “Methods” section:

Page 10, Lines 291-303: “The rheological properties of the inks L-PCL and H-PCL were assessed using a modular compact rheometer equipped with a 25 mm-diameter steel plate (Anton Paar, MCR302). Shear viscosity was characterized through rotation tests, employing a logarithmic sweep of controlled shear rate (10^{-3} - 10^3 s^{-1}). Each ink (L-PCL and H-PCL) underwent three repetitions of the shear rate vs. viscosity test. Yield stress behavior was examined through rotation tests with a logarithmic sweep of controlled shear stress (10^{-1} - 10^3 Pa). In these rotation tests (shear viscosity and yield stress), the working temperatures for the materials L-PCL and H-PCL were 70 °C and 110 °C, respectively. A linear temperature sweep in oscillatory mode was employed to assess the stored modulus G' and loss modulus G'' of inks at a shear strain of 1% and a frequency of 1 Hz. The inks L-PCL and H-PCL were swept over the temperature ranges of from 70 °C to 25 °C and from 110 °C to 25 °C, respectively. During the temperature sweep, the temperature dropped at a rate of 1 °C per minute.”

Detailed derivation in the “Supplementary Note 2” of the Supplementary Information:

Same as responding to your Comment #2 (in Pages 25-26 of this Response letter, see New text revised in the “Supplementary Note 2” of the Supplementary Information:).

New Supplementary Fig. 6 added in the Supplementary Information:

Supplementary Fig. 6 | Rheological characteristics of the polymer inks (L-PCL and H-PCL) used in this study. a, Dynamic viscosity of L-PCL and H-PCL at 70 °C and 110 °C, respectively. Both L-PCL and H-PCL exhibit non-Newtonian behavior, demonstrating the shear-thinning phenomenon at a high shear rate (about $10^2 s^{-1}$). Three repetitions were conducted. The shear rate is from $10^{-3} s^{-1}$ to $10^3 s^{-1}$. The zero-shear viscosities η_0 of L-PCL and H-PCL are approximately 382.299 Pa·s and 36.323 Pa·s, respectively. **b,** Relationship between shear stress and shear strain using rotational tests with controlled shear stress (CSS). Both L-PCL and H-PCL exhibit the yield stress. **c,** Storage G' and loss G'' modulus of the H-PCL ink for temperature

scanning (from 110 °C to 25 °C) through oscillatory measurement. **d**, Storage G' and loss G'' modulus of the L-PCL ink for temperature scanning (from 70 °C to 25 °C) through oscillatory measurement. During material deformation, elasticity and viscosity are measured by the storage modulus G' and loss modulus G'', respectively.

The materials and methods section contains some information on the properties of the ink:

L-PCL (dynamic viscosity, 39 Pa s; density, 1303.5 kg/m³) and H-PCL (dynamic viscosity, Pa s; density, 1216.8 kg/m³) were used as printing inks, and the inks were considered incompressible, continuous, and isotropic in the CFD simulation. The flow rate Q obtained by this method was called the CFD result, shown in Fig. 2a-b.

Q4: What does isotropic means here? Were the inks considered as Newtonian fluids with constant viscosity? Because this is obviously not the case, the materials used here do show a yield stress and shear thinning behaviour. Figure 5 in the supplementary information shows two viscosity curves for the inks, how were they measured? And how reliable are they? They are too good to be true! Viscosity is not enough to characterise the complex behaviour of DIW formulations. And it is very strange that formulations that (from the images provided) have a yield stress, and elastic components, show such a plateau in viscosity at those shear rate values. The authors should improve this area of the work because it is directly related to the flow properties of the material and therefore to the use of Q manipulation to create gradient structures.

Response: Thank you for your insights. We sincerely thank you for kindly copying relevant information about these comments from the original manuscript, as shown in the gray text above. Based on your comments, we have revised the resubmitted manuscript and responded as follows.

Fluids are isotropic, which is the assumed condition for describing fluid flow in mathematical terms. For Newtonian fluids, the stress tensor and the deformation rate tensor are related as linear chi-square functions, and their general linear relationship is as follows:

$$P_{ij} = -p\delta_{ij} + 2\mu(s_{ij} - \frac{1}{3}s_{kk}\delta_{ij}) + \mu' s_{kk}\delta_{ij} \quad (R2)$$

where P_{ij} is the stress tensor, p is the isotropic pressure, s_{ij} is the deformation rate tensor, s_{kk} is the isotropic volume deformation rate tensor, δ_{ij} is the Kronecker symbol, and μ' is the expansion viscosity coefficient. If the nature of the fluid does not match the above equation, it is a non-Newtonian fluid. In this study, the viscosity of the polymer materials (H-PCL and L-PCL) showed a gradual decrease in viscosity with increasing shear rate, indicating that both H-PCL and L-PCL are non-Newtonian fluids, as shown in Supplementary Fig. 6a (corresponding to the old Supplementary Fig. 5 in the original manuscript).

The rheological properties of inks L-PCL and H-PCL were measured using a modular compact rheometer with a 25 mm-diameter steel plate (Anton Paar, MCR302). Following your suggestion, we have extended the shear rate test range to 10^{-3} - 10^3 s^{-1} from 10^{-1} - 10^2 s^{-1} . Shear viscosity was characterized using rotation tests with a logarithmic sweep of controlled shear rate (10^{-3} - 10^3 s^{-1}). Moreover, to increase the objectivity and fairness of the rheological data on viscosity, the shear rate vs. the viscosity test was repeated three times for each ink (H-PCL and L-PCL), as shown in Supplementary Fig. 6a. Besides, yield stress behavior was evaluated by rotation tests with a logarithmic sweep of controlled shear stress (10^{-1} - 10^3 Pa) (Supplementary Fig. 6b). In rotation tests (shear rate vs. shear viscosity and shear strain vs. shear stress), the working temperatures of the L-PCL and H-PCL materials were 70 °C and 110 °C, respectively. The linear temperature sweep in oscillatory mode was selected to measure the stored modulus G' and loss modulus G'' of inks at a shear strain of 1% and a frequency of 1 Hz. The L-PCL and H-PCL inks were swept over the temperature ranges of from 110 °C to 25 °C and from 70 °C to 25 °C, respectively (Supplementary Fig. 6c-d). In the temperature sweep, the drop in temperature is 1 °C per minute.

The rheological data (shear rate vs. shear viscosity, as shown in Supplementary Fig. 6a) of the polymer materials (H-PCL and L-PCL) exhibit a constant shear viscosity at small shear rates and a low (near zero) shear viscosity at very high shear rates, which satisfies the characteristics of the Carreau Yasuda model (non-Newtonian fluid). The intrinsic equation of the Carreau Yasuda model is shown below:

$$\eta = \eta_{\infty} + (\eta_0 - \eta_{\infty}) \times [1 + (\gamma\lambda)^a]^{(n-1)/a}$$

where η , γ , λ , n , and a are the shear viscosity, shear rate, relaxation time, flow behavior index, and Yasuda index, respectively. Besides, η_0 and η_{∞} are the zero shear viscosity and infinite shear viscosity, respectively. The rheological data (shear rate vs. shear viscosity) were used to fit the Carreau Yasuda model, yielding the R-Square values (R^2 , also known as the coefficient of determination) of 0.943 and 0.807 for H-PCL and L-PCL, respectively⁵³. For the Carreau Yasuda model, the various factors were obtained for the H-PCL ($\lambda=0.051$, $n=0.802$, $a=217.786$, $\eta_0=382.299$ Pa s, and $\eta_{\infty}=0$ Pa s) and L-PCL ($\lambda=0.014$, $n=0.840$, $a=217.786$, $\eta_0=36.323$ Pa s, and $\eta_{\infty}=0$ Pa s) materials, as demonstrated in Supplementary Fig. 7.

Overall, following your comments, we have revised the “Results” section (Page 4, Lines 105-115), *Rheological characterizations* (Page 10, Lines 291-303) and *CFD simulation of ink flow* (Pages 14-15, Lines 457-473) in the “Methods” section, and Supplementary Figs. 6-7 (Supplementary Information) in the new manuscript, as also shown below.

New text (*Rheological characterizations*) added in the “Methods” section:

Page 10, Lines 291-303: “The rheological properties of ink materials L-PCL and H-PCL were measured using a modular compact rheometer with a 25 mm-diameter steel plate (Anton Paar,

MCR302). Shear viscosity was characterized through rotation tests with a logarithmic sweep of controlled shear rate (10^{-3} - 10^3 s⁻¹). The shear rate vs. viscosity test was repeated three times for each ink (H-PCL and L-PCL) separately. Yield stress behavior was evaluated by rotation tests with a logarithmic sweep of controlled shear stress (10^{-1} - 10^3 Pa). In rotation tests (shear viscosity and yield stress), the working temperatures of the L-PCL and H-PCL materials were 70 °C and 110 °C, respectively. The linear temperature sweep in oscillatory mode was selected to measure the stored modulus G' and loss modulus G'' of inks at a shear strain of 1% and a frequency of 1 Hz. The L-PCL and H-PCL inks were swept over the temperature ranges of from 110 °C to 25 °C and from 70 °C to 25 °C, respectively. In the temperature sweep, the drop in temperature is 1 °C per minute.”

New text (CFD simulation of ink flow) added in the “Methods” section of the Main Text:

Pages 14-15, Lines 457-473: “The commercially available software COMSOL Multiphysics 6.0 (Dassault Systèmes, USA) was employed to develop a CFD simulation model for calculating the average ink feeding velocity v at the nozzle tip's outlet under various printing parameters. The rheological data, as depicted in Supplementary Fig. 6a, illustrate that the polymer materials (H-PCL and L-PCL) display constant shear viscosity at small shear rates and low (near zero) shear viscosity at very high shear rates, consistent with the characteristics of the Carreau Yasuda model for non-Newtonian fluids. The intrinsic equation of the Carreau Yasuda model is presented below:

$$\eta = \eta_{\infty} + (\eta_0 - \eta_{\infty}) \times [1 + (\gamma\lambda)^a]^{(n-1)/a}$$

where η , γ , λ , n , and a are the shear viscosity, shear rate, relaxation time, flow behavior index, and Yasuda index, respectively. η_0 and η_{∞} are the zero shear viscosity and infinite shear viscosity, respectively. The rheological data (shear viscosity vs. shear rate) were used to fit the Carreau Yasuda model, resulting in R-Square values (R^2 , also known as the coefficient of determination) of 0.943 and 0.807 for H-PCL and L-PCL, respectively⁵³. The Carreau Yasuda model parameters for inks H-PCL ($\lambda=0.051$, $n=0.802$, $a=217.786$, $\eta_0=382.299$ Pa s, and $\eta_{\infty}=0$ Pa s) and L-PCL ($\lambda=0.014$, $n=0.840$, $a=217.786$, $\eta_0=36.323$ Pa s, and $\eta_{\infty}=0$ Pa s) are illustrated in Supplementary Fig. 7.”

New Supplementary Fig. 6-7 added in the Supplementary Information:

Supplementary Fig. 6 | Rheological characteristics of the polymer inks (L-PCL and H-PCL) used in this study. **a**, Dynamic viscosity of L-PCL and H-PCL at 70 °C and 110 °C, respectively. Both L-PCL and H-PCL exhibit non-Newtonian behavior, demonstrating the shear-thinning phenomenon at a high shear rate (about $10^2 s^{-1}$). Three repetitions were conducted. The shear rate is from $10^{-3} s^{-1}$ to $10^3 s^{-1}$. The zero-shear viscosities η_0 of L-PCL and H-PCL are approximately 382.299 Pa·s and 36.323 Pa·s, respectively. **b**, Relationship between shear stress and shear strain using rotational tests with controlled shear stress (CSS). Both L-PCL and H-PCL exhibit the yield stress. **c**, Storage G' and loss G'' modulus of the H-PCL ink for temperature scanning (from 110 °C to 25 °C) through oscillatory measurement. **d**, Storage G' and loss G'' modulus of the L-PCL ink for temperature scanning (from 70 °C to 25 °C) through oscillatory measurement. During material deformation, elasticity and viscosity are measured by the storage modulus G' and loss modulus G'' ,

respectively.

Supplementary Fig. 7 | Carreau Yasuda model fitting for H-PCL and L-PCL materials. For H-PCL ink, the relaxation time λ is 0.051, the flow behavior index n is 0.802, the Yasuda index a is 217.786, the zero shear viscosity η_0 is 382.299 Pa s, and the infinite shear viscosity η_∞ is 0 Pa s. For L-PCL ink, the relaxation time λ is 0.014, the flow behavior index n is 0.840, the Yasuda index a is 217.786, the zero shear viscosity η_0 is 36.323 Pa s, and the infinite shear viscosity η_∞ is 0 Pa s.

Q5: Results in figure 2a are not enough to support and discuss the CFD work done (what’s the purpose of the CFD?). It must be clarified what the “model” within the pipe (syringe/nozzle) is and it has to be clarified that the “model” is not investigating the flow and state of the material once it comes out of the nozzle.

Response: We appreciate your comments. Indeed, in Fig. 2a, the CFD simulation model was used to simulate the laminar flow of viscous ink within the nozzle and tip. This CFD work was used to calculate the average feeding velocity v at the nozzle tip's outlet, and then the flow rate Q of ink extrusion was calculated using the newly added Equation (2) (i.e., $Q = v \times \pi d^2 / 4$). Considering that Equation (1) (i.e., $D = (4Q/V\pi)^{0.5}$) is an important foundation for our proposed FDA-3DP strategy, we developed the above CFD simulation model to provide the flow rate Q for different processing parameters (nozzle tip shape, nozzle tip inner diameter, extrusion air pressure, and working temperature) and inks with different rheological properties. Based on your comments, we have clarified that the CFD model in this work was used to simulate the flow of viscous ink inside the nozzle and tip (Page 4, Lines 103-115).

Additionally, following your comments, we have newly done the following works: 1. Characterized the rheological properties of inks H-PCL and L-PCL (see Supplementary Fig. 6), 2. Fitted their intrinsic equations using the Carreau Yasuda model (non-Newtonian fluids) (see Supplementary Fig. 7), and 3. Calculated the extrusion flow rate Q (see Fig. 2a-b) using the

newly revised CFD simulation model. The response Figure R2 shows that this CFD model could be used to predict the average feeding velocity v and then calculate the flow rate Q through Equation (2) (i.e., $Q = v \times \pi d^2 / 4$). In conclusion, following your comments, we have modified Fig. 2a-b, the “Results” section (Page 4, Lines 103-115), and the “CFD simulation of ink flow” of the “Methods” section (Pages 14-15, Lines 457-495) of the revised manuscript.

New Figure (Fig. 2a-b) revised in the Main section:

response Figure R2 | The top and bottom pictures are old and new Fig. 2a-b, respectively.

New text revised in the “Results” section of the Main Text:

Page 4, Lines 103-115: “To determine Q at different processing parameters (nozzle tip shape, nozzle tip's inner diameter, extrusion air pressure, and working temperature) and inks with different rheology, we developed a computational fluid dynamics (CFD) model to simulate the flow of viscous ink within the nozzle and tip in DIW 3D printing (Supplementary Fig. 7). The CFD simulation calculates the average feeding velocity, defined as v (mm s^{-1}), of viscous ink at the nozzle tip's outlet. Subsequently, following the fluid flow law, the extrusion flow rate Q can be determined from the CFD simulation, as indicated in the equation below:

$$Q = v \times \pi d^2 / 4 \quad (2)$$

where Q is the extrusion flow rate ($\text{mm}^3 \text{ s}^{-1}$), v is the average ink feeding velocity at the nozzle tip's outlet obtained from CFD simulation (mm s^{-1}), d is the inner diameter of the nozzle tip's outlet (mm), and π (Pi) is the ratio of a circle's circumference to its diameter. As shown in Fig. 2a,b, the CFD results for the flow rate Q coincide with the weighing and imaging results (Supplementary Fig. 8 and Supplementary Note 1).”

New text (CFD simulation of ink flow) added in the “Methods” section of the Main Text:

Same as responding to your Comment #1 (in pages 20-21 of this Response letter, see New text (CFD simulation of ink flow) added in the “Methods” section of the Main Text:).

Q6: Figure 6 in supplementary information provides some velocity profiles for different conditions. More details needs to be provided on how the velocity profiles within the nozzle are determined.

Response: Thank you for your comment. In response to your comment about the velocity profiles within the nozzle and to present them clearly to the reader, we have newly revised Supplementary Fig. 8 and added content to the “CFD simulation of ink flow” of the “Methods” section (Page 15, Lines 474-488) in the revised manuscript. Please note that the new Supplementary Fig. 8 in the revised manuscript corresponds to the old Supplementary Fig. 6 in the original manuscript.

Specifically, in CFD simulation (Page 15, Lines 474-488), we have done the following using the software COMSOL Multiphysics 6.0 (Dassault Systèmes, USA) to obtain the simulation results: (1) the flow velocity distribution inside the nozzle and tip was obtained using the 2D plot command (Supplementary Fig. 8a-d); (2) the flow velocity distribution at the nozzle tip's outlet was measured using the linear resultant plot command (Supplementary Fig. 8e); and (3) the average ink feeding velocity v at the outlet of the nozzle tip was calculated using the boundary probe and then was used to calculate the extrusion flow rate Q through $Q = v \times \pi d^2 / 4$ (Fig. 2a-b).

New text (CFD simulation of ink flow) added in the “Methods” section of the Main Text:

Page 15, Lines 474-488: “Using the Geometric Drawing tool, we constructed the computational domain of the nozzle tip based on their actual dimensions. The top and bottom of the computational domain were given as the inlet and outlet, respectively. The inlet air pressure was set according to the experimental requirements (400 or 500 kPa). The remaining walls were set to be smooth and slip-free. The reference temperature was set to room temperature (25 °C). Assuming that the ink flows inside the nozzle tip without experiencing heat changes. The

viscous inks (L-PCL and H-PCL) were considered incompressible, continuous, and isotropic. The fluid flow was set to be laminar. The computational domain was meshed using regular cell sizes. In COMSOL, the flow velocity distribution inside the nozzle and tip was obtained using the 2D plot command (Supplementary Fig. 8a-d), and the flow velocity distribution at the nozzle tip's outlet was measured using the linear resultant plot command (Supplementary Fig. 8e). The average ink feeding velocity v at the outlet of the nozzle tip was calculated in COMSOL using the boundary probe (Fig. 2a,b). The flow rate Q can be determined by substituting the obtained average ink feeding velocity v into Equation (2) (i.e., $Q = v \times \pi d^2 / 4$). The flow rate Q obtained in this way was called the CFD result, shown in Fig. 2a-b.”

New Supplementary Fig. 8 added in the Supplementary Information:

Supplementary Fig. 8 | Flow velocity distribution in CFD simulation. a, Flow velocity distribution in 1st row of Fig. 2(iii). Processing parameters include H-PCL ink, extrusion air pressure of 500 kPa, and nozzle tip inner diameter of 400 μ m. **b,** Flow velocity distribution in 2nd row of Fig. 2(iii). Processing parameters include H-PCL ink, extrusion air pressure of 500 kPa, and nozzle tip inner diameter of 500 μ m. **c,** Flow velocity distribution in 3rd row of Fig. 2(iii). Processing parameters include H-PCL ink, extrusion air pressure of 400 kPa, and nozzle tip inner diameter of 400 μ m. **d,** Flow velocity distribution in 4th row of Fig. 2(iii). Processing parameters include L-PCL ink, extrusion air pressure of 400 kPa, and nozzle tip inner diameter of 400 μ m.

2(iii). Processing parameters include H-PCL ink, extrusion air pressure of 400 kPa, and nozzle tip inner diameter of 400 μm . **d**, Flow velocity distribution in 4th row of Fig. 2(iii). Processing parameters include L-PCL ink, extrusion air pressure of 500 kPa, and nozzle tip inner diameter of 400 μm . **e**, Quantification of flow velocity at the nozzle tip's outlet.

Responses to Reviewer #3:

The organization of the paper is confusing, with the "Materials and methods section" at the end of the document.

Response: Thank you for your comment. Based on your and the Editor's comments, we have carefully standardized the resubmitted manuscript according to the formatting instructions of *Nature Communications*. Besides, we have moved the "Methods" section (see Page 9, Line 257) after the "Discussion" section and before the "Data availability" section. Following your comments, we have thoroughly revised our manuscript and improved the quality of our work. Therefore, we sincerely ask you to re-evaluate the possibility of our work to be published in *Nature Communications*.

It also has other concerns:

Q1: - What is the main objective of the paper?

Response: We thank the reviewer for pointing out this concern. The main objective of this manuscript is summarized below. Currently, although direct ink writing (DIW) is one of the most widely used 3D printing (also known as additive manufacturing) technologies, the constant working parameters (e.g., printing velocity V and height H) during the fabrication process limit the manufacturing of complex 3D gradient porous matters. Therefore, in this paper, we developed a filament diameter-adjustable 3D printing (FDA-3DP) strategy that enables conventional DIW 3D printers to produce 1D, 2D, and 3D gradient matters with tunable heterogeneous pores through continuously variable process parameters V and H . Our application examples demonstrated the potential of this strategy in several fields, including tissue engineering (bone, meniscus, and blood vessel), flexible electronics, and 4D printing. Detailed text can be found in the "Abstract" section of the Main Text (see Pages 1-2, Lines 17-35).

In conclusion, in this study, we introduce a strategy for DIW-based 3D printing of complex porous products. Our work may provide inspiration for readers and peers to fabricate 3D heterogeneous porous structures through continuously variable process parameters V and H . Thanks again for your concern.

Q2: - What is the novelty of the paper with respect to previous works?

Response: Thank you for your question. For DIW-based 3D printing of complex porous products, previous works (see Supplementary Table 1 in Supplementary Information) focused on overall variation in the filament diameter (*Acta Biomater.* **90**, 37-48, 2019), spacing (*Biofabrication* **8**, 045007, 2016), and intersection angle (*Addit. Manuf.* **38**, 101760, 2021)

between different regions of interest. Although these strategies were capable of generating the axial gradients in pore sizes, they faced challenges in DIW printing complex gradients with high shape fidelity, such as radial and 3D heterogeneous structures. In this manuscript, our proposed FDA-3DP strategy addresses this challenging issue. Using the FDA-3DP strategy, we showcased the multi-disciplinary applications of this strategy in creating horizontal and radial gradient structures, “HIT” letter-embedded structures, metastructures, tissue-mimicking scaffolds (bone, meniscus, and blood vessel), a flexible electronic, and a time-driven device (4D printing). To our knowledge, existing extrusion-based 3D printing technologies face significant challenges for the applications we have demonstrated above.

Q3: - Line 89. The dimensional results of the computerized tomography should be added. What is the value of dimensional error?

Response: Thank you for your comment. In old Line 89 (corresponding to Fig. 1d), we apologize for not showing the dimensional results of the computed tomography (CT) and their errors in the original manuscript. Following your comments, we have added the presentation of the dimensional results (10 mm×10 mm×10 mm) of the CAD model with a horizontal gradient in pore sizes, as shown in the newly added Supplementary Fig. 5a. Then, we have newly printed three horizontal gradient porous samples corresponding to old Line 89, and processed them using CT scanning with a resolution of 20 μm (SCANCO, Switzerland). 3D models of the printed samples were reconstructed using 3D visualization software Bruker's SkyScan, as depicted in the newly added Supplementary Fig. 5b. Finally, as shown in Supplementary Fig. 5 and Supplementary Table 2, the dimensional results of the CT-based 3D models and their errors were evaluated along the *x-y-z* directions (10.30±0.40 mm, 10.40±0.26 mm, and 9.83±0.15 mm), respectively.

Please note that the line numbers have been updated due to the formatting changes based on your and the Editor's comments. The old “Line 89” corresponds to the new “Page 3, Lines 89-94” in the revised manuscript. The relevant figure (Supplementary Fig. 5), table (Supplementary Table 2), and text descriptions (Page 3, Lines 89-94) have been updated as below and in the revised manuscript.

New Supplementary Fig. 5 added in the Supplementary Information:

Supplementary Fig. 5 | Dimensional results of the horizontal gradient CAD model and corresponding printed samples. **a**, Cube boundary (10 mm×10 mm×10 mm) of the CAD model presented in Fig. 1d(i-ii). **b**, Dimensional measurements of the printed samples depicted in Fig. 1d(iii) along the x , y , and z directions are 10.30±0.40 mm, 10.40±0.26 mm, and 9.83±0.15 mm, respectively.

New text added in the “Results” section of the Main Text:

Page 3, Lines 89-94: “The micro-computed tomography (μ -CT) reconstruction verifies the congruence of the printed sample's dimensions (10.30±0.40 mm, 10.40±0.26 mm, and 9.83±0.15 mm in the x - y - z directions, respectively) with those of the computer-aided design (CAD) cube model (10 mm×10 mm×10 mm), as shown in Fig. 1d(ii-iii), Supplementary Fig. 5, and Supplementary Table 2. Additionally, it clearly illustrates the anticipated horizontal gradient in pore sizes from 0 to 1128 μ m (Fig. 1d(iv) and Supplementary Video 1).”

New Supplementary Table added in the Supplementary Information:

Supplementary Table 2 | Dimensional values for the horizontal gradient model and samples.

Three dimensional	CAD model	μ -CT-based 3D model (mm)				
		Sample 1	Sample 2	Sample 3	Average	Error
x -axis	10 mm	10.7	10.3	9.9	10.30	0.40
y -axis	10 mm	10.5	10.1	10.6	10.40	0.26
z -axis	10 mm	9.7	9.8	10.0	9.83	0.15

Q4: - Please provide the values of the rest of the printing parameters: extrusion multiplier, infill ratio, etc.

Response: Thanks for your comment. Our responses are given in the following paragraphs.

- (1) The extrusion multiplier, usually expressed as a percentage, is the flow rate of viscous ink through the 3D printer's nozzle. The default setting for the extrusion multiplier is usually 100 %, which indicates that the normal extrusion flow Q is fully maintained. In our work, all samples were printed at 100 % extrusion multiplier (i.e., full extrusion flow). Thank you again for your suggestion. We have added this printing parameter in the “Results” section (Page 4, Line 98) of the revised manuscript.
- (2) The infill ratio also commonly takes values between 0 % and 100 %, with 0 % and 100 % indicating that the part is hollow and completely solid, respectively. The infill ratio is the percentage of overall space occupied by the material inside the object, and its larger value indicates less porosity inside the object. In this work, to fabricate gradient pore structures, we developed a varying infill density method (i.e., FDA-3D printing strategy) based on the law of conservation of volume. We have added content that clearly explains this parameter in the revised manuscript (Page 3, Lines 67-68).
- (3) The working temperatures of inks L-PCL and H-PCL were 70 °C and 110 °C, respectively. Following your comment, we have also added working temperature in the “Methods” section (Page 10, Lines 298-299) of the revised manuscript.

Q5: - Line 266. The explanation about the transfer of the data from CAD model to the 3D printing machines should be explained in more detail.

Response: Thank you for your suggestion. We apologize for not explaining the design-to-manufacturing workflow clearly in the original manuscript. As a proof of concept, Fig. 3 and Supplementary Fig. 14 illustrates the design and fabrication of the horizontal gradient pore structure in a cube (10 mm×10 mm×10 mm). The G-codes files are the key link in this design-to-fabrication workflow. Details are provided in the “Results” section (Page 5, Lines 156-161; Page 6, Lines 181-187) and the “Methods” section (*Design-to-fabrication workflow*, Pages 10-11, Lines 313-342) of the revised manuscript.

Details of data from CAD model to 3D printing machines in the “Results” section

Page 6, Lines 156-161: “Before producing G-codes for FDA-3D printing, we performed the design of gradient pore structures. Considering the fluid flow of extruded viscous inks, the filament cross section is not ideally circular but oblong^{43,44}. Accordingly, we established the relationship between filament cross-section and V (Supplementary Note 4), and then created 3D gradient models with FDA and supporting layers using the CAD software Grasshopper based on the FDA-3DP strategy (*Design-to-fabrication workflow* in Methods).”

Page 6, Lines 181-187: “ V and H of the FDA and supporting layers that matched the collapse-free model (see phase diagram of printing parameters in Fig. 3i) were written into a customized fabrication G-codes file (Fig. 3j). Then, we printed the horizontal gradient sample by running the G-codes file via our commercial extrusion 3D printer (Fig. 3k,l, Fig. 1d(iii), Supplementary Fig. 14, and Supplementary Video 4). The serial numbers of the keyframes of the FDA-3D printing process in Fig. 3l correspond to those in the phase diagram of the printing parameters in Fig. 3i.”

Fig. 3 Design and fabrication of the horizontal gradient structure via our proposed FDA-3DP strategy. **a** Conventional grid pattern for 3D printing trajectory. **b** Design gradient of the pore structure. The pore size is the value of filament spacing minus filament width l_w (Supplementary Fig. 3). **c** Data

combination of conventional pattern and design gradient based on Equation (1) and Supplementary Note 4. **d** FDA single-layer model with the horizontal gradient in pore sizes and the customized V and H (i-ii) and D (iii). **e** FDA multi-layer model obtained through layer-by-layer stacking of FDA single-layer models. **f** A collapse region existing between the target cube and the designed FDA multi-layer model (Supplementary Fig. 4). **g-i** Acceptable FDA-3DP model without collapse (**i**) created by complementary stacking of FDA layers (**g**) and supporting layers (**h**). Customized printing parameters corresponding to the designed model are shown in the phase diagram at the bottom of (**i**). In the phase diagram, gray area, FDA layer; green area, supporting layer; blue line, V ; and red line, H . **j** G-codes file obtained from the customized printing parameters in (**i**) following motion rules of the Regenovo 3D printer. **k** FDA-3D printing of the designed gradient model by executing the customized G-codes file. **l** Keyframes of FDA layers and supporting layers in the FDA-3D printing process. The serial numbers in (**l**) correspond to those in the phase diagram of printing parameters in (**i**).

Supplementary Fig. 14 | Design-to-fabrication workflow between the computer and printer used in the FDA-3DP strategy. On the left and right are the operations performed by the computer and printer, respectively. The computer part includes 3D printer's control software (3D Bio-Architect, Regenovo Biotechnology Co., Ltd., China), simulation software (COMSOL Multiphysics 6.0, Dassault Systèmes, USA), and design software (Rhinoceros 3D embedded with a parametric design tool Grasshopper, Robert McNeel & Associates, USA). The extrusion-based 3D printer (Regenovo Bio-Architect WS) is from Regenovo Biotechnology Co., Ltd., China. The blue and black arrows are the forward trajectories of the design and manufacturing parts of this workflow, respectively.

Q6: - Line 307. When cutting the samples with a knife, do they suffer deformation?

Response: Thank you for your comment. In this experiment using a knife, we used a mixed ink configured from the polymer material polycaprolactone (PCL, average Mw 45,000) and the bioactive ceramic β -TCP (Sigma-Aldrich) in a weight ratio of 4:1. During the experiment, we found with the naked eye that cutting filament samples with a knife hardly caused any deformation of their cross-sectional shapes.

To quantitatively evaluate the impact of knife cutting on filament cross-section deformation, we designed a new experiment. In this experiment, the printed samples were first non-destructively scanned using a μ -CT equipment (SCANCO, Switzerland), and then processed by software Skyscan to obtain a series of filament cross-section data. Further, the μ -CT-scanned samples were cut with a knife, and evaluated for filament cross-section using an optical microscope (HiROX MXB-5040RZ, Japan). Following the above steps, we performed the filament cross-section evaluation at different printing velocities ($V = 2$ mm/s and $V = 5$ mm/s). As shown in Supplementary Fig. 15, the results of the filament cross sections from the μ -CT scanning and the cutting with a knife indicate that there is no significant difference.

We apologize for not explaining in detail the possible deformations caused by cutting filaments with a knife in the “*Filament cross-section quantification*” of the “Methods” section in the original manuscript (Page 12, Lines 384-386). Based on your suggestion, we have added Supplementary Fig. 15 and text descriptions below.

New Supplementary Fig. 15 added in the Supplementary Information:

Supplementary Fig. 15 | The agreement of the filament cross sections obtained through knife cutting and μ -

CT scanning. **a**, Filament cross section contours were obtained by cutting with a knife and scanning with a μ -CT device (SCANCO, Switzerland). Scale bar, 1 mm. **b**, Quantification of the filament cross sections through cutting and scanning, including area S , width l_w , and height l_h . Three μ -CT scan samples (lines with error bands) and six knife-cut samples (symbolic elements) were prepared and evaluated, respectively.

New text added in the “Methods” section of the Main Text:

Page 12, Lines 384-386: “The cross section results of the μ -CT scans demonstrated a high level of agreement with those obtained by cutting with a knife, showing that the knife treatment did not induce deformation in the H-PCL samples (Supplementary Fig. 15).”

Q7: - Line 380. A section with the conclusions is expected here.

Response: Thank you for your suggestion. Following your comments, we have moved the “Conclusions” section into the “Discussion” section (see Page 8, Lines 228-255). We are also required to do this in the formatting instructions of *Nature Communications* provided by the editor (see <https://www.nature.com/documents/ncomms-formatting-instructions.pdf>). For the article sections, *Nature Communications* declares that manuscripts resubmitted for publication must adhere to these formatting requirements as follows: Introduction, Results, Discussion (optional), Methods (optional), Data Availability, Code Availability (if applicable), References, Acknowledgements (optional), Author Contributions, and Competing Interests. Thanks again for your time. Changes have been carefully made to follow the correct format.

Q8: - Discussion should be expanded with the comparison to other authors’ results.

Response: Thank you for your comment. We have added a comparison of this work with other reported ones in the revised “Discussion” section (Page 8, Line 238-251), as shown below.

New text added in the “Discussion” section of the Main Text:

Page 8, Line 238-250: “Despite the advancements in extrusion-based 3D printing of gradient structures using the FDA-3DP strategy, some improvements could be made. First, to simplify the compensation mechanism, the printing velocity V of all supporting layers in this study was uniformly set to the constant V_{\max} (corresponding to D_{\min}). However, such an approach may lead to non-monotonic variations in pore gradients within the resulting structure³⁹. Hence, investigating continuous FDA-3D printing without a constant D_{\min} throughout the printing process could prove advantageous and present an exciting avenue for future research. Second, to simplify the CFD simulation model in this study, we derived the relationship between the printing height H and the filament diameter D , expressed in Equation (4) (i.e., $H = k \times D$). It is

worth noting that the filaments produced by the extrusion of viscous inks usually exhibit an oblong cross-section^{43,53,54}. The development of advanced CFD simulation models for the direct calculation of the filament cross-section height l_h (Supplementary Fig. 3) is a promising avenue for future exploration. On this basis, it is clear that $H = l_h$ will be set in FDA-3D printing.”

REVIEWER COMMENTS

Reviewer #1 (Remarks to the Author):

The authors have done an impressive work in addressing all of the comments. The manuscript showcased interesting results for potential applications in many fields. I have no further comment.

Reviewer #2 (Remarks to the Author):

Second review for Manuscript entitled: Gradient matters via filament diameter-adjustable 3D printing, submitted by Huawei Qu et al.

I am impressed by the effort that the authors have made to address all the feedback given, and in its current shape, overall, the work has potential for publication in Nature Coms (NOTE: after final minor comments and some additional results). This manuscript with the minor (but essential) changes detailed below, will provide a valuable contribution to the DIW field. But I do encourage the authors to strengthen their rheological results and discussion for the benefit of the DIW field, and scientific community in general.

Final comments and minor corrections:

1. The authors have included additional rheology measurements and analysis, such as G' and G'' vs temperature. This could be useful but it is not enough for DIW. They should at least include an amplitude sweep (from SAOS to LAOS, small to large amplitude oscillatory shear, 0.01% to at least 500%) to identify the LVR (linear viscoelastic region) and the yielding behaviour (stress vs strain curve from oscillatory experiment) in the MAOS-LAOS region. The measurement should be done at the extrusion/printing temperature. This is the bare minimum, but please consider the following points too. From this experiment you can determine the G' in the LVR region, the yield point (when G' drops below 95% of G'_{LVR}), the critical strain ($G'=G''$) value and the flow stress (plateau region around critical strain) in the stress vs strain plot. These values will give you an idea of the "material strength" and yielding behaviour, both very important in DIW.

2. On another note, the authors chose a 1% strain for the temperature sweep, why? this could be too high and not in the LVR.

3. And what's the purpose of the temperature sweep? Isn't it printing temperature 25C? Again, an amplitude sweep (LAOS) is the test you need to have some insight on the yielding behaviour.

4. in the manuscript: It is worth noting that the filaments produced by the extrusion of viscous (line 248) inks usually exhibit an oblong cross-section^{43,53,54}

Ah! This is because the "inks" are not recovering their initial elasticity upon deposition, but this is not true for all DIW formulations, and it should be avoided (if/when possible) to ensure print quality. To quantify and understand this behaviour (extent and timescale of the recovery) the authors could perform a sequence of two oscillatory tests: the first one in LAOS (at a strain value in which your materials are "yielded" I would expect this to be around $\sim 100\%$), and the second interval in SAOS (at a strain within the LVR!). Calculating the mutation number to quantify the recovery and putting into context with recent literature. If you do this, you will also have some insight on how your materials rebuild (plot G' and G'' vs time).

5. I insist be careful with the terms used, do not refer just as viscous inks. They are viscoelastic inks (G' values are high!), and likely elasto-viscoplastic and not just viscous. It is a common mistake to discuss and refer to the term viscosity in DIW, but this is insufficient. I encourage the authors to catch up with the recent literature linking rheology and printability in DIW of soft materials, and other materials (carbon based etc.) using oscillatory rheology (LAOS).

6. Along the same lines, the model used assumes that the viscosity of a non-Newtonian fluid can be modelled inelastically as a time-independent, i.e. instantaneous, scalar function of the rate of strain. Because of this the model used is not a full representation of the non-Newtonian behaviour. Other models Kelvin Voight, Maxwell, and more complex ones such as Saramito, KBR do account for the elastic behaviour too. Again, viscosity is not sufficient to describe complex fluids in DIW. Often for DIW "inks" the continuous shear data are not reliable. I encourage the authors to explicitly state that the model is used as a simple approximation of the flow behaviour (not considering elastic contributions, normal stresses etc.). But these "fluids" are not isotropic.

7. Supplementary Fig. 7 The fitting to the models are awful, especially in the shear thinning

region. Which means the n values obtained are pointless. You will be better off, just fitting the shear-thinning region to a simple power law behaviour to have a more accurate determination of the n (flow index) value. Don't put those fittings in the publication, either use a simple power law in the shear thinning region, or work on a better fitting adjusting all the parameters in the model.

8. Note that G'' , storage modulus (Pa), is not the viscosity.

9. I insist again, please do review the recent literature linking rheology and printability in DIW and hopefully that will help you improve your analysis and interpretation.

In short, there are still terms and discussions (in rheology and fluid mechanics) that could be further polished throughout the manuscript.

The rheology characterisation must be improved as explained above to ensure the data provided is relevant to DIW (temperature sweep is not useful, unless the temperature is playing a role in the printing process), and sound from a rheologist perspective as well.

The strength of this work and main contribution to the DIW is the G-code to create gradient structures, and this is worthy of publication, but the authors must ensure that questioning of the underlying fundamental aspects are fully resolved.

Reviewer #3 (Remarks to the Author):

The paper has improved significantly.

However, references 32 to 40 should be explained separately in more detail.

Point-by-point Responses to the Reviewers' Comments

Dear reviewers,

We thank all reviewers for re-reviewing our manuscript (entitled: *Gradient matters via filament diameter-adjustable 3D printing*). We value your constructive feedback and comments on this work. In response to your suggestions, we have carefully revised the manuscript, and the updated portions are highlighted in yellow. Below are point-by-point responses to the reviewers' comments and questions. Black indicates the original comments from the reviewers, and our responses are marked in blue.

Responses to Reviewer #1:

The authors have done an impressive work in addressing all of the comments. The manuscript showcased interesting results for potential applications in many fields. I have no further comment.

Response: We would like to thank reviewer #1 for your thorough review of our manuscript and for helping us improve its quality.

Responses to Reviewer #2:

Second review for Manuscript entitled: Gradient matters via filament diameter-adjustable 3D printing, submitted by Huawei Qu et al.

I am impressed by the effort that the authors have made to address all the feedback given, and in its current shape, overall, the work has potential for publication in Nature Coms (NOTE: after final minor comments and some additional results). This manuscript with the minor (but essential) changes detailed below, will provide a valuable contribution to the DIW field. But I do encourage the authors to strengthen their rheological results and discussion for the benefit of the DIW field, and scientific community in general.

Response: We sincerely acknowledge your appreciation of our efforts. And we thank you for the constructive comments on our manuscript and for helping us improve its quality. Following your suggestions on rheology, we have carefully modified them in the newly revised manuscript. Details can be found in the following point-to-point response. We sincerely ask you to re-evaluate the possibility of our work to be published in *Nature Communications*.

Final comments and minor corrections:

Q1: The authors have included additional rheology measurements and analysis, such as G' and

G'' vs temperature. This could be useful but it is not enough for DIW. They should at least include an amplitude sweep (from SAOS to LAOS, small to large amplitude oscillatory shear, 0.01% to at least 500%) to identify the LVR (linear viscoelastic region) and the yielding behaviour (stress vs strain curve from oscillatory experiment) in the MAOS-LAOS region. The measurement should be done at the extrusion/printing temperature. This is the bare minimum, but please consider the following points too. From this experiment you can determine the G' in the LVR region, the yield point (when G' drops below 95% of G'_{LVR}), the critical strain ($G'=G''$) value and the flow stress (plateau region around critical strain) in the stress vs strain plot. These values will give you an idea of the “material strength” and yielding behaviour, both very important in DIW.

Response: Greatly thanks for your comments, we fully agree that ink rheology is crucial for DIW-based 3D printing. Based on your comments, we have reviewed literature and online sources on amplitude oscillatory shear sweeps from SAOS to LAOS, enhancing our understanding of oscillatory rheology. To our knowledge, for viscoelastic inks with different rheological properties, there exist two distributions of the storage modulus (G') and loss modulus (G'') versus strain (γ), as summarized in the response Figure R1 below. γ_L denotes the boundary of the linear viscoelastic region (LVR). One (see response Figure R1-a) is that in the LVR (i.e., SAOS), the storage modulus (G') is higher than the loss modulus (G''), and in the non-LVR (i.e., LAOS), both the storage modulus (G') and loss modulus (G'') gradually decrease, and the loss modulus (G'') is higher than the storage modulus (G') after crossing. The other (see response Figure R1-b) is that the loss modulus (G'') is higher than the storage modulus (G') throughout the amplitude sweep. It is well known that the storage modulus (G') and loss modulus (G'') indicate the elastic and viscous behavior of materials, respectively. In the response Figure R1-a, within the LVR, G' is greater than G'' ($G' > G''$) and the material is a solid in the gel state (elastic); within the non-LVR, the material gradually transforms from solid to liquid (shear thinning). In the response Figure R1-b, the loss modulus (G'') is higher than the storage modulus (G') throughout the amplitude sweep, so the material consistently remains liquid (viscous).

response Figure R1 | The storage modulus (G') and loss modulus (G'') versus strain (γ). The linear region

represents the linear viscoelastic region (LVR), which is also the small amplitude oscillatory shear (SAOS). The nonlinear region represents the nonlinear viscoelastic region (non-LVR), which is also the large amplitude oscillatory shear (LAOS). The long dashed lines in both (a) and (b) show the LVR boundaries (γ_L). The short dashed line in (a) indicates the crossover point of the storage modulus (G') and loss modulus (G'') in the logarithmic coordinate system. Within the LVR of (a), G' is greater than G'' ($G' > G''$) and the material is a solid in the gel state (elastic); within the LVR of (b), G'' is greater than G' ($G'' > G'$) and the material is a liquid in a fluid (viscous).

In response to your concerns, using an Anton Paar rheometer (MCR302, Austria), we have additionally conducted amplitude sweeps for the H-PCL and L-PCL materials used in this work at a fixed angular frequency of 10 rad/s, as shown in Supplementary Fig. 6e-f. The amplitude oscillatory shear ranges from 0.01% to 3000%. The results of the storage modulus (G') and loss modulus (G'') versus strain (γ) indicate that the rheological properties of H-PCL and L-PCL inks are in accordance with the description in the response Figure R1-b. The LVR boundaries are up to 400% strain for H-PCL inks and over 3000% for L-PCL inks.

Supplementary Fig. 6e-f | e, Storage G' and loss G'' modulus versus strain γ of the H-PCL ink for the amplitude sweep (from 0.01% to 3000%) through oscillatory measurement at 110°C working temperature. f, Storage G' and loss G'' modulus versus strain (γ) of the L-PCL ink for the amplitude sweep (from 0.01% to 3000%) through oscillatory measurement at 70°C working temperature.

It is important for you to note that in this study, to clearly demonstrate the pore structure inside the printed scaffolds we chose hot-melt polymers (H-PCL and L-PCL) as the printing ink instead of hydrogels (e.g. GelMA). This is because polymers provide superior μ -CT imaging than hydrogels. In addition, to further enhance the 3D imaging of the printed gradient samples after scanning with a μ -CT device, we also homogeneously mixed the nano- β -TCP particles in the hot-melt polymer material. The working temperatures of H-PCL and L-PCL are 110 °C and 70 °C, respectively. The liquid hot-melt polymer transforms into solid structures upon being extruded in contact with room-temperature air (~ 25 °C), which is why we provided rheological data from temperature sweep in the previous revised manuscript (Supplementary Fig. 6c-d).

In conclusion, following your comments, we have newly implemented the amplitude sweeps for the H-PCL and L-PCL materials at 110°C and 70°C, and added corresponding Supplementary Fig. 6e-f and experimental methods (*Rheological characterizations*, Page 10, Lines 304-307) in the newly revised manuscript.

New text (*Rheological characterizations*) added in the “Methods” section:

Page 10, Lines 304-307: “The storage modulus G' and loss modulus G'' vs. strain γ were measured using amplitude oscillatory shear sweeps from 0.01% to 3000% at a fixed angular frequency of 10 rad/s. In amplitude sweeps, the working temperatures of H-PCL and L-PCL are 110 °C and 70 °C, respectively.”

New Supplementary Fig. 6e-f added in the Supplementary Information:

Supplementary Fig. 6 | Rheological characteristics of the polymer inks (L-PCL and H-PCL) used in this

study. **a**, Dynamic viscosity of L-PCL and H-PCL at 70 °C and 110 °C, respectively. Both L-PCL and H-PCL exhibit non-Newtonian behavior, demonstrating the shear-thinning phenomenon at a high shear rate (about 10^2 s^{-1}). Three repetitions were conducted. The shear rate is from 10^{-3} s^{-1} to 10^3 s^{-1} . The zero-shear viscosities η_0 of L-PCL and H-PCL are approximately 382.299 Pa·s and 36.323 Pa·s, respectively. **b**, Relationship between shear stress and shear strain using rotational tests with controlled shear stress (CSS). Both L-PCL and H-PCL exhibit the yield stress. **c**, Storage G' and loss G'' modulus of the H-PCL ink for temperature scanning (from 110 °C to 25 °C) through oscillatory measurement. **d**, Storage G' and loss G'' modulus of the L-PCL ink for temperature scanning (from 70 °C to 25 °C) through oscillatory measurement. During material deformation, elasticity, and viscosity are measured by the storage modulus G' and loss modulus G'' , respectively. **e**, Storage G' and loss G'' modulus versus strain γ of the H-PCL ink for the amplitude sweep (from 0.01% to 3000%) through oscillatory measurement at 110 °C working temperature. **f**, Storage G' and loss G'' modulus versus strain (γ) of the L-PCL ink for the amplitude sweep (from 0.01% to 3000%) through oscillatory measurement at 70 °C working temperature.

Q2: On another note, the authors chose a 1% strain for the temperature sweep, why? this could be too high and not in the LVR.

Response: Thank you for your comments. In the previous revised manuscript, according to the experience, we chose a 1% strain for the temperature sweep. Based on your Comment Q1 above, we have newly plotted the storage modulus (G') and loss modulus (G'') versus strain (γ) from 0.01% to 3000% through amplitude oscillatory shear sweeps, as shown in Supplementary Fig. 6e-f (see below). The added results indicate that, for both H-PCL and L-PCL materials, the 1% strain is in the linear viscoelastic region (LVR).

Supplementary Fig. 6e-f | **e**, Storage G' and loss G'' modulus versus strain γ of the H-PCL ink for the amplitude sweep (from 0.01% to 3000%) through oscillatory measurement at 110 °C working temperature. **f**, Storage G' and loss G'' modulus versus strain (γ) of the L-PCL ink for the amplitude sweep (from 0.01% to 3000%) through oscillatory measurement at 70 °C working temperature.

Q3: And what's the purpose of the temperature sweep? Isn't it printing temperature 25C? Again, an amplitude sweep (LAOS) is the test you need to have some insight on the yielding behaviour.

Response: We thank you for pointing out this concern. Notably, to clearly demonstrate the pore structure inside the printed scaffolds we chose hot-melt polymers as the printing ink instead of hydrogel materials (e.g. GelMA) due to the superior μ -CT-based 3D imaging of polymers over hydrogels. Temperature sweeps were utilized to evaluate the rheological properties of the hot-melt polymer materials (H-PCL and L-PCL). The working temperatures of H-PCL and L-PCL in the 3D printing barrel were 110°C and 70°C, respectively. The hot-melt polymer transforms into solid structures upon being extruded in contact with room-temperature air. We assumed a room temperature of 25°C. Therefore, H-PCL and L-PCL were subjected to temperature sweeps from 110°C to 25°C and from 70°C to 25°C, respectively, as shown in Supplementary Fig. 6c-d. In addition, as in response to your Comment Q1 above, based on your suggestion, we have newly implemented the amplitude sweep as shown in Supplementary Fig. 6e-f.

Supplementary Fig. 6c-d | c, Storage G' and loss G'' modulus of the H-PCL ink for temperature scanning (from 110 °C to 25 °C) through oscillatory measurement. d, Storage G' and loss G'' modulus of the L-PCL ink for temperature scanning (from 70 °C to 25 °C) through oscillatory measurement. During material deformation, elasticity, and viscosity are measured by the storage modulus G' and loss modulus G'' , respectively.

Supplementary Fig. 6e-f | e, Storage G' and loss G'' modulus versus strain γ of the H-PCL ink for the

amplitude sweep (from 0.01% to 3000%) through oscillatory measurement at 110°C working temperature. **f**, Storage G' and loss G'' modulus versus strain (γ) of the L-PCL ink for the amplitude sweep (from 0.01% to 3000%) through oscillatory measurement at 70°C working temperature.

Q4: in the manuscript: It is worth noting that the filaments produced by the extrusion of viscous (line 248) inks usually exhibit an oblong cross-section^{43,53,54}

Ah! This is because the “inks” are not recovering their initial elasticity upon deposition, but this is not true for all DIW formulations, and it should be avoided (if/when possible) to ensure print quality. To quantify and understand this behaviour (extent and timescale of the recovery) the authors could perform a sequence of two oscillatory tests: the first one in LAOS (at a strain value in which your materials are “yielded” I would expect this to be around ~100%), and the second interval in SAOS (at a strain within the LVR!). Calculating the mutation number to quantify the recovery and putting into context with recent literature. If you do this, you will also have some insight on how your materials rebuild (plot G' and G'' vs time).

Response: Thank you for your insights. We sincerely appreciate your constructive comments on the phenomenon of the presence of deposited filaments with an oblong cross-section. We believe that the nature of this problem can be explored through the two oscillatory tests within LAOS and SAOS, respectively, that you described above. Unfortunately, in this work, we mainly used the hot-melt polymer material H-PCL to demonstrate the ability of our proposed filament diameter-adjustable 3D printing (FDA-3DP) strategy to fabricate gradient structures. Oscillatory sweeps under sudden temperature changes between 110°C and 25°C using an Anton Paar rheometer is a difficult job, which is beyond the capabilities of the rheometers available in our laboratory. Experiments showed that it usually takes several minutes for the test temperature of a rheometer to drop from 110°C to 25°C. However, in actual extrusion-based 3D printing, it often takes only a few tens of seconds for H-PCL at 110°C to be extruded from the barrel and deposited and molded into filaments with an oblong cross-section at 25°C. In the future, the computational fluid dynamics (CFD) simulation is a promising approach to solve this issue, and this is the next project we are going to carry out.

Q5: I insist be careful with the terms used, do not refer just as viscous inks. They are viscoelastic inks (G' values are high!), and likely elasto-viscoplastic and not just viscous. It is a common mistake to discuss and refer to the term viscosity in DIW, but this is insufficient. I encourage the authors to catch up with the recent literature linking rheology and printability in DIW of soft materials, and other materials (carbon based etc.) using oscillatory rheology (LAOS).

Response: Thank you for your comment. Based on your Comment Q1 above, we have newly

plotted the storage modulus (G') and loss modulus (G'') versus strain (γ) from 0.01% to 3000% through amplitude oscillatory shear sweeps, as shown in Supplementary Fig. 6e-f (see the figure below). The added results show that both H-PCL and L-PCL materials exhibit the linear viscoelastic region (LVR) and they both behave as liquids in the print barrel. In addition, in response to this comment, we have replaced all “viscous” with “viscoelastic” in the latest revised manuscript.

Supplementary Fig. 6e-f | e, Storage G' and loss G'' modulus versus strain γ of the H-PCL ink for the amplitude sweep (from 0.01% to 3000%) through oscillatory measurement at 110°C working temperature. f, Storage G' and loss G'' modulus versus strain (γ) of the L-PCL ink for the amplitude sweep (from 0.01% to 3000%) through oscillatory measurement at 70°C working temperature.

Q6: Along the same lines, the model used assumes that the viscosity of a non-Newtonian fluid can be modelled inelastically as a time-independent, i.e. instantaneous, scalar function of the rate of strain. Because of this the model used is not a full representation of the non-Newtonian behaviour. Other models Kelvin Voight, Maxwell, and more complex ones such as Saramito, KBR do account for the elastic behaviour too. Again, viscosity is not sufficient to describe complex fluids in DIW. Often for DIW “inks” the continuous shear data are not reliable. I encourage the authors to explicitly state that the model is used as a simple approximation of the flow behaviour (not considering elastic contributions, normal stresses etc.). But these “fluids” are not isotropic.

Response: We are grateful for your insights. Following your comments, we have newly added the text description that the Carreau Yasuda model is only a simple approximation of the flow behavior without considering elastic contributions and normal stresses in the *CFD simulation of ink flow* of the “Methods” section (Page 15, Lines 473-475). In addition, we have deleted the word “isotropic” in the *CFD simulation of ink flow* of the “Methods” section (Page 15, Line 486).

New text (CFD simulation of ink flow) added in the “Methods” section:

Page 15, Lines 473-475: “Note that the Carreau Yasuda model is only a simple approximation of the flow behavior without considering elastic contributions and normal stresses.”

Q7: Supplementary Fig. 7 The fitting to the models are awful, especially in the shear thinning region. Which means the n values obtained are pointless. You will be better off, just fitting the shear-thinning region to a simple power law behaviour to have a more accurate determination of the n (flow index) value. Don't put those fittings in the publication, either use a simple power law in the shear thinning region, or work on a better fitting adjusting all the parameters in the model.

Response: Thank you for your comment. Indeed, after you pointed this out, we also recognized the poor fitting performance of the old Supplementary Fig. 7 in the shear thinning region, as depicted in (a) and (b) of response Figure R2. Following your suggestion, we have conducted a least-squares fitting of the Carreau Yasuda model by re-adjusting all parameters using the software OriginPro 2022 (OriginLab Corporation, USA), as shown in (c) and (d) of response Figure R2. The results reveal that the new fitting is significantly better than the old one. Further, based on the newly fitted material constitutive equations, we have redone the computational fluid dynamics (CFD) simulations and updated corresponding simulation results in the newly revised manuscript, including Fig. 2a-c, new Supplementary Fig. 7 (corresponding to old Supplementary Fig. 8), and new Supplementary Fig. 9 (corresponding to old Supplementary Fig. 10).

Additionally, in response to your comment, we have removed the figure of the fitting results from the Supplementary Information in the newly revised manuscript (i.e., the old Supplementary Fig. 7) and only show the new fittings in the “Methods” section (see Page 15, Lines 475-479).

New text (CFD simulation of ink flow) added in the “Methods” section:

Page 15, Lines 475-479: “All parameters of the Carreau Yasuda model were fitted using software OriginPro 2022 (OriginLab Corporation, USA) by the least squares (for H-PCL ink, $\eta_0=392.805\pm 1.516$ Pa s, $\eta_\infty=0\pm 36.563$ Pa s, $\lambda=0.010\pm 0.003$ s, $a=1.946\pm 0.322$, and $n=0\pm 0.444$; for L-PCL ink, $\eta_0=36.436\pm 0.072$ Pa s, $\eta_\infty=0\pm 15.263$ Pa s, $\lambda=0.004\pm 0.001$ s, $a=4.129\pm 1.117$, and $n=0.319\pm 0.551$).”

Old Carreau Yasuda model fitting

New Carreau Yasuda model fitting

response Figure R2 | The top (a and b) and bottom (c and d) pictures are the old and new fitted Carreau Yasuda models, respectively. The left (a and c) and right (b and d) pictures show H-PCL and L-PCL materials, respectively. In (a), the zero shear viscosity η_0 is 382.299 ± 4.019 Pa s, the infinite shear viscosity η_{∞} is 0 ± 607.208 Pa s, the relaxation time λ is 0.051 ± 0.020 s, the flow behavior index n is 0.802 ± 0.426 , and the Yasuda index a is 217.786 ± 614433.352 . In (b), the zero shear viscosity η_0 is 36.323 ± 0.302 Pa s, the infinite shear viscosity η_{∞} is 0 ± 109.588 Pa s, the relaxation time λ is 0.014 ± 0.007 s, the flow behavior index n is 0.840 ± 0.605 , and the Yasuda index a is $202.374 \pm 6.234 \cdot 10^{11}$. In (c), the zero shear viscosity η_0 is 392.805 ± 1.516 Pa s, the infinite shear viscosity η_{∞} is 0 ± 36.563 Pa s, the relaxation time λ is 0.010 ± 0.003 s, the flow behavior index n is 0 ± 0.444 , and the Yasuda index a is 1.946 ± 0.322 . In (d), the zero shear viscosity η_0 is 36.436 ± 0.072 Pa s, the infinite shear viscosity η_{∞} is 0 ± 15.263 Pa s, the relaxation time λ is 0.004 ± 0.001 s, the flow behavior index n is 0.319 ± 0.551 , and the Yasuda index a is 4.129 ± 1.117 .

Q8: Note that G'' , storage modulus (Pa), is not the viscosity.

Response: Thank you for your comment. In response to your concern, we have reviewed the literature and found that the storage modulus (G') and loss (G'') modulus indicate the elastic and viscous behavior of the material, respectively. As you pointed out, the unit of the storage modulus (G') and loss (G'') modulus is Pa and the unit of viscosity (η) is Pa s.

Q9: I insist again, please do review the recent literature linking rheology and printability in DIW and hopefully that will help you improve your analysis and interpretation.

Response: We are grateful for your helpful comments to improve the quality of our manuscript. We review the recent literature¹⁻³ linking rheology and printability in DIW.

1. Rau, D. A., Williams C. B. & Bortner M. J. Rheology and printability: A survey of critical relationships for direct ink write materials design. *Prog. Mater. Sci.* **140**, 101188 (2023).
2. Kamkar, M., *et al.* Large amplitude oscillatory shear flow: Microstructural assessment of polymeric systems. *Prog. Polym. Sci.* **132**, 101580 (2022).
3. Rau, D. A., Bortner M. J. & Williams C. B. A rheology roadmap for evaluating the printability of material extrusion inks. *Addit. Manuf.* **75**, 103745 (2023).

Q10: In short, there are still terms and discussions (in rheology and fluid mechanics) that could be further polished throughout the manuscript.

The rheology characterisation must be improved as explained above to ensure the data provided is relevant to DIW (temperature sweep is not useful, unless the temperature is playing a role in the printing process), and sound from a rheologist perspective as well.

The strength of this work and main contribution to the DIW is the G-code to create gradient structures, and this is worthy of publication, but the authors must ensure that questioning of the underlying fundamental aspects are fully resolved.

Response: Thanks for your time. Following your comments, we have improved the description involving rheology, such as replacing “viscous” with “viscoelastic” and removing the word “isotropic” in the latest revised manuscript. In addition, we have performed amplitude sweep tests. Since inks H-PCL and L-PCL are hot-melt polymers, which solidify to a solid upon contact with room temperature air (about 25°C) during 3D printing, temperature sweeps may provide some helpful information to the reader. Finally, thank you for recognizing our work on creating gradient structures through G-codes, and also for helping us improve the quality of this manuscript.

Responses to Reviewer #3:

The paper has improved significantly. However, references 32 to 40 should be explained separately in more detail.

Response: We would like to thank reviewer #3 for your careful review of our manuscript and for helping us improve its quality. In response to your comment about references 32 to 40, we have described them in more detail in Supplementary Table 1 of Supplementary Information and the “Results” section of the Main Text (Page 3, Line 64) in the newly revised manuscript. A detailed description of references 32-40 is provided in Supplementary Table 1, as shown below.

Supplementary Table 1 | A comparison between the FDA-3DP strategy and the reported DIW 3D printing methods for creating gradient pore structures, focusing on gradient dimension, gradient resolution, and shape fidelity. A check mark (✓) denotes feasibility or compatibility, while a wrong mark (✗) denotes infeasibility or unattainability. A hyphen mark (-) indicates not mentioned in the literature.

Changing parameters	Gradient dimension ¹				Gradient resolution ²	Shape fidelity ³		Ref.
	1D			2D & 3D		Resolution level	Fidelity level	
	Horizontal	Axial	Radial					
Diameter/spacing/angle within layers	✓	✗	✗	✗	Low	Low	✗	32,34,36,37
Diameter/spacing between layers	✗	✓	✗	✗	Medium	Medium	✓	33,35
Diameter/spacing in radial regions	✗	✗	✓	✗	Medium	Low	✗	36,37,51
Width of oblong filaments	✓	✓	✓	2D ✓ 3D ✗	High	Low	✗	38-40
Diameter & selectively adding support layers	✓	✓	✓	✓	High	High	✓	This work

¹ Gradient dimension includes 1D, 2D, and 3D. 1D represents one direction of the *x-y-z* coordinate axis.

² Gradient resolution indicates the ability to create gradient pores.

³ Shape fidelity is the difference between the designed model and the fabricated sample.

REVIEWERS' COMMENTS

Reviewer #2 (Remarks to the Author):

Third and final review for Manuscript entitled: Gradient matters via filament diameter-adjustable 3D printing, submitted by Huawei Qu et al.

The authors have diligently addressed all the feedback given and edited the manuscript accordingly, which has significantly improved (rheology characterisation, model fits etc.). They have also added new and recent references in the rheology aspects. My only comment on the latter, is that it is preferable to cite relevant original work, instead of broad reviews/perspectives. I have no additional comments.